# Refractoriness of STING therapy is relieved by AKT inhibitor through effective vascular disruption in tumour

Seung-hwan Jeong[1], Myung Jin Yang[1,2], Seunghyeok Choi[1,2], JungMo Kim[2] & Gou Young Koh [1,2✉]

Stimulator of interferon genes (STING) promotes anti-tumour immunity by linking innate and adaptive immunity, but it remains unclear how intratumoural treatment with STING agonists yields anti-tumour effects. Here we demonstrate that intratumoural injection of the STING agonist cGAMP induces strong, rapid, and selective apoptosis of tumour endothelial cells (ECs) in implanted LLC tumour, melanoma and breast tumour, but not in spontaneous breast cancer and melanoma. In both implanted and spontaneous tumours, cGAMP greatly increases TNFα from tumour-associated myeloid cells. However, compared to spontaneous tumour ECs, implanted tumour ECs are more vulnerable to TNFα-TNFR1 signalling-mediated apoptosis, which promotes effective anti-tumour activity. The spontaneous tumour's refractoriness to cGAMP is abolished by co-treatment with AKT 1/2 inhibitor (AKTi). Combined treatment with cGAMP and AKTi induces extensive tumour EC apoptosis, leading to extensive tumour apoptosis and marked growth suppression of the spontaneous tumour. These findings propose an advanced avenue for treating primary tumours that are refractory to single STING agonist therapy.

---

[1] Graduate School of Medical Science and Engineering, Korea Advanced Institute of Science and Technology, Daejeon, Republic of Korea. [2] Center for Vascular Research, Institute for Basic Science, Daejeon, Republic of Korea. ✉email: gykoh@kaist.ac.kr

Stimulator of interferon genes (STING), encoded by *TMEM173*, activates the innate immune system in response to cytosolic double-stranded DNA derived from viral or bacterial infection, chemical- or irradiation-induced cellular damage, and DNA leakage from the nucleus or mitochondria due to pathologic conditions, such as cancer[1–4]. DNA sensor, cyclic GMP-AMP synthase (cGAS), produces STING-activating ligand, cyclic GMP-AMP (cGAMP), from cytosolic ATP and GTP[5]. Upon activation via binding to cGAMP, STING is transferred from the endoplasmic reticulum to the Golgi apparatus, where it triggers Tank-binding kinase 1 followed by downstream signal activations including the nuclear factor-κB pathway and type I interferon production[6,7]. Enhanced interferon production induces cytotoxic effects directly against cancer cells, as well as activating dendritic cell (DC) maturation and promoting CD8+ T cell priming in tumour-draining lymph nodes (TDLN)[8–10]. Throughout these actions, STING agonists shift anti-tumour immunity from immunologically silenced "cold" tumours to active "hot" tumours[11]. By doing so, they have been considered as synergistic agents for immune checkpoint inhibitors targeting PD-1 and PD-L1, which rescue exhausted T cells to kill tumour cells[12–14]. Although STING agonists has been tried for cancer therapy under tremendous attention, underlying mechanisms behind the anti-tumour effects are yet poorly understood[15,16].

Murine-specific STING agonist, 5,6-dimethylxanthenone-4-acetic acid (DMXAA), which also has been known as a tumour vascular disrupting agent (VDA), exerts a potent anti-tumour effect that is caused by tumour endothelial cell (EC)-specific apoptosis and extensive haemorrhage within tumour[17]. Tumour antigen release from dead tumour cells is the first step in adaptive immunity generation; therefore, tumour vascular destruction by STING agonists might be indispensable for achieving sufficient anti-tumour immunity[18–20]. Notably, TNFα has been proposed as a mediator of STING-induced tumour EC apoptosis[21,22]. However, the source of TNFα upon STING activation remains unknown, and it is unclear why STING activation triggers apoptosis specifically to tumour ECs. Moreover, most previous studies have been conducted in subcutaneous tumour implantation models, which have limited ability to reflect human primary tumours[23–25].

Activation of intracellular AKT signalling plays a critical role in survival in several cell types, including ECs[26–28]. Of note, AKT is a major downstream molecule for conveying intracellular signalling of vascular growth factors and their receptors including angiopoietin-1/Tie2 and VEGF-A/VEGFR2[29,30], which are key molecules in tumour angiogenesis. Accordingly, the AKT pathway has been considered an attractive therapeutic target since AKT hyper-activation is associated with tumour aggressiveness and poor response to treatment. However, despite promising results in preclinical models, clinical trials of AKT inhibitors have failed to prove effectiveness, and none of the tested agents are currently used for cancer treatment[31,32].

Here, we show that intratumoural STING agonist induces effective apoptosis of tumour endothelial cells (ECs) in implanted tumours, but not in spontaneous tumours. The spontaneous tumour's refractoriness to STING agonist is abolished by additional treatment with AKT 1/2 inhibitor (AKTi) through effectual apoptosis of tumour ECs.

## Result

### STING agonist cGAMP is a strong tumour vascular disrupting agent.
To explore the effects of a STING agonist on tumour vessels and tumour growth, we used an established subcutaneous LLC tumour model (Fig. 1a). As expected, intratumoural (i.t.) injection of cGAMP (14 μg/70 μl of PBS) suppressed tumour growth by 84% compared with PBS alone (Fig. 1b, c). To evaluate the initial effects of cGAMP on the tumour microenvironment, we performed i.t. injection of cGAMP or PBS into LLC tumours, and sampled the tumours 24 h later (Fig. 1d). Strikingly, in cGAMP-treated tumours, 37% of ECs were positive for the apoptosis marker cleaved caspase3 (C.Casp3), while C.Casp3 positivity was observed in <2% of ECs in PBS-treated tumours. Moreover, cGAMP-induced apoptosis in 43% of all cells within tumour (hereafter described as "whole tumour cells"), while PBS induced apoptosis in 6% of whole tumour cells (Fig. 1e, f). C. Casp3+ apoptosis was detected mainly in tumour ECs at 3 h after cGAMP injection, was additionally detected in the surrounding cells at 6 h, and was prevalent in almost all cells at 24 h (Fig. 1g). Flow cytometry analysis revealed a reduced population of live ECs and increased number of dead cells as early as 3 h after cGAMP injection, and these changes further progressed within 24 h (Fig. 1h–k). Likewise, the systemic murine STING agonist DMXAA induced apoptosis in 47% of ECs of tumour vessels, and in 47% of whole tumour cells at 24 h (Supplementary Fig. 1a–c). No apparent apoptotic signals were detected in the ECs of other organs in the tumour-bearing mice treated with DMXAA (Supplementary Fig. 1d). Additionally, cGAMP doses ranging from 0.1 μM to 1.0 mM did not alter the cellular viabilities of cultured LLC cells or HUVECs (Supplementary Fig. 1e). These findings implied that the apoptosis of tumour ECs and tumour cells was not directly derived from cGAMP cytotoxicity. Further analyses revealed that cGAMP increased vascular leakage by 2.8-fold, induced red blood cell leakage by 12.2-fold, reduced blood perfusion by 54%, and increased hypoxia by 9.2-fold at 24 h (Fig. 1l, m). Similar findings were observed in the B16F10 melanoma implantation model. Intratumoural cGAMP injections suppressed tumour growth by 95% (Supplementary Fig. 1f–h). Apoptosis was observed in 24% of tumour ECs, and in 27% of whole tumour cells (Supplementary Fig. 1i–k). These findings indicated that cGAMP is a strong and rapid tumour vascular disrupting agent.

### STING pathway of bystander cells plays a key role in tumour EC apoptosis.
STING plays diverse roles in tumour growth depending on cells composing tumour cells or tumour microenvironment[3,33–35]. To further examine the initial effects of cGAMP on tumour EC apoptosis and anti-tumour growth, we used the LLC tumour model with *STING* KO mice (Goldenticket, *STING*gt/gt)[13] (Fig. 2a). The cGAMP-induced tumour EC apoptosis and anti-tumour growth were completely abrogated in *STING* KO mice (Fig. 2b–f). Moreover, in *STING* KO mice, cGAMP-treated tumours did not show a reduced live tumour EC population, or increased dead cell population (Fig. 2g–j). To dissect the role of intrinsic STING in tumour cells, STING in LLC cells was depleted via transduction of a lentiviral vector encoding shSTING (Fig. 3a, b). As a control, the cells were transduced with shCon or nothing (NOT) (Fig. 3a, b). LLCs transduced with shSTING and shCon did not differ in cGAMP-induced anti-tumour growth (each by 93%) (Fig. 3c–e), vascular disruption following tumour EC apoptosis (28% and 21%, respectively), or tumour apoptosis (34% and 29%, respectively) (Fig. 3f–i). Furthermore, to evaluate whether the cGAMP-induced tumour EC apoptosis is mediated directly through STING activation in tumour ECs, we generated an inducible, endothelial cell-specific STING-deleted mice (*STING*iΔEC) by crossing *STING*fl/fl mice with *VE-cadherin*-Cre-ERT2 mice[36] (Supplementary Fig 2a). Littermates of *VE-cadherin*-Cre-ERT2 were used as control wild type (WT) mice. Two weeks before LLC implantation, tamoxifen (2 mg) was administered to WT and *STING*iΔEC mice for five

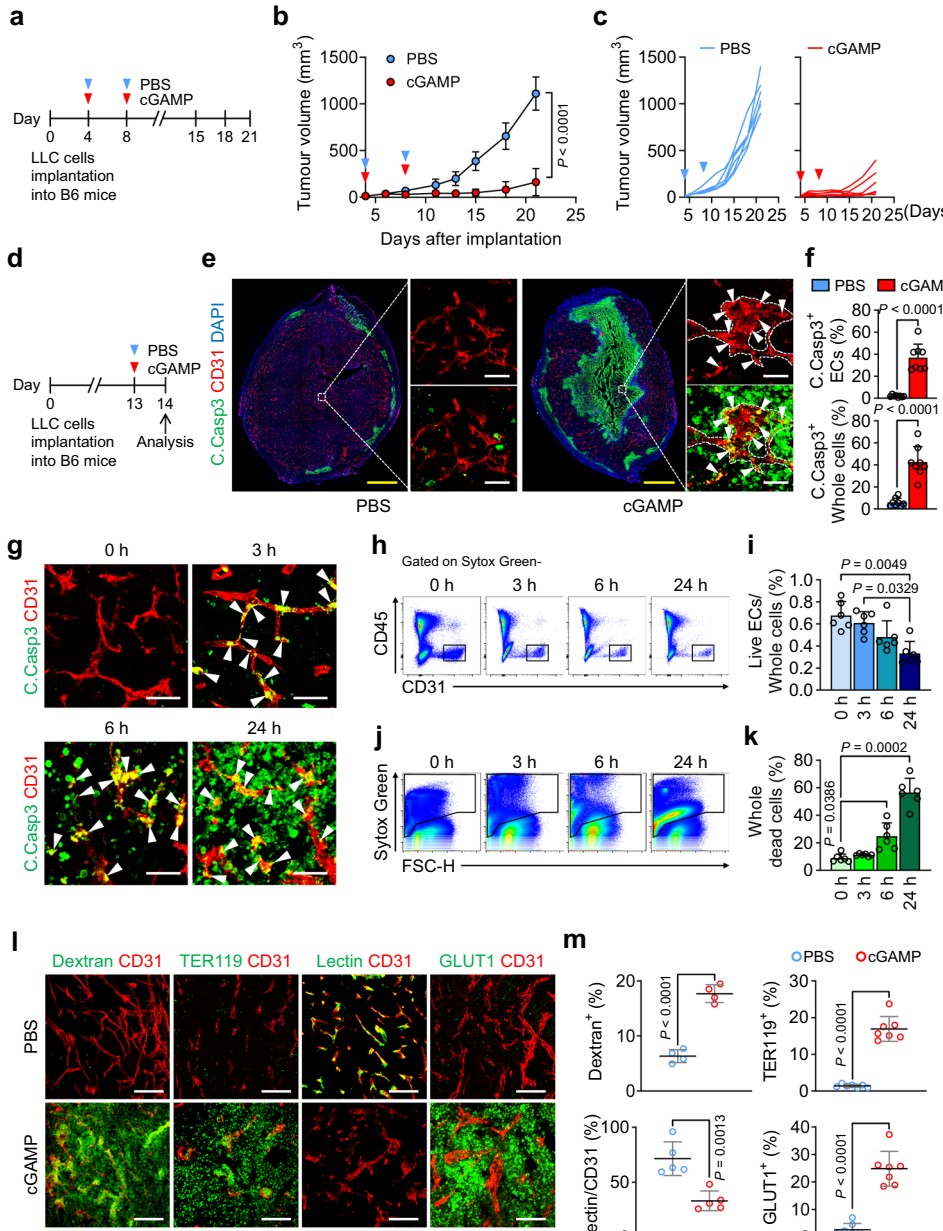

**Fig. 1 Intratumoural injection of cGAMP strongly induces tumour vascular disruption in implanted LLC tumour. a–c** Diagram depicting generation of implanted LLC tumour in B6 mice and treatment schedule of intratumoural (i.t.) PBS or cGAMP. Comparison of LLC tumour growths. $n = 6$ mice/group from two independent experiments. Dots and bars indicate mean ± SD. Plot indicates each individual tumour growth. **d–f** Diagram depicting generation of LLC tumour, i.t. PBS or cGAMP treatment, and tumour sampling at 24 h later. Representative images and comparisons of apoptosis in tumour ECs and whole tumour cells (whole cells). Dashed lines demarcate tumour vascular lining, while white arrowheads indicate apoptotic ECs. Scale bars, 1.0 mm (yellow bars) and 50 μm (white bars). Each dot indicates a value from one mouse and $n = 8$ mice/group from two independent experiments. Vertical bars indicate mean ± SD. **g–k** Representative images showing temporal responses of tumour ECs and whole tumour cells following i.t. cGAMP treatment in implanted LLC tumour. $n = 6$ mice/group. Note that progressive tumour EC apoptosis followed by extensive tumour cell apoptosis. Scale bars, 50 μm (**g**). Representative flow cytometry plots and comparisons showing gradual changes of live EC and dead cell populations. Each dot indicates a value from one mouse and $n = 6$ mice/group from three independent experiments. Vertical bars indicate mean ± SD. **l, m** Impaired vascular function 24 h after i.t. cGAMP treatment. Representative Images and comparisons of dextran leakage ($n = 4$ images/4 mice), TER119+ RBC leakage ($n = 7$ images/7 mice), lectin+ vascular perfusion ($n = 5$ images/5 mice) and GLUT1+ hypoxic area ($n = 7$ images/7 mice). Scale bars, 100 μm. Each dot indicates a value from one mouse from four independent experiments. Horizontal bars indicate mean ± SD. $P$ values by two-tailed t-test (**b, f, m**), Kruskal–Wallis test followed by Dunn's test (**i**) or Welch's one-way ANOVA test followed by Dunnett's T3 test (**k**). Source data are provided as a Source Data File.

times with 2-day interval (Supplementary Fig 2b). Of note, there were no significant differences in tumour growth, tumour EC apoptosis and whole tumour cell apoptosis between WT and $STING^{i\Delta EC}$ mice (Supplementary Fig. 2c–g), implying that the cGAMP-induced tumour EC apoptosis was not mediated through

direct STING activation of tumour ECs. Together, the cGAMP-induced tumour vascular destruction and anti-tumour growth could be derived from STING activation of bystander cells in the tumour microenvironment, rather than from intrinsic STING activation of tumour cells or tumour ECs.

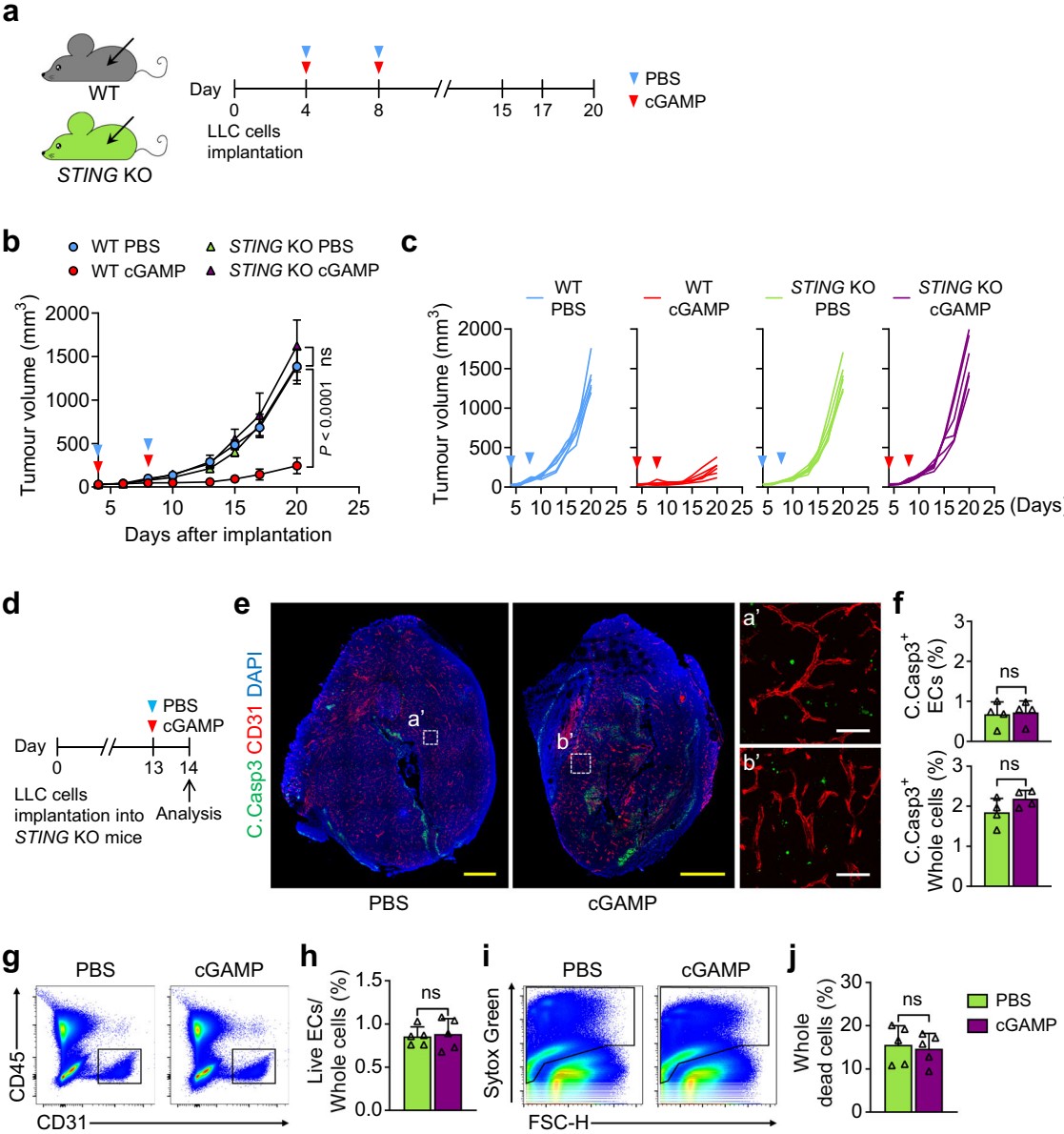

**Fig. 2 cGAMP-induced tumour EC apoptosis is abrogated in STING KO mice. a–c** Diagram depicting generation of implanted LLC tumour and treatment schedule of i.t. PBS or cGAMP in WT and *STING* KO mice. Comparison of LLC tumour growths. n = 6 mice/group from three independent experiments. Dots and bars indicate mean ± SD. Plot indicates each individual tumour growth. *P* values by Welch's one-way ANOVA test followed by Dunnett's T3 test. ns, not significant. **d–j** Diagram depicting generation of implanted LLC tumour in *STING* KO mice, i.t. PBS or cGAMP treatment, and sampling of tumours at 24 h later. Representative images and comparisons of apoptosis in tumours EC and whole tumour cells (whole cells). Each dotted-line box region is magnified and displayed in the right panels. Scale bars, 1.0 mm (yellow) and 100 µm (white). Each dot indicates a value from one mouse and n = 4 mice/group from two independent experiments (**f**). Representative flow cytometry plots and comparisons of populations of live ECs and dead cells in whole tumour cells. Each dot indicates a value from one mouse and n = 5 mice from two independent experiments (**h**, **j**). Vertical bars indicate mean ± SD. *P* values by two-tailed t-test. ns, not significant. Source data are provided as a Source Data file.

**Tip-like and proliferative tumour ECs are vulnerable to STING activation-induced apoptosis**. To further understand STING agonist-induced vascular destruction in the tumours, we performed single-cell RNA sequencing (scRNA-seq) on ECs obtained from LLC tumours (Fig. 4a, b). The cGAMP-treated tumour ECs exhibited differentially expressed genes (DEGs) related to leucocyte recruitment and the interferon stimulation pathway, such as *ACKR1*, *CCL5*, *CXCL9*, *CXCL10*, *IRF7*, *IRGM1*, *ISG15*, *OASL2*, *PHF11D*, and *SERPING1* (Fig. 4c). To identify the transcriptomic changes after STING activation, the scRNA-seq datasets from these two groups were integrated and analysed after batch effect removal. Through unsupervised

clustering, we identified four subpopulations: stalk-like, tip-like, proliferative, and arterial EC (Fig. 4d, e). Notably, the scRNA-seq analysis suggested that, compared to PBS-treated ECs, the cGAMP-treated ECs exhibited an increase of stalk-like EC population but reductions of tip-like and proliferative EC populations (Fig. 4f). Accordingly, compared with PBS-treated ECs, the cGAMP-treated ECs had a 5.2-fold higher population of selectin P[+] (SELP[+], a representative marker for stalk-like ECs) ECs, while they had a 41% less population of placental growth factor[+] (PGF[+], a representative marker for tip-like ECs) ECs (Fig. 4g–j). To identify cGAMP treatment-mediated changes in the signalling pathways and biological processes,

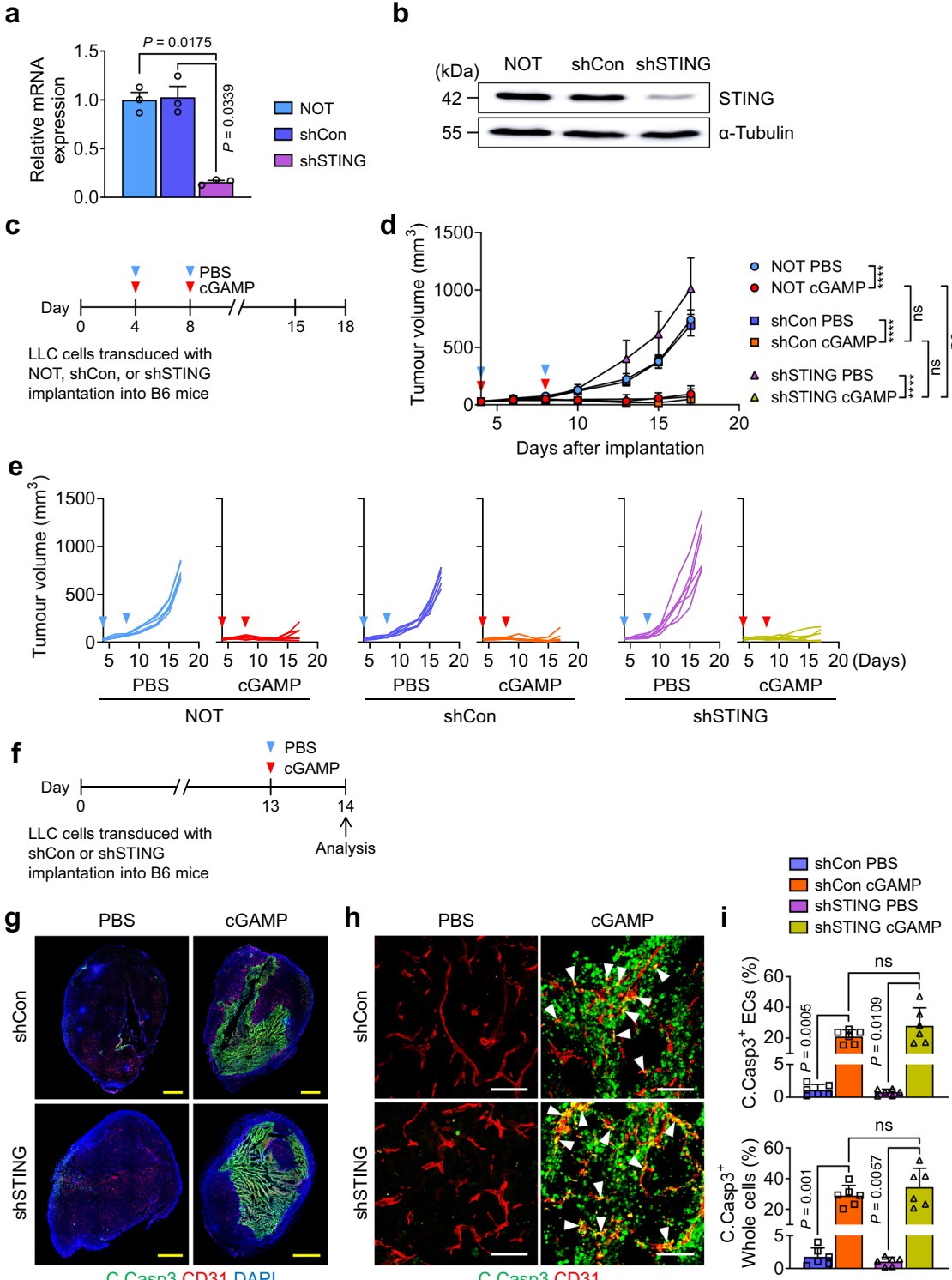

**Fig. 3 STING pathway of tumour cells is not required for cGAMP-induced tumour EC apoptosis. a**, **b** mRNA and protein levels of STING in cultured LLC cells transduced with nothing (NOT), shControl (shCon) or shSTING. Each dot indicates a value from one sample and n = 3 from two independent experiments. Vertical bars indicate mean ± SD. **c**–**e** Diagram depicting generation of implanted LLC tumour by injection of the LLC cells transduced with NOT, shCon or shSTING, and treatment schedule of i.t. PBS or cGAMP in B6 mice. Comparison of LLC tumour growths. n = 6 mice/group from three independent experiments. Dots and bars indicate mean ± SD. Plot indicates each individual tumour growth. **f**–**i** Diagram depicting generation of implanted LLC tumour by injection of the LLC cells transduced with shCon or shSTING, i.t. PBS or cGAMP treatment, and sampling of tumours at 24 h later. Representative images and comparisons of apoptosis in tumour ECs and whole tumour cells (whole cells). White arrowheads indicate apoptotic ECs. Scale bars, 1.0 mm (yellow) and 100 μm (white). Each dot indicates a value from one mouse and n = 6 mice from four independent experiments. Vertical bars indicate mean ± SD. P values by Welch's one-way ANOVA test followed by Dunnett's T3 test (**a**, **d**, **i**). ****P < 0.0001; ns, not significant. Source data are provided as a Source Data file.

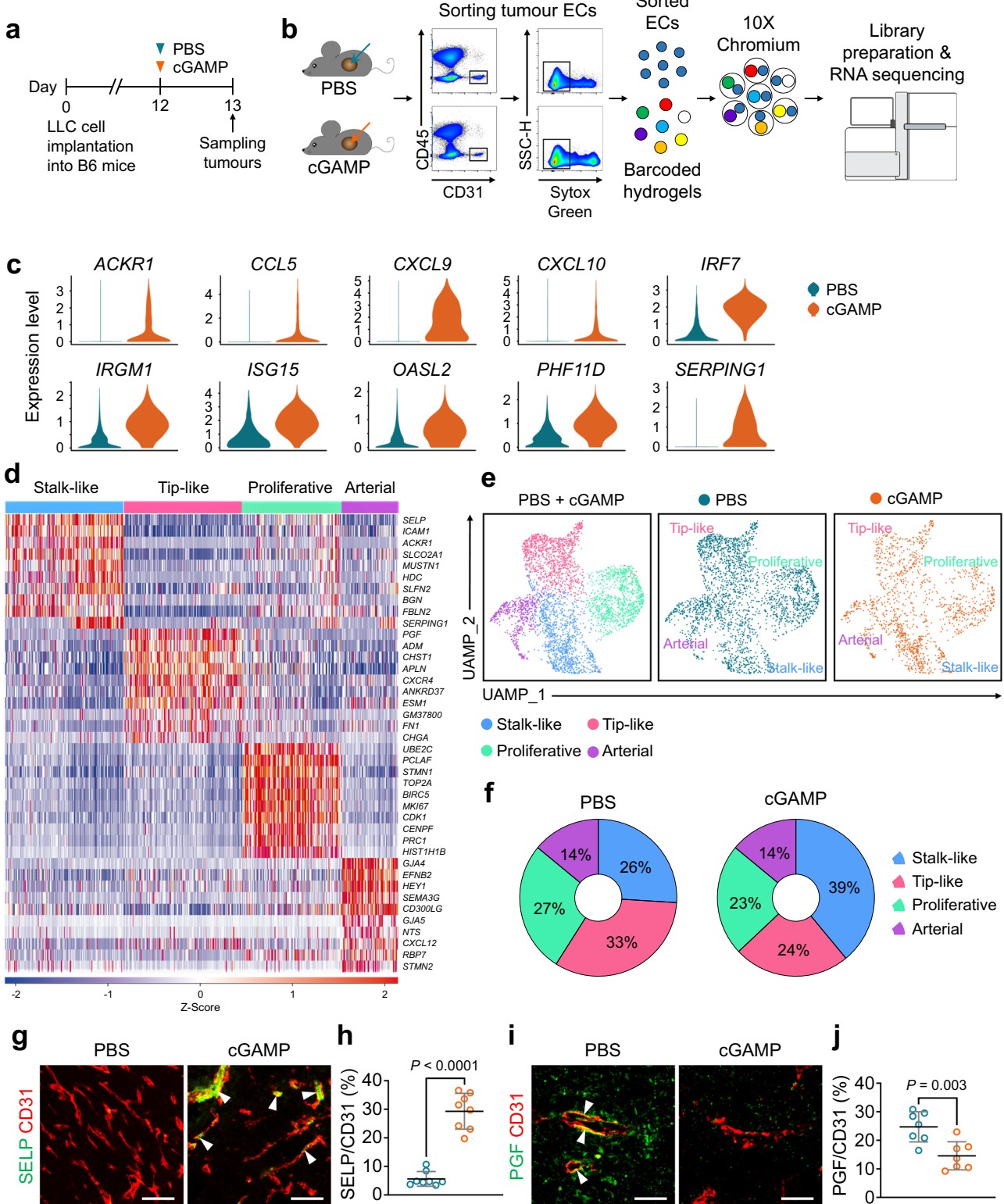

we generated triplicates of pseudo-bulk RNA expression for each scRNA-seq dataset, and performed gene set enrichment analysis (GSEA) (Supplementary Fig. 3a). In cGAMP-treated ECs, the apoptosis pathway was among the top up-regulated KEGG gene sets, and the TNFα signalling pathway was among the top up-regulated hallmark gene sets (Supplementary Fig. 3b–g).

**TNFα-TNFR1 mediates STING activation-induced tumour EC apoptosis.** Based on the GSEA outcomes, we hypothesised that the tumour EC apoptosis might be caused by TNFα produced due to STING activation, since TNFα exerts a potent anti-tumour effect derived from tumour vascular destruction[37–39]. In fact, compared to PBS-treated tumours, cGAMP-treated tumours showed 3.9-fold and 2.2-fold increases of TNFα and IFNγ,

**Fig. 4 Tumour ECs of implanted LLC tumours are vulnerable to STING activation-induced apoptosis. a, b** Diagrams depicting generation of LLC tumour, i.t. PBS or cGAMP treatment, tumour sampling at 24 h after the treatment, isolation of tumour ECs by FACS, and scRNA-seq analysis using a droplet-based platform. **c** Violin plots depicting the normalized expression levels of top-ranked differentially expressed genes between PBS- and cGAMP- treated tumour ECs. **d** Heatmap visualizing distinctive expression profiles of tumour EC subpopulations. Scaled expression levels of top ten differentially expressed genes for the indicated each cluster are shown. **e** UMAP plots comparing four clusters of tumour ECs derived from PBS- or cGAMP-treated tumours by unsupervised clustering of integrated dataset. Each dot represents a single EC. **f** Donut plot showing composition and difference in tumour EC subpopulations between PBS- and cGAMP- treated tumour ECs. **g–j** Representative images and comparisons of SELP[+]/CD31[+] stalk-like ECs and PGF [+]/CD31[+] tip-like ECs (white arrowheads) between PBS- and cGAMP-treated tumour ECs. Scale bars, 50 μm. Each dot indicates a value from one mouse and $n = 8$ mice/group for (**g, h**) and 7 mice/group for (**i, j**) from four independent experiments. Horizontal bars indicate mean ± SD. $P$ values by two-tailed t-test. Source data are provided as a Source Data file.

respectively (Supplementary Fig. 4a, b). Intratumoural TNFα injection induced 10% tumour EC apoptosis, and co-treatment with TNFα and IFNγ further increased tumour EC apoptosis to 22% (Supplementary Fig. 4c–e). While i.t. injection of TNFα or IFNγ alone did not induce significant apoptosis in LLC tumours, co-treatment with TNFα and IFNγ induced whole tumour cell apoptosis to 24% (Supplementary Fig. 4c-e). Similar to those in WT mice, i.t. TNFα injection-induced tumour EC apoptosis by 15.9%, and co-treatment with TNFα and IFNγ further increased tumour EC apoptosis to 43.4% in *STING* KO mice (Supplementary Fig. 4c–e), While i.t. IFNγ alone did not induce significant apoptosis in LLC tumour, i.t. TNFα induced whole tumour cell apoptosis by 11.9% in *STING* KO mice. However, co-treatment with TNFα and IFNγ induced whole tumour cell apoptosis by 37.6% in *STING* KO mice (Supplementary Fig. 4c–e). To further examine the role of TNFα in cGAMP-induced anti-tumour effects, we utilized anti-TNFα neutralizing antibody (Ab) (Supplementary Fig. 4f). Anti-TNFα neutralizing Ab diminished cGAMP-induced anti-tumour growth by 47%, and reduced apoptosis of tumour ECs and whole tumour cells by 55% and 48%, respectively (Supplementary Fig. 4f–k). In the *TNFR1* KO mice, cGAMP-induced anti-tumour growth was reduced by 64%, and apoptosis of tumour ECs and whole tumour cells was reduced by 65% and 66%, respectively (Supplementary Fig. 5). These findings imply that TNFα-TNFR1 activation in tumour bystander cells including tumour-associated macrophages, rather than in tumour cells and ECs, substantially contributed to the cGAMP-induced extensive tumour EC apoptosis and anti-tumour growth.

**TAMCs-generated TNFα mediates STING activation-induced tumour EC apoptosis.** To confirm whether tumour-associated myeloid cells (TAMCs) including tumour-associated macrophages were the main sources of TNFα[21,40], we generated myeloid cell-specific STING-depleted mice (*STING*[ΔMC]) by crossing *STING*[fl/fl] mice with LysM-Cre mice[41] (Fig. 5a, b). In each experiment, LysM-Cre-negative but flox/flox-positive littermates were defined as WT mice. TNFα in the cGAMP-treated LLC tumours was reduced by 81% in *STING*[ΔMC] mice compared with WT mice (Fig. 5c, d). In contrast, IFNβ and IFNγ in the cGAMP-treated LLC tumours was similar between *STING*[ΔMC] mice and WT mice (Fig. 5c, d). These findings suggested that TAMCs are main source of TNFα, while they are not a main source of IFNβ and IFNγ upon STING activation[42]. Of note, *STING*[ΔMC] mice exhibited reduced apoptosis of tumour ECs and tumour cells by 74% and 56%, and delayed anti-tumour growth by 41%, respectively (Fig. 5e–i). Number of tumour-infiltrating CD8[+] T cells (TICD8TC) were not significantly increased in the LLC tumour in *STING*[ΔMC] mice following cGAMP treatment, but were increased by 8-fold in WT mice (Fig. 5j, k). These findings indicate that STING agonist-induced activation of TAMCs are not only major sources of TNFα that rapidly induces tumour EC

apoptosis, but also responsible sources for recruiting TICD8TC to tumour microenvironment.

**cGAMP fails to induce tumour vascular destruction in *MMTV-PyMT* spontaneous breast cancer.** To evaluate whether cGAMP-induced tumour EC apoptosis might be clinically useful, we used a spontaneous breast cancer model, the *MMTV-PyMT* mice. Unexpectedly, i.t. cGAMP treatment did not induce apoptosis of tumour ECs or tumour cells, although it slightly delayed tumour growth (Fig. 6a–f). To investigate this difference, we generated an implanted breast tumour model through the inoculation of tumour cells that were harvested from the *MMTV-PyMT* spontaneous breast cancer, into the mammary fat pads (Fig. 6g). In this implanted breast tumour, i.t. cGAMP injection induced apoptosis of tumour ECs (37%) and whole tumour cells (47%) (Fig. 6h–j). Moreover, i.t. injections of cGAMP suppressed tumour growth by 91% (Fig. 6k–m). Thus, cGAMP-induced anti-tumour growth was remarkable in implanted breast tumour, but negligible in *MMTV-PyMT* spontaneous breast cancer (Fig. 6n). On the other hand, those two models showed comparable cGAMP-induced increases of tumoural TNFα and IFNγ (Fig. 6o–q). To ensure these findings, we generated another implanted breast tumour model through the inoculation of 4T1 breast carcinoma cells into the mammary fat pad (Supplementary Fig. 6a). Similar to those in implanted *MMTV-PyMT* cells breast tumour, i.t. cGAMP injection-induced apoptosis of tumour ECs (48%) and whole tumour cells (38%) and suppression of tumour growth by 81% in implanted 4T1 cells breast tumour (Supplementary Fig. 6b-f). These findings suggested that the apoptotic responses of tumour ECs and whole tumour cells against cGAMP-induced TNFα or IFNγ might be truncated in the spontaneous tumour.

**Tumour ECs of implanted breast tumours express transcriptome profiles sensitive to STING activation-induced apoptosis.** To investigate why cGAMP did not induce apoptosis in the spontaneous breast tumour, we performed bulk RNA sequencing (RNA-seq) of the ECs from normal mammary fat pads, *MMTV-PyMT* spontaneous breast tumours, and implanted breast tumours treated with PBS or cGAMP. All these three groups apparently responded to cGAMP with up-regulation of a transcriptome related to immune activation and stimulation of type I interferon pathways (Fig. 7a–d). Following cGAMP treatment, apoptosis-related genes were enriched in the ECs of both *MMTV-PyMT* and implanted breast tumours. However, only the ECs of cGAMP-treated implanted breast tumours exhibited high expression of pro-apoptotic genes, including *CASP3*, *CASP8*, *APF1*, *BID*, *BAX*, and *BAK* (Supplementary Fig. 7a, b). In all three groups, cGAMP treatment led to the enrichment of genes related to TNFα signalling via NF-κB (Supplementary Fig. 7c). Importantly, genes related to the PI3K-AKT-mTOR pathway (the strongest EC survival pathway against EC apoptosis) were

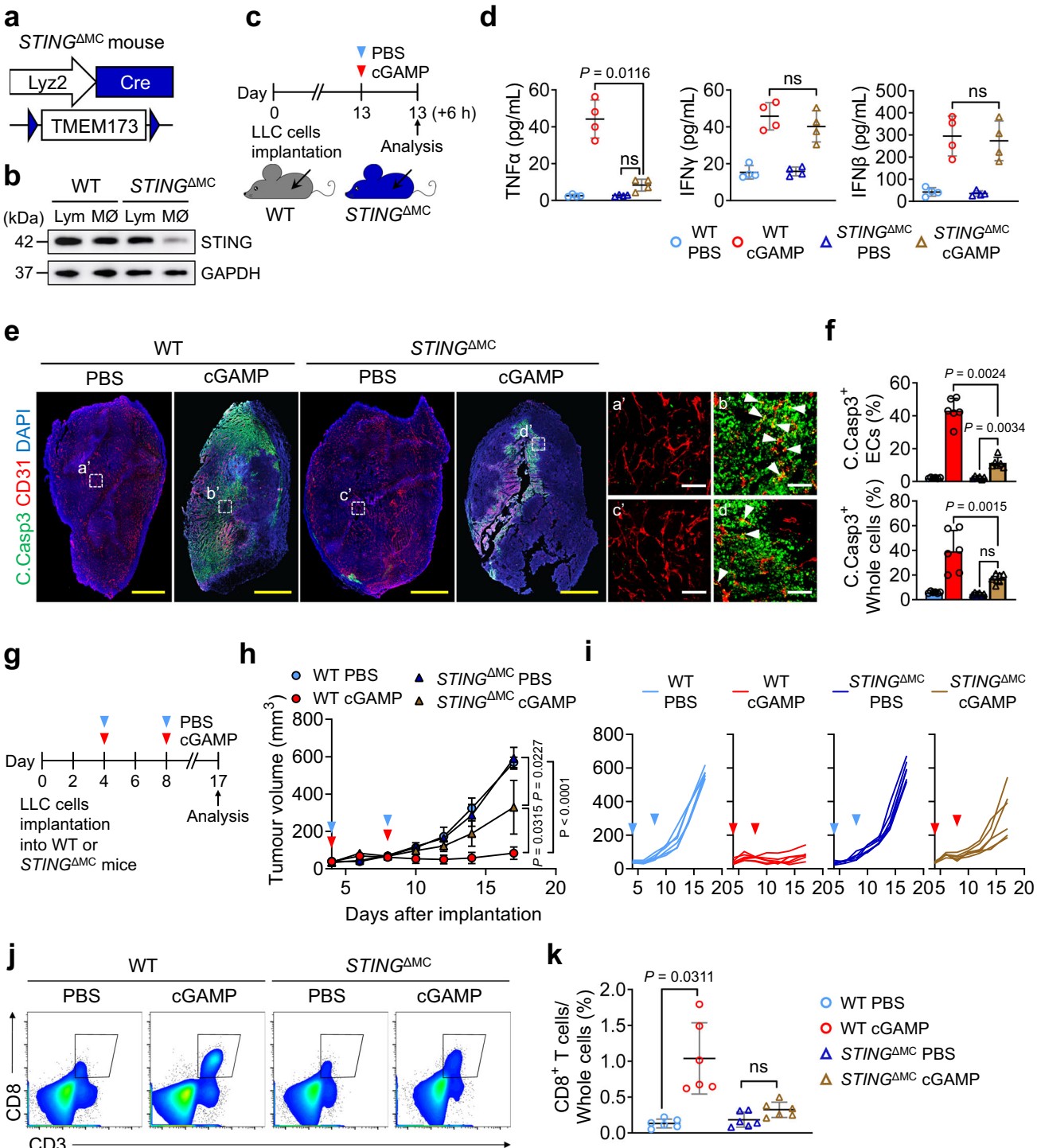

enriched following cGAMP treatment in *MMTV-PyMT* and implanted breast tumours but not in normal mammary fat pads (Supplementary Fig. 7d), which is consistent with a previous report[43]. Gene sets related to tip-like ECs and angiogenesis were highly expressed only in the ECs of implanted breast tumours, while gene sets related to proliferative ECs were highly expressed in the ECs of both *MMTV-PyMT* and implanted breast tumours (Fig. 7e, f, i). On the other hand, gene sets related to stalk-like and arterial ECs were enriched in the ECs of normal mammary fat pads (Fig. 7g, h). These findings imply that tip-like and proliferative ECs could be vulnerable to STING activation-induced apoptosis in tumours.

**Tip-like and proliferative ECs are vulnerable to STING activation-induced apoptosis, but less sensitive to spontaneous tumour *versus* to implanted tumour.** To gain a further insight on the response of tumour ECs to STING agonist, we performed scRNA-seq on the ECs of *MMTV-PyMT* spontaneous tumour and its orthotopic implanted breast tumour, which were sampled at 3 h after i.t PBS or cGAMP (Fig. 8a). Compared with those treated with PBS, cGAMP highly increased expressions of the genes related to immune activation and stimulation of type I interferon pathway such as *CCL5, CXCL9, CXCL10, IRF7, IRGM1, ISG15, OASL2* and *PHF11D* in both tumour ECs (Fig. 8b, c). Five distinct EC clusters- stalk-like, tip-like, proliferative,

**Fig. 5 TAMCs-derived TNFα is a crucial mediator for STING agonist-induced apoptosis of Tumour ECs. a, b** Diagram depicting generation of myeloid cell-specific STING-deleted (STING$^{\Delta MC}$) mice. Confirmation of STING depletion in macrophages (MØ) but not in lymphocytes (Lym) of STING$^{\Delta MC}$ mice by immunoblotting from two independent experiments. **c, d,** Diagram depicting generation of implanted LLC tumour in WT and STING$^{\Delta MC}$ mice, i.t. PBS or cGAMP treatment, and sampling of tumours at 6 h later. Comparisons of TNFα, IFNγ and IFNβ protein levels in tumour lysates treated with PBS or cGAMP between WT and STING$^{\Delta MC}$ mice. Each dot indicates a value from one mouse and n = 4 mice/group from four independent experiments. Horizontal bars indicate mean ± SD. **e, f** LLC tumours were sampled 24 h after i.t. PBS or cGAMP injection in WT and STING $^{\Delta MC}$ mice. Representative images and comparisons of apoptosis in tumour ECs (white arrowheads) and whole tumour cells (whole cells). Scale bars, 1.0 mm (yellow bars) and 100 μm (white bars). Each dot indicates a value from one mouse and n = 6 mice/group from four independent experiments. Vertical bars indicate mean ± SD. **g–k** Diagram depicting schedule of LLC cells implantation, treatment, and sampling in WT and STING$^{\Delta MC}$ mice. Comparisons of tumour growth. n = 6 mice/group from four independent experiments. Dots and bars indicate mean ± SD. Plot indicates each individual tumour growth. Representative flow cytometry plots and comparisons showing tumour infiltrating CD8$^+$ T cell populations in whole tumour cells. n = 6 mice/group from four independent experiments. Horizontal bars indicate mean ± SD. P values by Welch's one-way ANOVA test followed by Dunnett's T3 test (**d, f, h, k**). ns, not significant. Source data are provided as a Source Data file.

arterial and AQP7$^+$ ECs were present in the pooled ECs of both spontaneous and implanted tumour treated with PBS or cGAMP (Fig. 8d, e and Supplementary Fig. 8a-c). AQP7$^+$ ECs are a breast-specific, fully-differentiated subpopulation that highly expresses AQP7, CD35 and FABP5 mRNAs and actively participates in the glycerol and fatty acid metabolism and transport[44]. Distinctiveness of five clustering was largely blunted in the pooled ECs treated with cGAMP compared with those treated with PBS (Fig. 8d, e), implying that tumour ECs were largely affected by i.t. STING agonist. The scRNA-seq analysis suggested that cGAMP treatment markedly reduced the population of proliferative ECs in spontaneous tumour, while it markedly reduced both proliferative and tip-like ECs in implanted tumour (Fig. 8f, g). The latter findings are similar to those in implanted LLC tumours (Fig. 4f–j). Gene ontology enrichment analysis on the ECs of implanted tumour versus spontaneous tumour revealed differential transcriptional responses to both PBS and cGAMP treatment (Supplementary Fig. 8d). The genes related to cellular responses to cytokine stimulus and interferon-γ were enriched in all EC clusters of the implanted tumours treated with PBS (Supplementary Fig. 8d), implying that implanted tumour ECs could be more responding to anti-tumour immunotherapy compared with spontaneous tumour ECs. Of note, the genes related to apoptosis were enriched in the tip-like ECs of the implanted tumours treated with cGAMP (red underline in Supplementary Fig. 8d), implying that the tip-like ECs are vulnerable to STING agonist. Because there was a lack of proliferative ECs mainly due to apoptotic death in the cGAMP-treated tumours (Supplementary Fig. 8d), we could not analyse the character of the proliferative ECs. These findings denote that tip-like and proliferative ECs are vulnerable to STING activation-induced apoptosis, and these can partly explain why only the ECs of implanted, but not spontaneous, breast cancer underwent apoptosis following cGAMP treatment. Moreover, the ECs retaining maturation gene profiles within tumours such as arterial, stalk-like and AQP7$^+$ ECs (Fig. 8d, e and Supplementary Fig. 8a, b) are likely to be resistant to STING activation-induced apoptosis.

**Combined treatment with cGAMP and AKT 1/2 inhibitor induces apoptosis of tumour ECs.** Compared to the implantation tumour model, the spontaneous tumour model more closely resembles primary human solid tumours[45]; thus, STING agonist monotherapy could not produce satisfactory outcomes in clinical use[46,47]. Based on the bulk RNA sequencing data (Supplementary Fig. 7d) and a previous report[43], we hypothesized that inhibition of AKT signalling could potentiate spontaneous tumour to combined TNFα and IFNγ-induced tumour EC apoptosis. In fact, immunohistochemical analysis on the tissue array of human breast ductal adenocarcinomas revealed that tumour ECs in tumour core region had highly activated AKT compared with the ECs in adjacent normal tissue region (Supplementary, Fig. 9).

We chose AKT 1/2 inhibitor as AKT inhibitor (AKTi) and confirmed its effects by examining phosphorylation statuses of AKT at S473 and its downstream FOXO1 at T24 in cultured HUVECs and ECs of MMTV-PyMT spontaneous breast tumour. Consistent with previous studies[48,49], AKTi (10 μM) completely abolished TNFα-induced phosphorylation of AKT and FOXO1 in HUVECs (Fig. 9a), while i.p. AKTi (50 mg/kg of body weight) largely induced nuclear localization of FOXO1 in the tumour ECs (Fig. 9b). These findings indicate that AKTi effectively inhibits its pathways in vitro and in vivo. We found that addition of TNFα, IFNγ, or TNFα + IFNγ did not significantly alter cell apoptosis or viabilities in cultured HUVECs, LLC cells, or tumour cells derived from MMTV-PyMT breast tumours (Fig. 9c–g).

Further addition of AKTi led to reduction of cell viability by 89% and 93% and induction of apoptosis by 4% and 20%, respectively in HUVECs cultured with TNFα or TNFα + IFNγ (Fig. 9c–e). In cultured LLC cells and MMTV-PyMT tumour cells, AKTi addition led to a reduction of cell viability by 37% and 35%, respectively, regardless of the inclusion of TNFα and/or IFNγ (Fig. 9f, g).

Accordingly, in spontaneous breast cancer of MMTV-PyMT mice, combined treatment with i.t. cGAMP and i.p. AKTi induced apoptosis of up to 28% and 34% in tumour ECs and whole tumour cells, while either cGAMP or AKTi alone did not induce notable apoptosis (Fig. 10a–d). Tumour antigens that originate from extensive tumour cell apoptosis provoke dendritic cell (DC) maturation, promoting antigen presentation to establish anti-tumour immunity[6,50]. Therefore, we next evaluated the DCs in tumour drainage lymph nodes (TDLNs) at 24 h after treatments to determine whether the increased tumour cell apoptosis induced by co-treatment with cGAMP and AKTi contributed to DC maturation (Fig. 10a). Even in the absence of tumour cell apoptosis, cGAMP treatment-induced DC maturation, manifested by 4-, 6-, and 2-fold increases of CD80, CD86, and MHC-II, respectively, compared to PBS treatment (Fig. 10e-g). Moreover, co-treatment with cGAMP and AKTi further enhanced DC maturation, as demonstrated by the 2-, 1.7-, and 3-fold increases of CD80, CD86, and MHC-II, respectively, compared to cGAMP monotherapy (Fig. 10e-g).

Combined treatments with i.t. cGAMP and i.p. AKTi led to the suppression of tumour growth by 76% in MMTV-PyMT breast tumour mice. In contrast, treatment of these mice with i.t. cGAMP or i.p. AKTi alone suppressed tumour growth by 35% or 0%, respectively (Fig. 10h–j). Moreover, co-treatment increased the population of TICD8TC by 75-fold, while treatment with i.t. cGAMP alone non-significantly increased TICD8TC by 14-fold and i.p. AKTi alone yielded no change (Fig. 10k, l). Thus, our results demonstrated that AKTi potentiated the cGAMP-induced

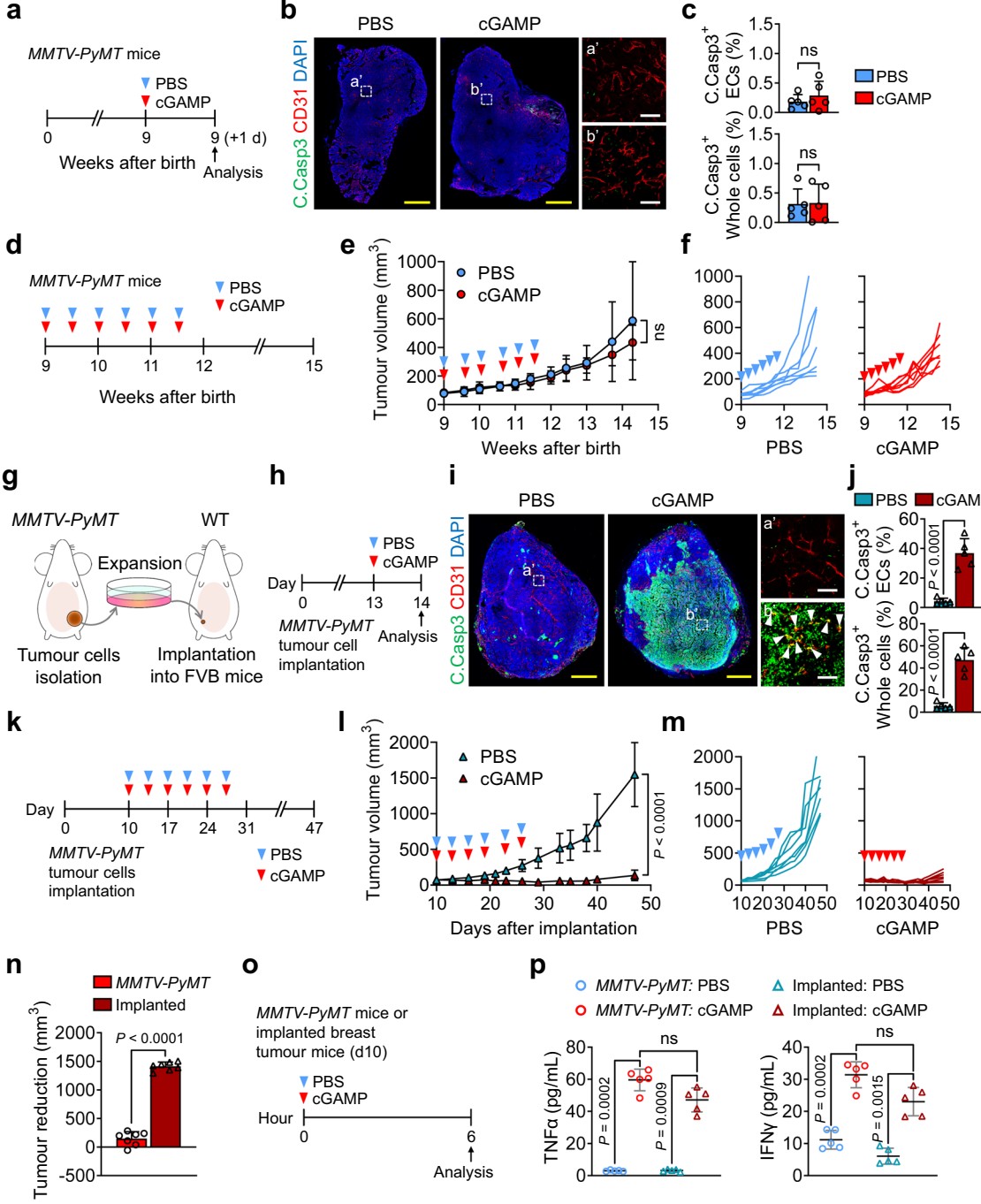

**Fig. 6 cGAMP fails to induce tumour EC apoptosis and anti-tumour growth in *MMTV-PyMT* spontaneous breast cancer. a–c** Diagram depicting treatment and sampling in 9 weeks-old *MMTV-PyMT* mice. Representative images and comparisons of apoptosis in tumour ECs and whole tumour cells (whole cells). Scale bars, 1.0 mm (yellow bars) and 100 μm (white bars). *n* = 5 mice/group from two independent experiments. Vertical bars indicate mean ± SD. **d–f** Diagram depicting treatment and sampling in 9 weeks-old *MMTV-PyMT* mice. Comparison of tumour growths. *n* = 7 mice/group from two independent experiments. Plots and bars indicate mean ± SD. Plot indicates each individual tumour growth. **g** Diagram depicting generation of an orthotopic implanted breast tumour in syngeneic FVB mice. **h-j** Diagram depicting treatment and sampling in implanted breast tumour mice. Representative images and comparisons of apoptosis in tumour ECs (white arrowheads) and whole tumour cells. Scale bars, 1.0 mm (yellow bars) and 100 μm (white bars). *n* = 5 mice/group from two independent experiments. Vertical bars indicate mean ± SD. **k–m** Diagram depicting treatment and tumour growth in implanted breast tumour mice. Comparisons of tumour growths. *n* = 7 mice/group from two independent experiments. Plot and bars indicate mean ± SD. Plot indicates each individual tumour growth. **n** Comparison of reduction of tumour volume by cGAMP treatment between spontaneous and implanted breast tumours at day 47 after the implantation. **o, p** Diagram depicting treatment and sampling 6 h later in 9-week-old *MMTV-PyMT* mice and its implanted breast tumour mice 10 day (d10) after implantation. Comparisons of TNFα and IFNγ levels in tumour lysates. *n* = 5 mice/group from four independent experiments. Horizontal bars indicate mean ± SD. *P* values by two-tailed t-test (**c, e, j, l, n**) or Welch's one-way ANOVA test followed by Dunnett's T3 test (**p**). ns, not significant. Source data are provided as a Source Data file.

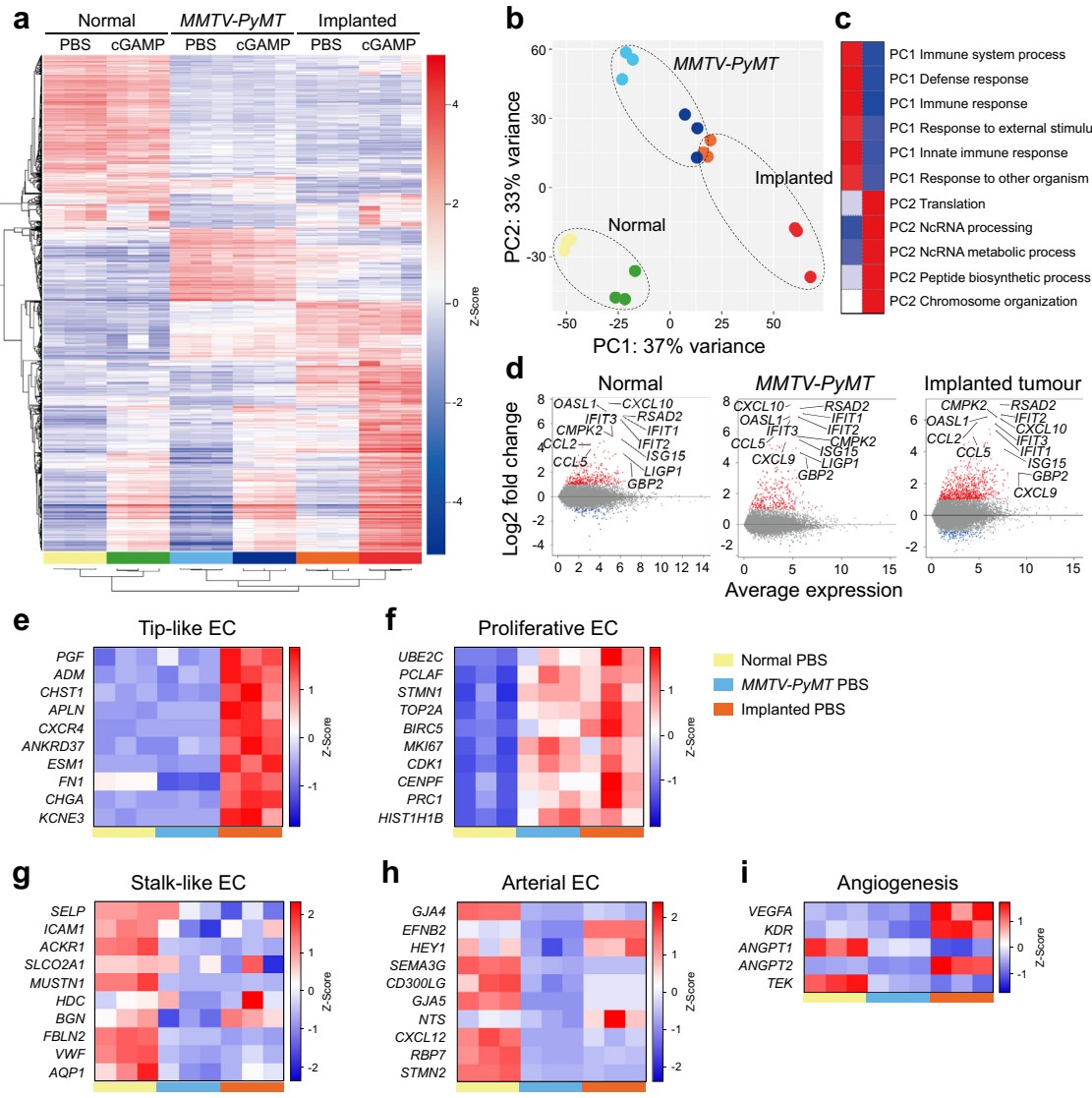

**Fig. 7 Implanted breast tumour ECs exhibit angiogenic and inflammatory transcriptome profiles. a** Heatmap of the RNA-seq data of ECs from normal mammary pads, *MMTV-PyMT* spontaneous breast tumours, and implanted breast tumours (see Fig. 6). *n* = 3 mice/group from three independent experiments. **b** Principal component analysis (PCA) of the RNA-seq data. **c** Gene ontology analysis of PCA axis. **d** MA plots showing differentially expressed genes by cGAMP treatment in each group. **e–i** Heatmaps comparing genes related to tip-like ECs, proliferative ECs, stalk-like ECs, arterial ECs, and angiogenesis in the ECs of normal mammary pads and spontaneous and implanted tumours treated with PBS.

anti-tumour effect by inducing extensive apoptosis of tumour ECs, enhancing immunogenic tumour cell apoptosis, and increasing recruitment of TICD8TC in spontaneous breast cancer.

**AKTi potentiates cGAMP-induced anti-tumour effect in spontaneous melanoma and LLC tumour.** We next wondered whether co-treatment with cGAMP and AKTi would also be highly effective for suppressing tumour growth in another spontaneous tumour, we adopted *Tyr-CreER*[T2];*Braf*[CA];*Pten*[loxP] spontaneous melanoma mice[51] (Supplementary Fig. 10a). Combined treatment with i.t. cGAMP and i.p. AKTi induced apoptosis in tumour ECs and whole tumour cells by 47% and 53%, respectively, while either cGAMP or AKTi alone did not induce notable apoptosis in spontaneous melanoma (Supplementary Fig. 10a–d). The combined treatment led to the suppression of tumour growth by 91% in *Tyr-CreER*[T2];*Braf*[CA];*Pten*[loxP] spontaneous melanoma mice. In contrast, treatment of these mice with i.t. cGAMP or i.p. AKTi alone suppressed tumour growth by 44% or 10%, respectively (Supplementary Fig. 10e–g). We further

examined whether co-treatment with cGAMP and AKTi would also be highly effective for suppressing tumour growth in implanted LLC tumour. Combined treatment with i.t. cGAMP and i.p. AKTi increased apoptosis of tumour ECs and whole tumour cells by 86% and 641%, respectively. However, although i.t. cGAMP alone increased apoptosis of tumour ECs and whole tumour cells by 70% and 568%, i.p. AKTi alone had negligible effects on them (Supplementary Fig. 11a–c). Accordingly, compared to treatment with PBS alone, co-treatments suppressed tumour growth by 81%, while treatments with cGAMP or AKTi alone suppressed tumour growth by 60% or 3%, respectively (Supplementary Fig. 11d–f). Moreover, co-treatment or cGAMP alone increased the TICD8TC population by 8-fold or 3-fold, while the TICD8TC population was not significantly changed in tumours treated with AKTi alone (Supplementary Fig. 11 g, h). Thus, we found that AKTi also potentiated the cGAMP-induced anti-tumour effect in implanted tumours, by further inducing extensive apoptosis of tumour ECs and increasing recruitment of TICD8TC into the tumour.

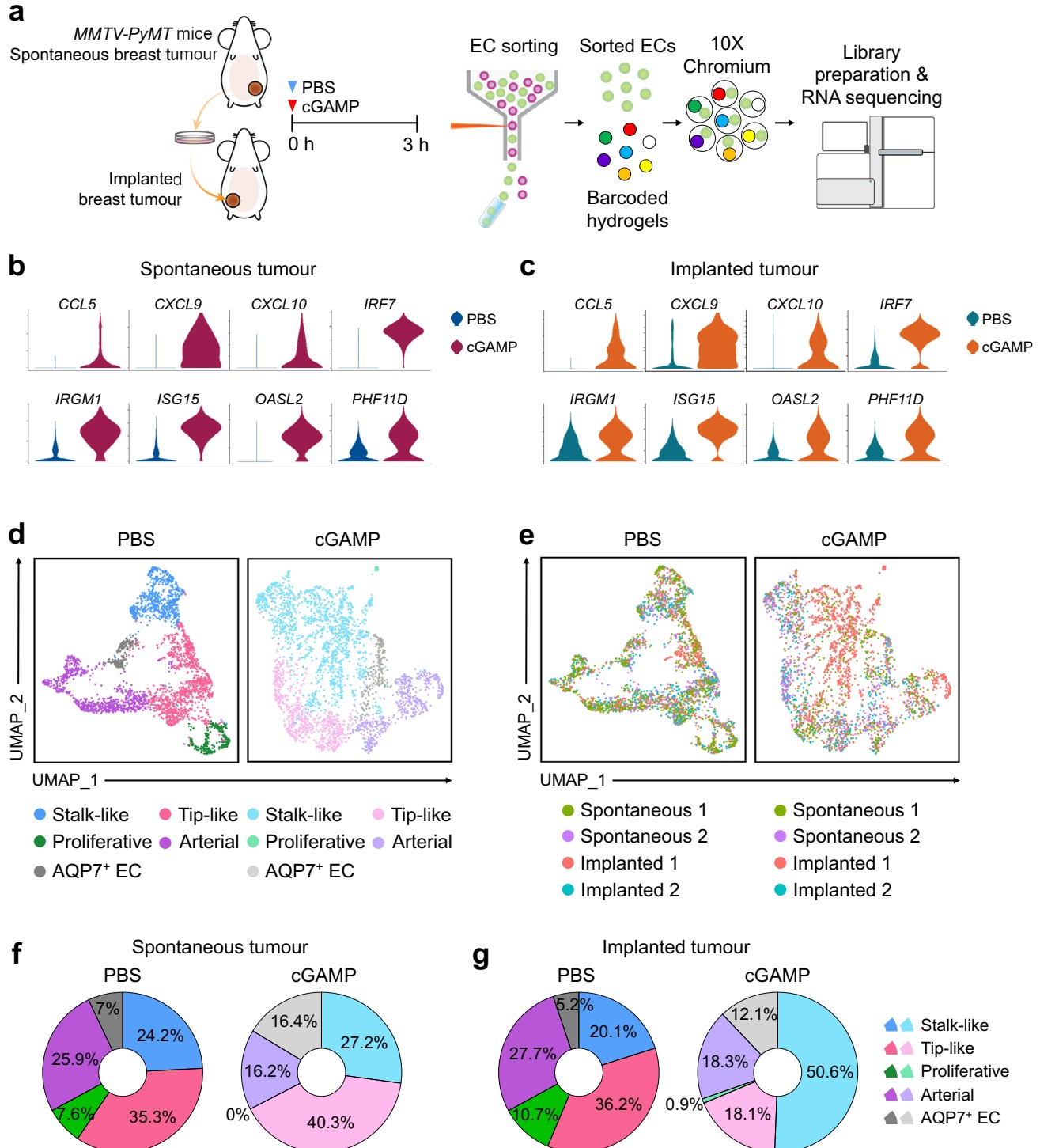

**Fig. 8 Implanted, but not spontaneous, breast tumour ECs are vulnerable to STING activation-induced apoptosis. a** Schematic diagram for depicting treatment, sampling, and scRNA-seq of tumour ECs in *MMTV-PyMT* spontaneous and their implanted breast tumours treated with i.t. PBS or cGAMP. *n* = 2 mice/each group. **b, c** Violin plots depicting the normalized expression levels of top-ranked differentially expressed genes in spontaneous and implanted tumour ECs. **d** UMAP plots comparing five clusters of tumour ECs by unsupervised clustering, integrating datasets in spontaneous and implanted tumour ECs between PBS and cGAMP treatment. **e** The origin of datasets in each cluster. Each dot represents single cell. **f, g** Compositional differences of tumour EC subpopulations by treatments in spontaneous and implanted tumour ECs.

## Discussion

In this study, we examined the poorly understood role of the STING agonist cGAMP as a tumour-specific VDA, and demonstrated that this early activity of cGAMP is a key for establishing the anti-tumour effect. We also showed that refractoriness of i.t.

STING therapy could be relieved by the addition of AKTi through effective tumour EC apoptosis in spontaneous tumours (Supplementary Fig. 12).

One representative VDA, flavone acetic acid, selectively block vascular flow in the tumour core by promoting tumour EC

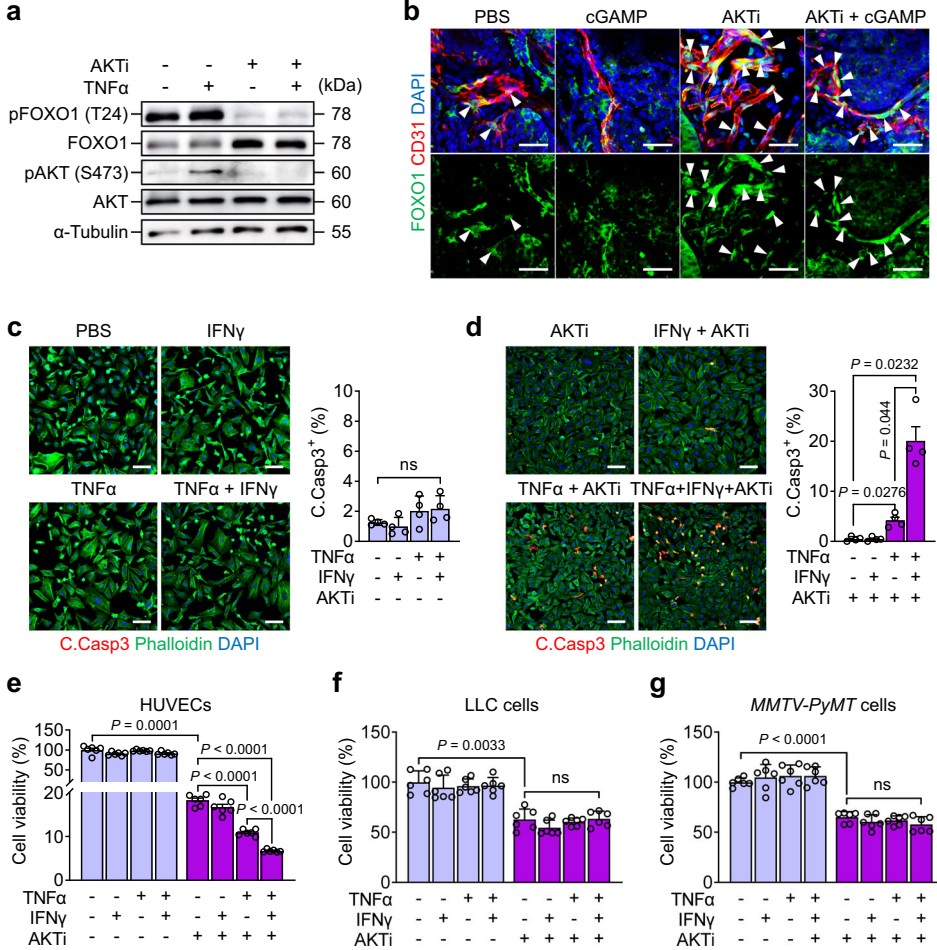

**Fig. 9 AKT inhibitor sensitizes cultured HUVECs to TNFα-induced apoptosis. a** Immunoblot analysis of FOXO1, phosphorylated FOXO1 at threonine 24 [pFOXO1 (T24)], AKT, phosphorylated AKT at serine 473 [pAKT (S473)] and α-tubulin in cultured HUVECs at 30 min after indicated treatments. Three independent experiments showed similar findings. **b** Representative images of FOXO1 subcellular localization in the ECs of LLC tumour. PBS (70 μl), cGAMP (14 μg/70 μl) or AKTi (50 mg/kg) was injected 3 h before sampling. White arrowheads indicate nuclear localization of FOXO1. Scale bars, 50 μm. Five independent experiments showed similar findings. **c, d** Representative images and comparison of apoptosis in cultured HUVECs treated with indicated agents. TNFα (200 ng/ml), IFNγ (300 ng/ml) or AKTi (10 μM) was added and incubated for 6 h. Scale bars, 100 μm. Dots indicate values from four independent experiments. Vertical bars indicate mean ± SD. *P* values by Welch's one-way ANOVA test followed by Dunnett's T3 test. ns, not significant. **e–g** Viabilities of cultured HUVECs, LLC cells, and MMTV-PyMT tumour cells by indicated treatments. TNFα (200 ng/ml), IFNγ (300 ng/ml) or AKTi (10 μM) was added and incubated for 24 h. Dots indicate values from six independent experiments. Vertical bars indicate mean ± SD. *P* values by Welch's one-way ANOVA test followed by Dunnett's T3 test. ns, not significant. Source data are provided as a Source Data file.

apoptosis, leading to powerful anti-tumour effect in murine tumour models[52]. Its derivatives have been developed from the trait of its chemical structure possessing amenable site with active region of xanthenone-4-acetic acid, which gave rise to the development of DMXAA as potent as 16-fold[53]. Interestingly, the scheme of DMXAA on anti-tumour effect has changed from VDA to immunotherapeutic agent as a STING agonist[54,55]. Currently, i.t. STING agonist therapy is generally recognized to increase the production of type I interferon in tumours, which promotes CD8[+] T cell priming and infiltration into the tumour, eventually converting immunologically "cold" tumours to "hot" tumours[9,14,20]. Therefore, STING therapy has been considered a promising treatment for use in combination with immune checkpoint inhibitors, since they are more effective in hot tumours than cold tumours[56]. However, throughout this study, we addressed the strong and rapid apoptotic role of STING agonist, cGAMP, in tumour ECs as a VDA. Through scRNA-seq and bulk RNA-seq analyses of the tumour ECs, we revealed that tip-like and proliferative EC subpopulations were vulnerable to STING activation-induced apoptosis. More importantly, we

found that TAMCs-derived TNFα was a key mediator of the extensive apoptosis of tumour ECs (Supplementary Fig. 12).

STING plays multifaceted roles in tumour progression depending on target cells such as tumour cell or cells composing tumour microenvironment such as ECs, cancer-associated fibroblasts, and immune cells[3,33–35]. For instance, Yang et al.[14] have recently reported that STING agonists normalize tumour vessels mediated by up-regulating the genes related to type I/II interferon and vascular stabilization and enhancing pericyte coverage. However, it occurs several days after i.t. injection with a prior and marked reduction of tumour ECs. In comparison, this study shows that STING agonist strongly and rapidly induces tumour EC apoptosis within a day after the injection. Moreover, our findings indicate that the massive tumour cell death, following extensive cGAMP-induced tumour EC apoptosis, led to sufficient TICD8TC expansion and suppressed tumour growth in implanted tumours. On the other hand, when tumour ECs were resistant to cGAMP-induced apoptosis, the sparse tumour cell death led to failures of TICD8T expansion and suppression of tumour growth in spontaneous tumours (Supplementary Fig. 12).

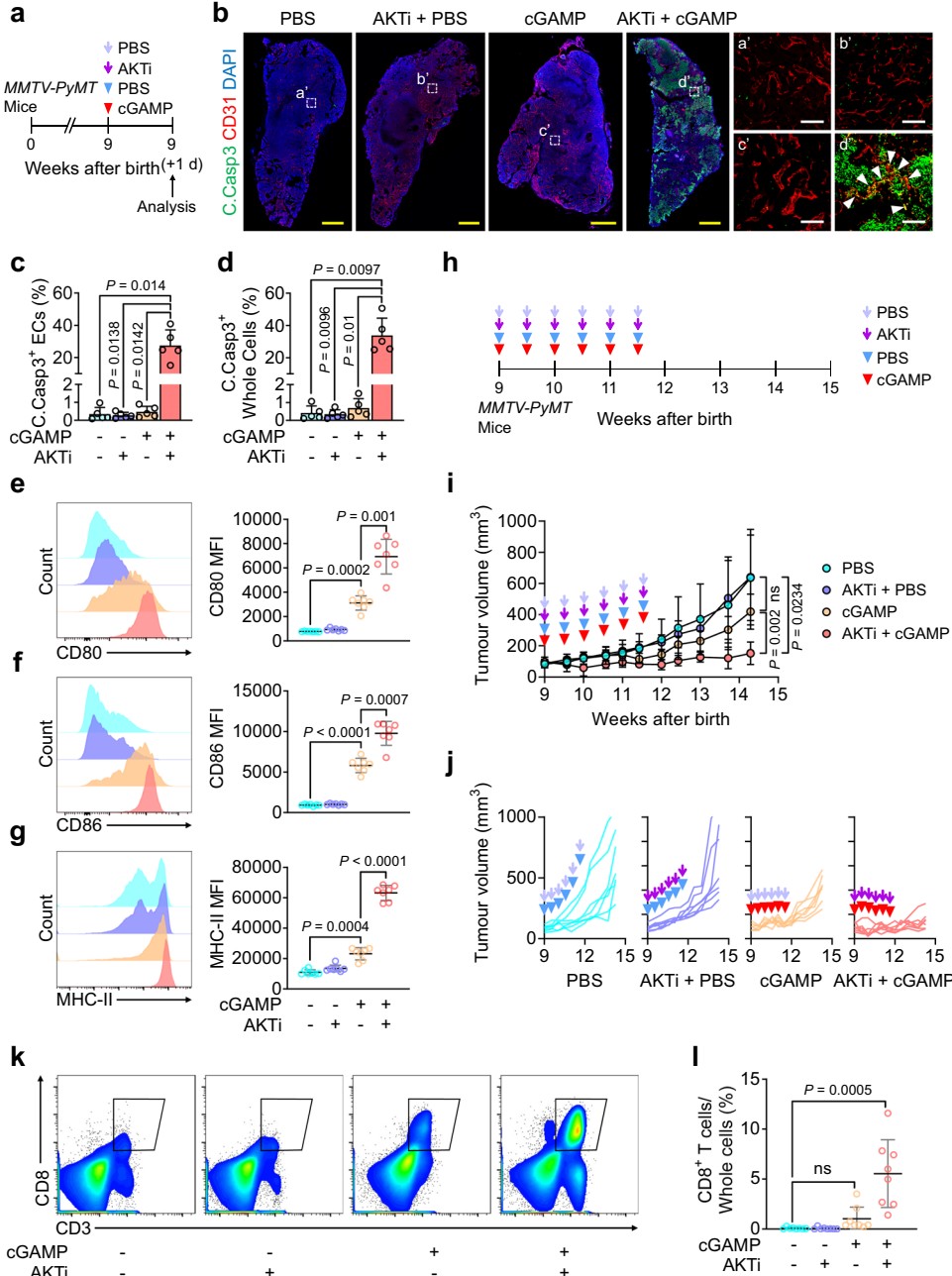

**Fig. 10 AKT inhibitor potentiates STING activation-induced tumour EC apoptosis and anti-tumour effect in MMTV-PyMT spontaneous breast cancer.** **a–g** Diagram depicting treatment and tumour sampling in *MMTV-PyMT* mice. b–d, Representative images of tumour vessels and apoptosis. Each dotted-line box region is magnified and displayed in right panels. White arrowheads indicated tumour EC apoptosis. Scale bars, 1.0;mm (yellow bars) and 100 μm (white bars). Comparisons of apoptosis of tumour ECs and whole tumour cells. Each dot indicates a value from one mouse and $n = 5$ mice/group from four independent experiments (**c**, **d**). Vertical bars indicate mean ± SD. **e–g** Representative flow cytometry plots and comparisons of CD80, CD86, and MHC-II expression on the DCs in each TDLN. Each dot indicates a value from one mouse and $n = 7$ mice/group from four independent experiments (**e–g**). Horizontal bars indicate mean ± SD. **h–j** Diagram depicting treatment schedule in *MMTV-PyMT* mice. Comparisons of tumour growths. $n = 7$ mice/group from four independent experiments. Vertical bars indicate mean ± SD. Plots indicate each individual tumour growth. **k, l** AKTi or cGAMP was injected twice with 3-day interval in 9-week-old *MMTV-PyMT* mice and tumours were sampled 1 week later. Representative flow cytometry plots and comparison of CD8$^+$ T cells. Each dot indicates a value from one mouse and $n = 8$ mice/group from four independent experiments. Horizontal bars indicate mean ± SD. *P* values by Welch's one-way ANOVA test followed by Dunnett's T3 test (**c–g**, **i**) or Kruskal–Wallis test followed by Dunn's test (**l**). ns, not significant. Source data are provided as a Source Data file.

AKT activation plays a central role in survival in several cell types, including ECs[26–28]. Our analysis actually revealed that the tumour ECs of human breast ductal adenocarcinomas retain highly phosphorylated AKT compared with the ECs in adjacent normal tissue region. Nevertheless, our findings indicate that

AKTi alone did not induce apoptosis in tumour ECs and whole tumour cells in spontaneous tumours. We hypothesized that AKT inhibition might enforce STING activation-induced apoptosis of tumour ECs in primary tumours based on our RNA-seq data and a previous report[43]. Indeed, AKTi treatment abrogated

unresponsiveness to STING therapy, mediated through effective apoptosis of tumour ECs. Moreover, AKTi potentiated the cGAMP-induced anti-tumour effect, which partly could be mediated by enhancing the innate immune responses of tumour cells and bystander cells to cGAMP (Supplementary Fig. 12). Supporting this scenario, previous studies[57] demonstrated that intracellular STING signalling and AKT pathways are antagonistically interacted for tumour progression mediated through innate immune responses, while AKTi increases T cell persistence and infiltration as well as early memory phenotype in tumour sites to enhance anti-tumour effect[58,59].

In conclusion, our present findings unravel long-standing questions regarding the mechanism underlying STING activation-induced tumour EC apoptosis. Our results highlight the critical role of STING agonist as a VDA, which results in extensive tumour cell death releasing massive tumour antigen, supporting the activation of anti-tumour immunity. Based on these findings, we propose that combination treatment of STING agonist and AKT inhibitor should be considered for clinical human tumours, which could be refractory to single therapy with either agent alone.

## Methods

**Mice**. The experiments on the mice were conducted according to the approval (KA2017-30) of the Animal Care Committee of Korea Advanced Institute of Science and Technology (KAIST). Specific pathogen-free C57BL/6 J, FVB/NJ, Balb/c, STING knockout (C57BL/6J-Sting1$^{gt}$/J), MMTV-PyMT transgenic [FVB/N-Tg (MMTV-PyVT)634Mul/J], LysM-Cre (B6.129P2-Lyz2$^{tm1(cre)Ifo}$/J), TNFR1 knockout (C57BL/6-Tnfrsf1a$^{tm1Imx}$/J), and B6.Cg-Tg(Tyr-cre/ERT2)13Bos Braf$^{tm1Mmcm}$ Pten$^{tm1Hwu}$/BosJ (Tyr-Cre-ER$^{T2}$;Braf$^{CA}$;Pten$^{loxP}$) mice were purchased from Jackson Laboratory (USA). VE-cadherin-Cre-ER$^{T2}$ mice[36] were provided by Prof. Yoshiyaki Kubota (Keio University, Japan), transferred, established and bred in SPF animal facilities at KAIST. STING1$^{tm1a(EUCOMM)Hmgu}$ mice, which have conditional knockout potential with frt-flanked lacZ and neomycin resistance cassette followed by loxP-flanked exon 6, were purchased from EUMMCR. To generate STING$^{flox}$ mice, STING$^{tm1a(EUCOMM)Hmgu}$ mice were crossed with FLP1 recombinase expressing mice (B6;SJL-Tg(ACTFLPe)9205Dym/J), which were purchased from Jackson laboratory. All mice were fed with ad libitum access to standard diet (PMI lab diet) and water, and were anesthetized by i.p. injection of a combination of anaesthetics (80 mg/kg of ketamine and 12 mg/kg of xylazine) before all the procedures and being sacrificed.

**Tumour models and treatment regimens**. Lewis lung carcinoma (LLC) cells, B16F10 melanoma cells and 4T1 breast carcinoma cells were purchased from American Type Culture Collection. To generate implanted LLC or B16F10 tumour model, $1 \times 10^6$ LLC cells, B16F10 cells, or LLC cells transduced with shControl or shSTING lentiviral particles were subcutaneously injected into the right flank of 8–9-week-old male C57BL/6 J mice. For the experiments with MMTV-PyMT spontaneous breast tumours, the treatment began when the female mice were 9-week old. To generate an orthotopic implanted breast tumour model, breast tumour nodules of MMTV-PyMT transgenic mice were harvested and digested with enzyme buffer containing 2 mg/ml collagenase type II (Worthington), 0.1 mg/ml DNase I (Roche), and 0.8 mg/ml dispase (Gibco) at 37 °C for 30 min. The dissociated cells underwent trypsinization with 0.25% trypsin/EDTA for 5 min to remove fibroblasts, and then diluted with fresh growth medium. Tumour cells were expanded on culture plates for 2 days, diluted $5 \times 10^5$ cells in 25 μl of PBS were mixed with 25 μl of Matrigel (Corning), and implanted to the mammary fat pad of 8–9-week-old female FVB mice. To generate an orthotopic implanted 4T1 breast tumour model, $5 \times 10^5$ cells of 4T1 cells in 25 μl of PBS were mixed with 25 μl of Matrigel, and implanted to the mammary fat pad of 8–9-week-old female syngeneic Balb/c mice. To generate a spontaneous melanoma model, a droplet of tamoxifen (20 mg/ml) was swabbed on the back of Tyr-Cre-ER$^{T2}$;Braf$^{CA}$;Pten$^{loxP}$ mice for two consecutive days. After 2 weeks, flat and melanin pigmented lesions appeared, and then grew into nodular mass at 4 weeks after tamoxifen swabbing.

STING agonist 3'3'cGAMP (cGAMP, 14 μg/70 μl of PBS; Invivogen) or the same volume of PBS was intratumourally (i.t.) injected at indicated time points. For blocking TNFα, anti-TNFα monoclonal antibody (clone XT3.11, 15 mg/kg of body weight, diluted in 100 μl PBS; #BE0058, BioXcell) or isotype control antibody (clone HRPN, 15 mg/kg of body weight diluted in 100 μl PBS, Rat IgG1; #BE0088, BioXcell) was intraperitoneally (i.p.) administered. i.t. injection of TNFα (210 ng/70 μl of PBS; Sigma), IFNγ (350 ng/70 μl of PBS; R&D systems) or the combination (in 70 μl of PBS) was performed at day 13 after tumour implantation. For inhibition of AKT pathway, AKT 1/2 kinase inhibitor (AKTi, 50 mg/kg of body weight; Sigma #A6730) was i.p. injected with i.t. PBS (70 μl) or cGAMP (14 μg/70 μl) at indicated time points before sampling. To avoid confounding effect caused by

i.t. injection per se, the same volume (70 μl) of PBS was injected into the tumour of control group. To prevent the backflow leakage, i.t. injection was performed slowly over 1 min while holding the injection site with smooth forceps, followed by gentle manual compression for 15 sec. Tumour volume was calculated according to the formula $0.5 \times A \times B^2$, where $A$ is the longest diameter of a tumour and $B$ is its perpendicular diameter.

**Cell culture and treatments**. All tumour cells for this study were cultured in Dulbecco's modified Eagle medium (DMEM) supplemented with 10% foetal bovine serum, penicillin (100 U/ml), and streptomycin (100 mg/ml) at 37 °C with 5% $CO_2$. LLC cells were transduced with lentiviral particles encoding shRNA targeting STING (Santa Cruz, #sc-154411) or empty control (Santa Cruz, #sc-108080) following the manufacturer's protocol. Briefly, $5 \times 10^4$ LLC cells are transduced with $1 \times 10^5$ infectious units of virus of each lentiviral particle. After incubation with puromycin for 2 days, 6 puromycin resistant colonies were picked and expanded. Knockdown efficiency was evaluated by qRT-PCR and immunoblotting in each expanded sample. Pooled primary cultured HUVECs (Lonza) were grown in EGM2 medium (Lonza), and passages between 3-6 were used for the experiments.

**Cell viability assay**. Cell viability assay was performed using Cell Counting Kit-8 (CCK-8, Dojindo Molecular Technologies) following manufacturer's instruction. HUVECs, LLC or MMTV-PyMT tumour cells were seeded onto 96-well plate at a density of 5000 cells/well. After 24 h of incubation with DMEM (for LLC and MMTV-PyMT tumour cells) or EGM2 medium (for HUVECs), each well was added with indicated agents such as cGAMP (0–1.0 mM), TNFα (200 ng/ml), IFNγ (300 ng/ml) or AKTi (10 μM), and then 24 h later each well was incubated for 4 h with 10 μL of CCK-8 solution followed by measurement of the absorbance of 450 nm using microplate reader (Biotek).

**HUVEC apoptosis assay**. $5 \times 10^4$ HUVECs were seeded into 8-well glass slide (Lab-Tek, Thermo Fisher) with EGM2 medium, and after 24 h treated with TNFα (200 ng/ml), IFNγ (300 ng/ml), or in combination of TNFα and IFNγ with or without AKTi (10 μM). After 6 h, cells were fixed with 4% PFA for 10 min at RT followed by several washes. Cells were blocked with 5% goat serum in PBST (0.3% Triton X-100 in PBS) and then incubated overnight at 4 °C with anti-cleaved caspase 3 antibody (rabbit, #9661, Cell Signalling Technology,). After several washes, samples were incubated for 1 h at RT with following antibodies: Alexa Fluor 488-conjugated anti-rabbit IgG (#111-545-144, Jackson ImmunoResearch), Alex Fluor 594-conjugated phalloidin (#A12381, Invitrogen). Nuclei were stained with 2,6-diamidino-2-phenylindole (DAPI, #D1306, Invitrogen) with 1:1,000 dilution. The samples were then mounted with fluorescent mounting medium (DAKO) and immunofluorescent images were acquired using an LSM880 confocal microscope (Carl Zeiss).

**Histological analyses**. For immunofluorescence (IF) staining, primary tumours were fixed for 12 h in 4% paraformaldehyde (PFA), dehydrated in 30% sucrose solution for 24 h, and embedded in tissue freezing medium (Leica). Frozen blocks were cut into 50 μm sections. For IF staining of liver, lung, kidney, and spleen, the mice were anesthetized and perfused with 1% PFA. For IF staining of lung, the lungs were inflated with 1% agarose through the trachea before 1% PFA perfusion. The harvested organs were fixed in 4% PFA for 4 h at 4 °C, cut into 120 μm sections using vibratome (Leica), and then fixed in 1% PFA for 1 h at 4 °C followed by several washes. For IF staining of retina, whole-mounted retinas were used as previously described[60]. Briefly, whole-mount of retinas and retinal pigment epithelium (RPE)-choroid-sclera complexes was performed. Eyeballs were enucleated and fixed in 4% PFA for 20 min at RT. The retina or RPE-choroid-sclera complexes were dissected from the eyeball and fixed in 1% PFA for 1 h at RT followed by several washes. Sections or whole-mount retina were blocked with 5% goat serum in PBST (0.3% Triton X-100 in PBS) and then incubated overnight at 4 °C with following primary antibodies with 1:200 dilution: anti-CD31 (hamster, clone 2H8, #MAB1398Z, Millipore), anti-cleaved caspase 3 (rabbit, #9661, Cell Signalling Technology), anti-Ter119 (rat, #14-5921-82, eBioscience), anti-GLUT1 (rabbit, #07-1401, Millipore), anti-SELP (rat, #550289, BD Bioscience), or anti-PGF (rabbit, #MBS9607235, BIOSS). After several washes, the sections were incubated for 2 h at RT with following secondary antibodies with 1:1000 dilution: Alexa Fluor 488- or 647-conjugated anti-rabbit IgG (#111-545-144 and #111-605-144, Jackson ImmunoResearch), Alexa Fluor 594- or 647-conjugated anti-hamster IgG (#127-585-160 or #127-605-160, Jackson ImmunoResearch,) or Alexa Fluor 594-conjugated anti-rat IgG (#112-585-167, Jackson ImmunoResearch). Nuclei were stained with DAPI (Invitrogen) with 1:1,000 dilution. The samples were then mounted with fluorescent mounting medium (DAKO) and immunofluorescent images were acquired using an LSM800 or LSM880 confocal microscope (Carl Zeiss). To examine AKT activation in the tumour ECs, we purchased a paraffin embedded, sectioned tissue array containing human breast ductal adenocarcinomas and their adjacent normal tissues (31 cases of patients) (USBiomax, #OD-CT-RpBre03-004). The sections were deparaffinised, antigen retrieved, incubated with a primary antibody (phospho-AKT Ser473 antibody, #4060, Cell Signaling Technology), amplified the signal with the chromogen, counterstained, and mounted for

visualization according to the manufacturer's instruction (Ventana BenchMark XT Staining Systems). The images were acquired using Axio Zoom.C16 (Carl Zeiss).

**In vivo vascular leakage and perfusion assay**. We evaluated tumour vessel leakage by intravenous injection of 100 μl of rhodamine-conjugated dextran (25 mg/ml, 70 kDa, Sigma) 30 min before euthanization. For vascular perfusion analysis, 100 μl of DyLight 488-conjugated Lycopersicon esculentum lectin (1.0 mg/ml, Vector laboratory) was intravenously injected 30 min before euthanization. Mice were anesthetized and perfused by intracardiac injection of 1% PFA to remove circulating dextran or lectin.

**Flow cytometry analysis**. Harvested samples were digested with enzyme buffer containing 2 mg/ml collagenase type2-II (Worthington), 0.1 mg/ml DNase I (Roche), and 0.8 mg/ml Dispase (Gibco) at 37 °C for 30 min and filtered with a 40 μm nylon mesh to remove cell clumps. ACK lysis buffer was added and incubated for 5 min at RT to remove RBC. The single suspended cells were incubated in FACS buffer (5% bovine serum in PBS) with following antibodies with 1:200 dilution: BV711 rat anti-mouse CD45 (#563709, BD biosciences), BV421 rat anti-mouse CD31 (#562939, BD biosciences), APC/cyanine7 anti-mouse CD45.2 (#109824, Biolegend), FITC anti-mouse CD3ε (#100306, Biolegend), PE anti-mouse CD4 (#100512, Biolegend), APC anti-mouse CD8 (#100712, Biolegend), FITC anti-mouse CD45 (#553079, BD biosciences), PE anti-mouse CD11c (#117308, Biolegend), BV650 anti-mouse CD80 (#104732, Biolegend), PE/cyanine7 anti-mouse CD86 (#105014, Biolegend), Pacific blue anti-mouse MHC-II (#107620, Biolegend), PE anti-mouse CD11b (#12-0112-82, eBioscience) or APC-eFluor780 anti-mouse F4/80 (#47-4801-82, eBioscience). Concomitant nucleic acid staining was performed to determine live or death cells using DAPI (#564907, BD biosciences), V510 Ghost Dye (#13-0870-T500, Tonbo) or Sytox Green (#S7020, Invitrogen) with 1:1,000 dilution. Data were obtained with FACS aria II (BD Bioscience) and analysed using FlowJo (Tree Star Inc., Ashland, OR) software. We analysed following cell subsets: (i) dead cells, gated as Sytox Green+ cells; (ii) live ECs, gated as Sytox Green-/CD45-/CD31+ cells; (iii) CD8+ T cells, gated as DAPI-/CD45+/CD3ε/CD4-/CD8a+ cells; (iv) dendritic cells, gated as Ghost Dye-/CD45+/CD11c+ cells; (v) TAMCs or macrophages, gated as DAPI-/CD45+/CD11b+/F4/80+ cells; (vi) T lymphocytes, gated as DAPI-/CD45+/CD3ε + (Supplementary Fig. 13). The expression levels of CD80, CD86 and MHC-II on DC were presented as mean fluorescence intensity and compared among treatment groups.

**Quantitative real-time RT-PCR**. Total RNA from the sample was extracted using RNeasy Plus Mini Kit (Qiagen) according to the manufacturer's instructions. Total RNA was reverse transcribed into cDNA using GoScript$^{TM}$ Reverse Transcription Kit (Promega). Then, quantitative real-time PCR was conducted using FastStart Sybr Green Master mix (Roche) and QuantStudio$^{TM}$ 5 Real-Time PCR System (Applied Biosystems$^{TM}$). GAPDH was used as a reference gene. PCR primer sequences are listed below. *Tmem173* forward CGCACGAACTTGGACTACTG, reverse AAACATCCAACTGGACTGGACAT; *Gapdh* forward AGGTCGGTGTG AACGGATTTG, reverse TGTAGACCATGTAGTTGAGGTCA.

**Immunoblotting**. For immunoblotting analysis, cultured or sorted cellular samples were lysed with RIPA buffer supplemented with protease and phosphatase inhibitors (Cell Signalling Technology). After sonicated and denatured, the lysates were loaded on SDS-PAGE gel and transferred to nitrocellulose membranes. The membrane was blocked for 1 h at RT with 4% BSA in TBST (0.1% Tween 20 in TBS) and then stained with primary antibodies with 1:1,000 dilution overnight at 4ºC. Following antibodies were used as primary antibodies: anti-cleaved caspase 3 (#9661, Cell Signaling Technology), anti-STING (#NBP2-24683, Novus Biologicals), anti-GAPDH (#2118, Cell Signaling Technology), anti-FOXO1 (#2880 S, Cell Signaling Technology), anti-phospho-FOXO1(T24)/FOXO3a(T32) (#9464 S, Cell Signaling Technology), anti-AKT (#9272 s, Cell Signaling Technology), anti-phospho-AKT(S473) (#4060 s, Cell Signaling Technology), or anti-α tubulin (#sc-69970, Santa Cruz Biotechnology). After several washes, secondary antibodies were stained with 1:5,000 dilution for 1 h at RT and detected using Immobilon Western Chemiluminescent HRP substrate (Millipore). Following antibodies were used as secondary antibodies: anti-rabbit IgG HRP linked (#7074 S, Cell Signaling Technology) or anti-rat IgG HRP linked (#7077 S, Cell Signaling Technology). Imaging was performed on Amersham Imager 680 (GE healthcare).

**Enzyme-linked immunosorbent assay (ELISA)**. Tumours were sampled at indicated time points, transferred to Precellys lysing kit (Bertin) containing RIPA buffer supplemented with protease and phosphatase inhibitors (Cell Signalling Technology), and then homogenized with Precellys tissue homogenizer (Bertin). Samples were centrifuged at 12,000 *g* for 10 min and supernatant was collected. Protein concentrations were measured using Bicinchoninic acid (BCA) assay and normalized to contain the same amount of total protein for each assay. TNFα, IFNβ, and IFNγ were measured using ELISA kits (R&D systems) by microplate reader (Biotek) according to the manufacturer's instruction.

**Bulk RNA-sequencing (RNA-seq) and analysis**. Normal mammary fat pads and the tumours of *MMTV-PyMT* spontaneous breast tumours and implanted breast

tumours were harvested after 3 h of PBS or cGAMP injection. The samples were digested with enzyme buffer containing 2 mg/ml collagenase type II (Worthington), 0.1 mg/ml DNase I (Roche), and 0.8 mg/ml dispase (Gibco) for 30 min at 37 °C and filtered with a 40 μm nylon mesh. Cell suspensions were incubated with ACK lysis buffer for 5 min at RT to remove RBC and then stained with anti-CD31 microbeads (#130-097-418, Miltenyi Biotec). CD31+ ECs were positively selected using LS columns (#130-042-401, Miltenyi Biotec) and QuadroMACSTM Separator (#130-090-976, Miltenyi Biotec,) according to manufacturer's instructions. The CD31+ samples from MACS sorting were stained with Sytox Green (#S7020, Invitrogen,), BV711-conjugated anti-mouse CD45 (#563709, BD bioscience,) and BV421-conjugated CD31 (#562939, BD bioscience). ECs, gated as Sytox Green-/CD45-/CD31+, were sorted by FACS aria II (BD Bioscience). RNA was extracted using RNeasy Mini Kit (Qiagen) according to manufacturer's instruction, and RNA integrity number (RIN) was obtained by Bioanalyzer 2100 (Agilent Technologies). We constructed cDNA library with RNA with RIN over 9 using NEBNext® Ultra™ II Directional RNA Library Prep Kit (New England Bio-Labs) according to manufacturer's instruction. Libraries were validated with the BioAnalyzer and quantified by qPCR and Qubit Fluorometric Quantitation (Thermo Fisher). The NextSeq 500/550 mid Output v2 Kit was used for sequencing with a NextSeq 500 (Illumina) to generate 75-bp pair-end reads. The quality assessment of raw sequence data was performed using FastQC (Version: FastQC 0.11.3, http://www.bioinformatics.babraham.ac.uk/projects/fastqc/). No samples were discarded from the analysis. RNA-seq data analysis was performed as described previously[61] with some modifications. Sequenced reads were aligned to mouse reference genome (mm10) with STAR (version 2.7), and the aligned reads were used to quantify mRNA expression by using HTSeq-count (version 0.6.1). Unsupervised hierarchically clustered heatmap, principal component analysis (PCA), gene ontology analysis of PCA axis, and MA plot for differentially expressed genes were analysed using iDEP.91[62]. Gene set enrichment analysis (GSEA) was performed with Hallmark (H) or KEGG (CP) gene set collections of the Molecular Signature Database (https://www.gsea-msigdb.org/gsea/msigdb/index.jsp).

**Droplet-based single-cell RNA-sequencing (scRNA-seq)**. For scRNA-seq on LLC tumour ECs, FACS-sorted live ECs were collected 24 h after treatment of PBS or cGAMP injection in each one mouse respectively. For scRNA-seq on breast tumour ECs (*MMTV-PyMT* mice and their implantation mice), live ECs were collected from two mice for each group after 3 h of treatment with i.t. PBS or cGAMP injection. Live EC single cells were sorted and processed using 10X Chromium Single cell 3'Reagent Kit v3 (10X genomics) according to manufacturer's instructions. Briefly, cells were suspended in 0.5% BSA solution and mixed with RT reagent mix and RT primer then added to each channel of 10X chips. Cells were separated into Gel Beads in Emulsion where RNA transcripts from single cells were barcoded and then cDNA libraries were constructed and amplified. We used SPRI beads (Beckman Coulter) for appropriate size selection of cDNA and ligated with adaptor, and performed sample-index PCR. Double-sized size selection using SPRI beads was followed and then final library constructs were diluted in 10-fold to run on the Agilent Bionanalyzer High Sensitivity Chip. Single-cell library sequencing was conducted using Illumina Hiseq-X platform. The sequenced data of single-cell libraries were demultiplexed and aligned to mouse reference genome (mm10) by Cell Ranger software 3.0.0 provided by 10X Genomics. Raw expression matrices were then built by using Read10X function in Seurat (version 3.1.1). For cell-based quality control, low-quality cells detected with less than 500 genes and putative dead cells with high mitochondrial gene percentage (> 10% of total Unique Molecular Identifiers (UMI) counts) were discarded. For gene-based filtering, genes expressed in less than 3 cells were removed. The quality metrics of scRNA-seq was provided in Supplementary Table 1. After removal of unwanted cells and genes, normalization of raw expression matrices was performed by dividing UMI counts for each gene per cell by the total sum of UMI counts in a given cell, multiplied by 10,000 and log-transformed, producing log-counts per million (CPM) like values. Then, gene-based scaling was performed while regressing out variables such as number of UMIs and mitochondrial gene percentage. For clustering and downstream analysis R package Seurat was used. First, variable genes for datasets were identified by FindVariableFeatures function in Seurat with options: selection.method = "vst" and nFeatures = 2500. Then, independent datasets were integrated using FindIntegrationAnchors and IntegrateData function in Seurat. Then, Principal Component Analysis was performed, and top 30 principal components were used for further analysis including Uniform Manifold Approximation and Projection (UMAP) for two-dimensional visualization, building shared nearest neighbourhood graph and Louvain algorithm for cluster identification. After initial clustering, contaminating cell types that are Pecam1- were removed from the dataset. Finally, another round of clustering was performed on remaining Pecam1+ cells. Cluster-specific marker genes were regarded as differentially expressed genes for each cluster identified using FindMarkers function in Seurat on the RNA assay of Seurat object with following options: test.use = "MAST", min.pct = 0.25, min.diff.pct = 0.25. To perform gene set enrichment analysis from single cell RNA datasets in LLC tumour ECs, pseudo-bulk RNA expression matrices were built by bootstrapping. Triplicates of average normalized expression, computed on 1000 random sampled cells, for each dataset were made. The resulting pseudo-bulk expression matrices were used as input for GSEA on

gene set collections from the Molecular Signature Database (https://www.gsea-msigdb.org/gsea/msigdb/index.jsp). In scRNA-seq of breast tumour ECs, the differentially expressed genes in implanted tumour *versus MMTV-PyMT* tumour ECs were extracted and provided on gene ontology analysis through DAVID 6.8.

**Morphometric analyses**. Density measurements were performed with ImageJ software (https://imagej.nih.gov/ij/). The cleaved caspase 3+ apoptosis area, dextran leakage area, Ter119+ haemorrhage area, and GLUT1+ hypoxic area were presented as percentages per total mid-sectional tumour area. Cleaved caspase 3+ ECs, SELP+ ECs, and vascular perfusion were presented as percentages per CD31+ area in total mid-sectional tumour area. Apoptosis of whole tumour cells was presented as percentages per cleaved caspase 3+ area in total mid-sectional tumour area.

**Statistical analyses**. Data are presented as mean ± standard deviation (SD). Statistical significance was determined between the two groups with two-tailed t-tests (when parametric test was appropriate). To assess the statistical significance in more than two groups, Welch's one-way ANOVA followed by Dunnett's T3 test (when parametric test was appropriate) or Kruskal–Wallis test followed by Dunn's test (when non-parametric test was appropriate) was used. Statistical analysis was performed with Prism 8 (GraphPad). Statistical significance was set to $p < 0.05$.

**Reporting summary**. Further information on research design is available in the Nature Research Reporting Summary linked to this article.

## Data availability

Single-cell RNA-sequencing data and bulk RNA-sequencing data are available in National Center for Biotechnology Information's Gene Expression Omnibus under accession number GSE159013 (scRNA-seq on tumour ECs of LLC tumour), GSE159203 (bulk RNA-seq on ECs from normal mammary pad and breast tumours), and GSE171451 (scRNA-seq on tumour ECs from *MMTV-PyMT* spontaneous breast tumours and implanted breast tumours). The remaining data are available within the Article, Supplementary Information or Source Data file. Further information and requests for resources and reagents should be directed to and will be fulfilled by Gou Young Koh (gykoh@kaist.ac.kr). Source data are provided with this paper.

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

## Acknowledgements

The authors thank Sujin Seo and Junho Jung (IBS) for their technical assistances. We also thank Prof. Chan Kim (CHA University Cha Bundang Medical Center, Seongnam, South Korea) for *STING* KO mouse and Prof. Yoshiaki Kubota (Keio University, Tokyo, Japan) for *VE-cadherin-Cre*-ER[T2] mice. This study was supported by the Institute for Basic Science (IBS-R025-D1-2015 to G.Y.K.) funded by the Ministry of Science, ICT and Future Planning, Korea.

## Author contributions

S-H.J. designed and performed all experiments, analysed and interpreted the data. S.-H.J. and G.Y.K. co-wrote and edited the manuscript. M.J.Y. performed scRNA-seq and analysed the datasets. S.H.C. supported mouse works. J.M.K. performed bulk RNA-seq and analysed the datasets. G.Y.K. directed and supervised the project.

## Competing interests

The authors declare no competing interests.
