## [Peer Review File · Nature Communications]

REVIEWER COMMENTS

Reviewer #1 (Remarks to the Author): with expertise in cancer immunotherapy and STING

In this manuscript Jeong et al explore the role of STING agonism on inducing endothelial cell(EC) apoptosis in the tumor vasculature. First, they examine subcutaneously implanted LLC tumors and observe substantial EC apoptosis which is prevented in STING golden ticket KO mice, but not in mice with intact STING plus LLC specific STING suppression. They next dissected the impact of cGAMP treatment on endothelial cells in vivo by performing scRNAseq, and observed dependency upon TNF signaling, particularly derived from tumor macrophages. They then explored the role of cGAMP injection in primary breast tumors and failed to observe significant apoptosis, observing co-induction of genes related to PI3K signaling. They therefore combined cGAMP treatment with AKTi in cultured HUVECs and observed restoration of killing, prompting a therapeutic combination in both MMTV and LLC models where they observed AKTi mediated potentiation of cGAMP efficacy in vivo.

In general this is a strong manuscript with an extensive amount of work. I have several comments that should be addressed, however

1. The STING KO experiments systemically vs in tumors seems quite relevant and should not be relegated to Extended Data - at least some of these results should be presented in the main figures. Also, the authors concluded that the golden ticket KO system proves an effect of STING signaling from the microenvironment, but they should be clear that STING is also being disrupted in the microvasculature, so could this simply be that cGAMP has no direct effect on endothelial target cells. The only conclusion that can really be made by from experiments is that the effect is not due to downstream STING activation in tumor cells.
2. The macrophage STING KO data in Fig 3 does address this and the TNF neutralization study in particular is nice. But did the authors measure IFN-beta and whether this is also suppressed by macrophage STING knockout? This is a more relevant macrophage target of cGAMP than IFN-gamma, and could be playing a role on the EC phenotype too.
3. Extended data Fig 7 – these in vitro data are also somewhat buried by being in the Extended Data, but are quite important. They also need to assess the effects of IFN-beta on HUVECs, since as above I suspect this is the major interferon being released downstream of cGAMP in the TME by macrophages. Also did they ever look at the direct effects of cGAMP on HUVEC cell viability, either alone or in combination with TNF?
4. Also regarding the in vitro data in HUVECs did the authors confirm the impact of AKTi on inhibiting AKT activity (eg pAKT or pS6 levels)? Immunoblotting for these markers would also answer the question of pAKT or pS6 is being activated by TNF in these cells, or whether it is providing a parallel survival signal. Further details about the AKT inhibitor used should be provided and mentioned in the text, and its previous use cited to validate its potency/specificity. I had to look up methods and the Sigma website to figure out this was from Merck (? MK-2206).
5. Main Fig 6 – It would be ideal if they had pharmacodynamic evidence that the AKTi was also effectively inhibiting the pathway in vivo, as measured by endothelial cell pAKT or pS6.
6. Discussion section – The discussion section is quite brief and could be strengthened significantly

Reviewer #2 (Remarks to the Author): with expertise in immune sensing and anti-tumor immunity

Manuscript entitled "Refractoriness of STING Therapy Can be Oppressed by AKT Inhibitor Through Effective Tumour Vascular Disruption in Primary Tumour" by Jeong et al presented intriguing observations that intratumoral injection of STING agonist cGAMP, when in combination with the AKT1/2 inhibitor, induced a massive apoptosis of both tumor ECs and tumor cells and resulted in a remarkable suppression of primary tumor. Authors suggest that an interplay between TAM and ECs

mediated by the TNFalpha-TNFR1 signaling axis is involved in the death of ECs and tumor vascular disruption. As a result, authors proposed a new therapeutic strategy for improving anti-tumor effects of primary tumors that were refractory to single STING agonist therapy.

Overall, it is a manuscript focusing on a heatedly pursuing topic. The effects on combinational use of STING agonist and AKT inhibitor not only agreed with a recent report for an enhancement of cGAS-STING signaling by HER2-AKT inhibition (Chen et al, Nature Cell Biology, 2020; should be referred), but also set cGAMP as a tumor-specific VDA in primary tumor, by which it induced apoptosis of proliferative ECs subpopulations. The qualities of data are generally sound and support their claims. However, a few concerns prevent the acceptance of current manuscript.

Major points:

- 1) Authors detected a reduction of live ECs as early as 3 h post cGAMP injection (such as Fig. 1e, 1h/1i). However, the levels of CD31 positive ECs were unchanged (Fig. 1g), or even increased in some observations (Fig. 1l and 2g). Very confusing.
- 2) In Extended Data Fig. 2, the authors revealed an entire abrogation of both cGAMP-induced EC apoptosis and anti-tumor growth in STING KO mice. On the other hand, a combinational treatment of TNFalpha and IFNgamma significantly facilitated the EC apoptosis (Extended Data Fig. 5b, c). Did authors examine the combined effects of TNFalpha and IFNgamma on EC apoptosis in STING KO mice, in order to validate the TNFalpha-TNFR1 signaling axis interplayed between TAM and ECs?
- 3) A discussion should be amended: does AKTi potentiate the cGAMP-induced anti-tumor effects in spontaneous breast cancer by regulating the survival pathway of ECs and/or via enhancing the innate immune responses of tumor and bystander cells to cGAMP?
- 4) Introduction and Discussion are over sketchy.

Minor points:

- 1) Extended Data Fig. 1j, lacks annotations of PBS and cGAMP.
- 2) Extended Data Fig. 3d, lacks the statistical analysis.
- 3) The sequence of Extended Data Fig. 5a-c and 5d-f better to be exchanged.
- 4) An overlap of text in Fig. 5d.
- 5) Spelling errors: Line 23 of P9; Line 26 of P9.

Reviewer #3 (Remarks to the Author): with expertise in vascular oncology

In this study, Jeong et al. investigate the mechanism how STING agonists (cGAMP, DDXAA) disrupt tumor-associated blood vessels and reduce primary tumor growth in several mouse tumor models. Using STING and TNFR1 knockout mice (global ko and myeloid-specific ko in case of STING), they show that upon STING activation, tumor-associated myeloid cells release TNF-a that induces endothelial apoptosis within the tumor, reducing tumor growth in transplanted (LLC, B16, MMTV-PyMT-derived cell line) but not in a transgenic (MMTV-PyMT) tumor model. The resistance of MMTV-PyMT tumors towards STING agonists could be overcome using an AKT inhibitor.

This study is experimentally well performed, and the data is presented very clearly. My main criticism is that many findings presented here are not novel. It has been shown a long time before (even before the discovery of STING) that flavone acetic acid analogues such as DDXAA can act as vascular disrupting agents that trigger endothelial apoptosis and reduce primary tumor growth in multiple mouse tumor models, accompanied by a marked IFN response. It has also been shown that this involves macrophage-derived cytokines, including TNF-a (see Ref 27; Baguley, Lancet Oncol 2003, PMID 12623359; and references therein). The most significant new finding in this study is that a transgenic breast cancer model (MMTV-PyMT) is resistant to STING agonists, requiring additional AKT inhibition to induce endothelial apoptosis and tumor growth delay. However, this has been investigated in only one transgenic tumor model, and the clinical relevance of this observation is not clear.

Specific comments:

Major:

1. Are blood vessels equally resistant to STING agonists in other transgenic mouse cancer models? What is the clinical relevance of the finding that endothelial cells can only be forced into apoptosis using both STING agonists and AKT inhibitors in the MMTV-PyMT model? Are there any indications

that the AKT pathway is over-activated in the tumor-associated endothelium of (breast) cancer patients compared to normal tissue?

2. Other primary breast cancer models (e.g. E0771 or 4T1) should be investigated to elucidate whether the difference in STING agonist responsiveness is due to the cancer type or due to the type of model (e.g. transplanted cell line vs. transgenic, spontaneous model).

3. Is STING agonist / AKTi therapy responsiveness in the MMTV-PyMT model dependent on CD8+ T cells? Otherwise, the authors should reduce their statements in the results and discussion sections regarding anti-tumor immunity-promoting effects of this therapeutic approach.

4. "Proliferative" ECs are present in the MMTV-PyMT, yet this model does not respond to STING agonists alone. Thus, the conclusion that STING agonists predominantly target "tip-like" and "proliferative" ECs is not well founded but the data, and should be toned down. Furthermore, it would be interesting which type of vessels preferentially resist the treatment in the sensitive models. Are those predominantly large, mature (pericyte-covered) vessels as it was described for some anti-angiogenic therapies?

Minor:

1. Activated STING is known to induce a type-I interferon response, yet the authors chose to study the expression levels and function of type-II interferon (gamma) in their models. Why was that? Furthermore, the authors' finding that IFN-gamma could not induce vascular regression (Fig. S5a-c) is somewhat in contrast to a previous publication by Kammertoens et al (Nature 2017, PMID 28445461). This should be discussed at least.

2. The sequencing data (single cell and bulk) generated in this study should be uploaded to a repository and need to be available to the reviewers and the public.

3. Why was the transplanted MMTV-PyMT cell line tumor model treated so often (Figs. 4k-m)? Would treatment twice (as was done with the other transplantable models) suffice to achieve the same effects?

4. The volume used for intratumoral injections (70 µl) appears to be huge. Such an injection would likely lead to a massive pressure increase in the tumor tissue, which could lead to confounding effects. How did the authors avoid backflow from the injection site? Please comment.

5. Fig. 1E: The dotted outline of the blood vessels is very unclear and seems somewhat arbitrary. Please provide better / additional images to demonstrate how the vessel outlines were identified.

6. Fig. 2E: There is one gene name missing in this graph (there is one more row in the heat map than gene names). Furthermore, the color scheme (red - violet) is somewhat unfortunate, as it makes a poor contrast between highly and lowly expressed genes. Please consider changing to a red-blue color scheme as in Fig. 5.

4. Some reference seem to be unsuitable to support the sentences that cite them, e.g. reference 10 and 28. Please find better or additional references or rephrase the corresponding statements.

5. The LysM-Cre mouse does not specifically target (tumor-associated) macrophages, but many more myeloid cell types, including monocytes and granulocytes. Please rephrase the corresponding statements in the results.

6. CD8+ T cells should not be called "cytotoxic" unless expression of cytotoxic markers (e.g. granzyme) has been shown.

7. I would suggest to call the MMTV-PyMT tumors "transgenic" or "spontaneous" instead of "primary" to avoid confusion (also a transplanted tumor could be called "primary").

8. The AKT - mTOR pathway might not only protect endothelial cells from apoptosis, but could also directly interfere with STING activation in macrophages, e.g. via ULK1/2. Please discuss.

9. The discussion chapter is very short and too focused on the role of tumor immunity, which has not been demonstrated to be relevant for the treatment effects observed by the authors (see major comment 3). I would suggest a major revision of this chapter, including previous studies regarding the anti-tumoral effects of DDXAA and other flavone acetic acid derivatives.

Reviewer #4 (Remarks to the Author): with expertise in vascular oncology and single cell RNA sequencing

General:

In this manuscript, the authors use a range of tumor mouse models to determine the antitumor

effects of STING agonists. They show that intratumoral injection of cGAMP, a STING agonist, induces apoptosis of tumor endothelial cells in subcutaneous xenograft models of LLC- and B16-induced tumors, but not in primary breast tumors. The authors then show that apoptosis in implanted tumors is stimulated by TNF α -TNFR1 signaling, while in primary tumors, additional inhibition of AKT is required for efficient induction of apoptosis via this pathway. These findings have implications for treatment with STING agonists, suggesting that combination with AKT inhibitors might represent a more efficient strategy. As such, the manuscript presents potentially important data, but in its current form, it suffers from a number of shortcomings that will need to be addressed, as highlighted below.

Major comments:

1. The authors claim in the introduction: "Moreover, most previous studies have been conducted in subcutaneous tumour implantation models, which have limited ability to reflect human primary tumours." However, this reviewer feels that also this study was performed to a large extent in subcutaneous tumor implantation model. Crucial experiments, such as the single cell analysis, were done in subcutaneous LLC. Why was it not (also) performed in a primary tumor model? The paper would benefit from switching the focus from the LLC model to primary tumor models.
2. The authors document a difference in implanted versus spontaneous tumors in MMTV-PyMT model. This is a very interesting point, which could be important for interpretation of results from animal studies. Indeed, the authors present this as the crucial point of their study, and it is thus warranted to confirm these results in another spontaneous model, for example in one of the spontaneous lung tumor models. This would be particularly appropriate given the primary focus on the LLC model, and the fact that spontaneous MMTV-PyMT tumors show large heterogeneity in the data and a reduced growth rate compared to corresponding implanted tumors (compare Fig. 4e and 4i).
3. Generally speaking, the abstract, introduction and discussion sections require work. Currently, these three sections largely repeat each other. Discussion is limited to a summary of results, and the authors should instead discuss their findings in a context of the already published literature and how their results relate to a therapy of human patients. Tumors in patients are always spontaneous and can hardly be treated by intratumor injections. Furthermore, authors should discuss if similar differences between ECs in spontaneous and implanted tumors have ever been observed for other pathways that do not involve STING?
4. Among others, the authors should discuss a recent study in JCI, where LLC as well as breast MMTV-PyMT tumor models were extensively used for treatment with STING agonists (PMID: 31343989) e.g., do they also see a recovery of ECs and tumor vessel normalization at later timepoints after treatment as was reported in that study? The authors should also remove from their manuscript data already published elsewhere (such as extended data Fig.2 and more) and focus on the novel aspects of their own study.
5. Tumor ECs are stained by CD31, but how are tumor cells identified? If "tumor cells" are identified just as a whole tumor area, the authors should instead call it by its name - these are not tumor cells, but a whole bulk of tumor tissue including all the stromal cell types that are present. Statements like: "Extensive apoptosis was observed in 24% of tumour ECs, and in 27% of tumour cells (Extended Data Fig. 1i-k)" are not correct.
6. Following-up on the previous point, the percentages (and also fold changes in other cases) specified for each of these experiments make the text very difficult to read.
7. In their scRNAseq experiment, the authors show in Fig. 2d that single cells isolated from PBS-treated and cGAMP-treated animals overlap. This is also reflected in the clustering analysis, where no cluster is specific for either PBS- or cGAMP-treated group, indicating that there indeed is no difference. In light of this result, comparisons of gene expression between these two samples make little sense. Quantitative comparison of expression levels can however be done on a protein level e.g. immunostaining of tissue slides, as the authors did in case of SELP staining.
8. In Fig. 2f the authors show fractional distribution of individual clusters. The preparation of single cell suspensions used for scRNAseq is not specified in the methods section, but from the context it seems that the analysis was done on n=1 PBS-treated versus n=1 cGAMP-treated sample. If that is the case, any conclusions about the fraction size in the two samples are irrelevant – differences between the two samples might be just caused by chance. In case the authors want to compare the fractional contribution between conditions, they would need to generate at least three samples of each condition. The authors therefore need to refrain from quantitative claims about the distribution of EC phenotypes in their manuscript, based on their scRNAseq data (e.g. "compared

to PBS-treated ECs, the cGAMP-treated ECs exhibited a 13% increase of stalk-like ECs, and 9% and 4% reductions of tip-like and proliferative ECs, respectively (Fig. 2f). Another reason, why the comparison of fractions is not correct is that the treatment induces massive apoptosis of ECs. These dying cells are not captured in sequencing droplets. Hence, we are looking at an unknown fraction of surviving cells. How can we know that the stalk-like cell fraction increased, and the difference is not caused by a different fraction of apoptotic cells in this sample?

9. In Fig. 2e authors show that PGF is a marker of "stalk-like cells" while in Fig. 5e this gene is a part of a "tip-like" EC signature. As Placental growth factor is a recognized tip cell marker (del Toro et al., 2010; Strasser et al., 2010; Zhao et al., 2018, Goveia et al., 2020) the heatmap shown in Fig. 2 suggests that the clustering is sub-optimal, i.e. cluster of stalk-like ECs probably contains also part of tip ECs.

10. To facilitate an easy exploration of their data by the community, the authors should provide the reader with a list of top-ranking genes per cluster in their analysis in a supplementary table. Also, they should make available the quality metrics for their experiment to allow assessment of data quality – details of sample preparation (including info from how many animals each sample was prepared), number of sequenced cells per sample, number of genes/UMIs per cell, etc.

11. Finally, the authors include scRNAseq data on LLC implanted tumors and bulk RNAseq data on MMTV-PyMT tumors, as these data were not integrated and the models are not comparable, the conclusions like "These findings recapitulated the scRNA-seq data indicating that tip-like and proliferative EC subpopulations were vulnerable to STING activation-induced apoptosis,.. " are not substantiated. Similarly, the authors cannot make conclusions about tip and stalk like cell susceptibility to STING agonists in breast tumor as they don't know what the expression profile of EC phenotypes in breast tumors is. Are there stalk like cells and tip cells even present as separate clusters in breast tumors?

12. In Figure 3g, the authors included results only from cGAMP treated animals. Results in PBS-treated animals should be included as a control.

13. The order of panels as well of accompanying text describing data in Extended Figure 5 is mixed-up. At this point this section is difficult to interpret.

14. How well is cGAMP/Akt1i combination tolerated by the animals? Clearly, Akt1i is toxic to normal HUVEC (Extended data fig. 7a). Were any adverse effects observed?

Minor comments:

1. The p values indicated in figures sometimes do not fit in the image (e.g. Fig. 3e) or it is unclear to which comparison they refer (Fig. 6g). This makes the graphs overcrowded and difficult to interpret.

2. Axis titles are missing on several occasions – Fig. 2d, Fig. 6e-g.

3. Figure 1l, please use consistent color coding for CD31 staining.

4. In Figure 6, the AKT comes out of the blue sky. Could they explain better the rationale for the use of AKT inhibitors in the manuscript (not only in the discussion, but also in the introduction). Can the authors find some indications in their RNAseq data leading to the AKT pathway? It would make this part better integrated in the story.

5. Figure 6 panels i-j-k show exactly the same data. While showing the tumor growth in individual animals might have at least some added value, the bar graph shown in k does not bring anything new to the story and should be removed.

6. In Extended Data Fig. 6 it is not clear which sample represents which plot.

We have performed additional experiments and revised the manuscript to address the issues raised by the reviewers.

Reviewer #1

In this manuscript Jeong et al explore the role of STING agonism on inducing endothelial cell (EC) apoptosis in the tumor vasculature. First, they examine subcutaneously implanted LLC tumors and observe substantial EC apoptosis which is prevented in STING golden ticket KO mice, but not in mice with intact STING plus LLC specific STING suppression. They next dissected the impact of cGAMP treatment on endothelial cells in vivo by performing scRNAseq, and observed dependency upon TNF signaling, particularly derived from tumor macrophages. They then explored the role of cGAMP injection in primary breast tumors and failed to observe significant apoptosis, observing co-induction of genes related to PI3K signaling. They therefore combined cGAMP treatment with AKTi in cultured HUVECs and observed restoration of killing, prompting a therapeutic combination in both MMTV and LLC models where they observed AKTi mediated potentiation of cGAMP efficacy in vivo. In general this is a strong manuscript with an extensive amount of work. I have several comments that should be addressed, however

We appreciate this favourable and supportive comment.

1. The STING KO experiments systemically vs in tumors seems quite relevant and should not be relegated to Extended Data - at least some of these results should be presented in the main figures. Also, the authors concluded that the golden ticket KO system proves an effect of STING signaling from the microenvironment, but they should be clear that STING is also being disrupted in the microvasculature, so could this simply be that cGAMP has no direct effect on endothelial target cells. The only conclusion that can really be made by from experiments is that the effect is not due to downstream STING activation in tumor cells.

We appreciate these constructive and insightful comments. Following the reviewer's suggestion, we rearranged the Figures by transferring Extended Data Figure 2 and 3 to main Figure 2 and 3 in the revised manuscript.

In response to the reviewer's comment regarding the selective action of cGAMP on the tumour ECs, we newly made inducible, endothelial cell-specific STING deletion (STING^{ΔEC}) mice, and generated implanted LLC tumours and performed i.t. injection of cGAMP into them. We included this additional data and description into the revised manuscript (Extended Data Fig. 2).

In Results (p 6): Furthermore, to evaluate whether the cGAMP-induced tumour EC apoptosis is mediated directly through STING activation in tumour ECs, we generated an inducible, endothelial cell-specific STING-deleted mice ($STING^{i\Delta EC}$) by crossing $STING^{fl/fl}$ mice with $VE-cadherin-Cre-ER^{T2}$ mice (Okabe et al., *Cell*, 159(3), 584–596, 2014) (Extended Data Fig 2a). Littermates of $VE-cadherin-Cre-ER^{T2}$ were used as control wild type (WT) mice. Two weeks before LLC implantation, tamoxifen (2 mg) was administered to WT and $STING^{fl/fl}$ mice for five times with 2-day interval (Extended Data Fig 2b). Of note, there were no significant differences in tumour growth, tumour EC apoptosis and whole tumour cell apoptosis between WT and $STING^{fl/fl}$ mice (Extended Data Fig. 2c-g), implying that the cGAMP-induced tumour EC apoptosis was not mediated through direct STING activation of tumour ECs. Together, the cGAMP-induced tumour vascular destruction and anti-tumour growth could be derived from STING activation of bystander cells in the tumour microenvironment, rather than from intrinsic STING activation of tumour cells or tumour ECs.

Extended Data Fig. 2. STING pathway of tumour EC is not required for cGAMP-induced apoptosis. **a**, Diagram for generation of EC-specific STING deleted ($STING^{i\Delta EC}$) mice. **b–d** Diagram depicting generation of implanted LLC tumour in $STING^{i\Delta EC}$ mice, i.p. administrations of tamoxifen (2 mg), and treatment schedule of i.t. PBS or cGAMP. Comparison of tumour growth. $n = 6$ mice/group from four independent experiments. Plots and vertical bars indicate mean \pm SD. Plot indicates each individual tumour growth. P values by Welch’s one-way ANOVA test followed by Dunnett’s T3 test. **e–g**, Diagram depicting generation of implanted LLC tumour in

STING^{ΔEC} mice, i.p. administrations of tamoxifen (2 mg), treatment schedule of i.t. PBS or cGAMP, and sampling at 24 later. Representative images and comparisons of apoptosis in tumour ECs (white arrowheads) and whole tumour cells. Scale bars, 1.0 mm (yellow bars) and 100 μm (white bars). n = 6 mice/group from four experiments. Vertical bars indicate mean ± SD. P values by two-tailed t-test.

2. The macrophage STING KO data in Fig 3 does address this and the TNF neutralization study in particular is nice. But did the authors measure IFN-beta and whether this is also suppressed by macrophage STING knockout? This is a more relevant macrophage target of cGAMP than IFN-gamma, and could be playing a role on the EC phenotype too.

This is a valid point to be addressed. Our additional experiment revealed that the cGAMP treatment similarly increased IFNβ in the LLC tumours in WT and *STING*^{ΔMC} mice, which implies that TAMCs (tumour-associated myeloid cells) may not be an exclusive source of IFNβ (Abram et al., *Journal of Immunological Methods*, 2014; Bode et al., *Eur. J. Immunol.* 2016; Ivashkiv, *Nature Reviews Immunology*, 2018) and that the increased IFNβ could not play a significant role in the cGAMP-induced tumour EC apoptosis. We included this additional data and its description into the revised manuscript (Fig. 5d)

In Results (p 8): In contrast, **IFNβ and IFNγ** in the cGAMP-treated LLC tumours was **similar** between *STING*^{ΔMC} mice and WT mice (**Fig. 5c, d**). These findings suggested that **TAMCs are** main source of TNFα, while **they are** not a main source of **IFNβ and IFNγ** upon STING activation ⁴²

In Methods (p 21): TNFα, **IFNβ** and IFNγ were measured using ELISA kits (R&D systems) by microplate reader (Biotek) according to the manufacturer's instructions.

Fig. 5. TAMCs-derived TNFα is a crucial mediator for STING agonist-induced apoptosis of Tumour ECs. c, d, Diagram depicting generation of implanted LLC tumour in WT and *STING*^{ΔMC} mice, i.t. PBS or cGAMP treatment, and sampling of tumours at 6 h later. Comparisons of TNFα, IFNγ and IFNβ protein levels in tumour lysates treated with PBS or cGAMP between WT and *STING*^{ΔMC} mice. Each dot indicates a value from one mouse and n = 4 mice/group from four independent experiments. Horizontal bars indicate mean ± SD.

3. Extended data Fig 7 – these in vitro data are also somewhat buried by being in the Extended Data, but are quite important. They also need to assess the effects of IFN-beta on HUVECs, since as above I suspect this is the major interferon being released downstream of cGAMP in the TME by macrophages. Also did they ever look at the direct of effects of cGAMP on HUVEC cell viability, either alone or in combination with TNF?

We appreciate these constructive and insightful comment. Following the reviewer’s suggestion, we transferred Extended data Fig. 7 to main Figure 9 in the revised manuscript.

Like TNF α (300 ng/ml), IFN β (300 ng/ml) slightly (~15%) reduced cellular viability in primary cultured HUVECs but it did not induce an additional apoptotic effect with AKTi (10 μ M) (Fig 1a, only for the reviewer 1), while TNF α (300 ng/ml) induced an additional apoptotic effect with AKTi (revised Fig. 9e). Thus, although cGAMP stimulates IFN β in non-TAMCs, this increased IFN β is unlikely to be a major apoptotic cytokine as much as TNF α to the tumour ECs.

As already described in Extended Data Fig. 1e, cGAMP doses ranging from 0.1 μ M to 1.0 mM did not alter the cellular viabilities of cultured HUVECs. Our additional experiment revealed that the combination of TNF α and cGAMP reduced HUVECs viability by ~20% (Fig 1b, only for the reviewer 1). These findings and above (the response to comment 1) indicate that the apoptosis of tumour ECs is not directly derived from cGAMP cytotoxicity.

We show these additional findings only to the reviewer to keep the manuscript as concise as possible.

Only for reviewer Fig. 1. Viabilities of cultured HUVECs by indicated treatments. (a) IFN β (100 IU/100 μ L) or/and AKTi (10 μ M), (b) TNF α (200 ng/ml) or/and cGAMP (300 μ M) was added and incubated for 24 h. Dots indicate values from six independent experiments. Vertical bars indicate mean \pm SD. *P* values by Welch’s one-way ANOVA test followed by Dunnett’s T3 test. ns, not significant.

4. Also regarding the *in vitro* data in HUVECs did the authors confirm the impact of AKTi on inhibiting AKT activity (eg pAKT or pS6 levels)? Immunoblotting for these markers would also answer the question of pAKT or pS6 is being activated by TNF in these cells, or whether it is providing a parallel survival signal. Further details about the AKT inhibitor used should be provided and mentioned in the text, and its previous use cited to validate its potency/specificity. I had to look up methods and the Sigma website to figure out this was from Merck (? MK-2206).

5. Main Fig 6 – It would be ideal if they had pharmacodynamic evidence that the AKTi was also effectively inhibiting the pathway *in vivo*, as measured by endothelial cell pAKT or pS6.

Following the reviewer’s suggestions (comments 4 and 5), we performed additional experiment to confirm the impact of AKTi on inhibiting AKT activity in primary cultured HUVECs (revised Fig. 9a) and to validate pharmacodynamics evidence in tumour vessels (revised Fig. 9b). We also detail about the AKTi in the Text and Methods as following.

In Results (p 11): We chose AKT 1/2 inhibitor as AKT inhibitor (AKTi) and confirmed its effects by examining phosphorylation statuses of AKT at S473 and its downstream FOXO1 at T24 in cultured HUVECs and ECs of *MMTV-PyMT* spontaneous breast tumour. Consistent with previous studies^{48,49}, AKTi (10 μM) completely abolished TNFα-induced phosphorylation of AKT and FOXO1 in HUVECs (Fig. 9a), while i.p. AKTi (50 mg/kg of body weight) largely induced nuclear localization of FOXO1 in the tumour ECs (Fig. 9b). These findings indicate that AKTi effectively inhibits its pathways *in vitro* and *in vivo*.

In Methods (p. 17): For inhibition of AKT pathway, AKT 1/2 kinase inhibitor (AKTi, 50 mg/kg of body weight; Sigma #A6730) was i.p. injected with i.t. PBS (70 μl) or cGAMP (14 μg/70 μl) at indicated time points before sampling.

Fig. 9. AKT inhibitor sensitizes cultured HUVECs to TNFα-induced apoptosis
a, Immunoblot analysis of FOXO1, phosphorylated FOXO1 at threonine 24 [pFOXO1 (T24)], AKT, phosphorylated AKT at serine 473 [pAKT (S473)] and α-tubulin in cultured HUVECs at 30 min after indicated treatments. Three independent

experiments showed similar findings. **b**, Representative images of FOXO1 subcellular localization in the ECs of LLC tumour. PBS (70 μ l), cGAMP (14 μ g/70 μ l) or AKTi (50 mg/kg) was injected 3 h before sampling. White arrowheads indicate nuclear localization of FOXO1. Scale bars, 50 μ m. Five independent experiments showed similar findings.

6. Discussion section – The discussion section is quite brief and could be strengthened significantly

In response to reviewer's suggestion, we strengthened the significances of this study by inclusion of additional highlights in the revised Discussion as below.

Discussion: In this study, we examined the poorly understood role of the STING agonist cGAMP as a tumour-specific VDA, and demonstrated that this early activity of cGAMP is a key for establishing anti-tumour effect. We also showed that refractoriness of i.t. STING therapy could be relieved by addition of AKTi through effective tumour EC apoptosis in spontaneous tumours (**Extended Data Fig. 12**).

One representative VDA, flavone acetic acid, selectively block vascular flow in the tumour core by promoting tumour EC apoptosis, leading to powerful anti-tumour effect in murine tumour models (Rivera & Bergers, *Science*, 349(6249), 694–695, 2015). Its derivatives have been developed from the trait of its chemical structure possessing amenable site with active region of xanthone-4-acetic acid, which gave rise to the development of DMXAA as potent as 16-fold (Tozer et al., *Nature Reviews Cancer*, 5(6), 423–435, 2005). Interestingly, the scheme of DMXAA on anti-tumour effect has changed from VDA to immunotherapeutic agent as a STING agonist (Conlon et al., *The Journal of Immunology*, 190(10), 5216–5225, 2013; Downey et al., *PLoS ONE*, 9(6), 10–12, 2014). Currently, i.t. STING agonist therapy is generally recognized to increase the production of type I interferon in tumours, which promotes CD8⁺ T cell priming and infiltration into the tumour, eventually converting immunologically “cold” tumours to “hot” tumours (Corrales et al., *Cell Reports*, 11(7), 1018–1030, 2015; Demaria et al., *Proceedings of the National Academy of Sciences of the United States of America*, 112(50), 15408–15413, 2015; Yang et al., *Journal of Clinical Investigation*, 129(10), 4350–4364, 2019). Therefore, STING therapy has been considered a promising treatment for use in combination with immune checkpoint inhibitors, since they are more effective in hot tumours than cold tumours (Huang et al., *Nature Communications*, 10(1), 1–15, 2019). However, throughout this study, we addressed the strong and rapid apoptotic role of STING agonist, cGAMP, in tumour ECs as a VDA. Through scRNA-seq and bulk RNA-seq analyses of the tumour ECs, we revealed that tip-like and proliferative EC subpopulations were vulnerable to STING activation-induced apoptosis. More importantly, we found that TAMCs-derived TNF α was a key mediator of the extensive apoptosis of tumour ECs (**Extended Data Fig. 12**).

STING plays multifaceted roles in tumour progression depending on target cells such as tumour cell or cells composing tumour microenvironment such as ECs, cancer-associated fibroblasts, and immune cells (Bakhoun et al., *Nature*, 553(7689), 467–472, 2018; Qing Chen et al., *Nature*, 533(7604), 493–498, 2016; Gulen et al., *Nature*

Communications, 8(1), 2017; Song et al., *Scientific Reports*, 7, 39858, 2017). For instance, Yang et al. (Yang et al., *Journal of Clinical Investigation*, 129(10), 4350–4364, 2019) have recently reported that STING agonists normalize tumour vessels mediated by up-regulating the genes related to type I/II interferon and vascular stabilization and enhancing pericyte coverage. However, it occurs at several days after i.t. injection with a prior and marked reduction of tumour ECs. In comparison, this study shows that STING agonist strongly and rapidly induces tumour EC apoptosis within a day after the injection. Moreover, our findings indicate that the massive tumour cell death, following extensive cGAMP-induced tumour EC apoptosis, led to sufficient TICD8TC expansion and suppressed tumour growth in implanted tumours. On the other hand, when tumour ECs were resistant to cGAMP-induced apoptosis, the sparse tumour cell death led to failures of TICD8T expansion and suppression of tumour growth in spontaneous tumours (**Extended Data Fig. 12**).

AKT activation plays a central role in survival in several cell types, including ECs (Luo et al., *Cancer Cell*, 4(4), 257–262, 2003; Manning & Toker, *Cell*, 169(3), 381–405, 2017; Méndez-Pertuz et al., *Nature Communications*, 8(1), 1–16, 2017). Our analysis actually revealed that the tumour ECs of human breast ductal adenocarcinomas retain highly phosphorylated AKT compared with the ECs in adjacent normal tissue region. Nevertheless, our findings indicate that AKTi alone did not induce apoptosis in tumour ECs and whole tumour cells in spontaneous tumours. We hypothesized that AKT inhibition might enforce STING activation-induced apoptosis of tumour ECs in primary tumours based on our RNA-seq data and a previous report (Bieler et al., *Oncogene*, 26(39), 5722–5732, 2007). Indeed, AKTi treatment abrogated unresponsiveness to STING therapy, mediated through effective apoptosis of tumour ECs. Moreover, AKTi potentiated the cGAMP-induced anti-tumour effect, which partly could be mediated by enhancing the innate immune responses of tumour cells and bystander cells to cGAMP (**Extended Data Fig. 12**). Supporting this scenario, previous studies (Wu et al., *NATURE CELL BIOLOGY* |, 21, 2019) demonstrated that intracellular STING signalling and AKT pathways are antagonistically interacted for tumour progression mediated through innate immune responses, while AKTi increases T cell persistency and infiltration as well as early memory phenotype in tumour sites to enhance anti-tumour effect (Crompton et al., *Cancer Research*, 75(2), 296–305, 2015; Klebanoff et al., *JCI Insight*, 2(23), 2017).

In conclusion, our present findings unravel long-standing questions regarding the mechanism underlying STING activation-induced tumour EC apoptosis. Our results highlight the critical role of STING agonist as a VDA, which results in extensive tumour cell death releasing massive tumour antigen, supporting the activation of anti-tumour immunity. Based on these findings, we propose that combination treatment of STING agonist and AKT inhibitor should be considered for clinical human tumours, which could be refractory to single therapy with either agent alone.

References for responses to the comments of reviewer 1

Abram, C. L., Roberge, G. L., Hu, Y., & Lowell, C. A. (2014). Comparative analysis of the efficiency and specificity of myeloid-Cre deleting strains using ROSA-EYFP reporter mice. *Journal of Immunological Methods*. <https://doi.org/10.1016/j.jim.2014.05.009>

Bode, C., Fox, M., Tewary, P., Steinhagen, A., Hornung, V., & Steinhagen, F. (2016). Human plasmacytoid dendritic cells elicit a Type I Interferon response by sensing DNA via the cGAS-STING signaling pathway. *Eur. J. Immunol*, 46, 1615–1621.
<https://doi.org/10.1002/eji.201546113>

Ivashkiv, L. B. (2018). IFN γ : signalling, epigenetics and roles in immunity, metabolism, disease and cancer immunotherapy. *Nature Reviews Immunology*, 18(9), 545–558.
<https://doi.org/10.1038/s41577-018-0029->

Reviewer #2

Manuscript entitled “Refractoriness of STING Therapy Can be Oppressed by AKT Inhibitor Through Effective Tumour Vascular Disruption in Primary Tumour” by Jeong et al presented intriguing observations that intratumoral injection of STING agonist cGAMP, when in combination with the AKT1/2 inhibitor, induced a massive apoptosis of both tumor ECs and tumor cells and resulted in a remarkable suppression of primary tumor. Authors suggest that an interplay between TAM and ECs mediated by the TNFalpha-TNFR1 signaling axis is involved in the death of ECs and tumor vascular disruption. As a result, authors proposed a new therapeutic strategy for improving anti-tumor effects of primary tumors that were refractory to single STING agonist therapy. Overall, it is a manuscript focusing on a heatedly pursuing topic. The effects on combinational use of STING agonist and AKT inhibitor not only agreed with a recent report for an enhancement of cGAS-STING signaling by HER2-AKT inhibition (Chen et al, Nature Cell Biology, 2020; should be referred), but also set cGAMP as a tumor-specific VDA in primary tumor, by which it induced apoptosis of proliferative ECs subpopulations. The qualities of data are generally sound and support their claims. However, a few concerns prevent the acceptance of current manuscript.

We appreciate this favourable and encouraging comment. We additionally referred to the paper (Wu et al, Nature Cell Biology, 2019: HER2 recruits AKT1 to disrupt STING signalling and suppress antiviral defence and antitumour immunity) as reference 57 in the revised manuscript.

Major points:

1) Authors detected a reduction of live ECs as early as 3 h post cGAMP injection (such as Fig. 1e, 1h/1i). However, the levels of CD31 positive ECs were unchanged (Fig. 1g), or even increased in some observations (Fig. 1l and 2g). Very confusing.

This is a valid point that needs to be addressed. We replaced the images (Fig.1g, 0 h and Fig. 2g, PBS) with adequate representative images in the revised manuscript. Regarding Fig. 1l, CD31+ tumor vessels were usually dilated and disrupted at 24 h after i.t. cGAMP, so that they looked to constitute higher vascular density. However, in fact, they were reduced as quantitated by live ECs.

2) In Extended Data Fig. 2, the authors revealed an entire abrogation of both cGAMP-induced EC apoptosis and anti-tumor growth in STING KO mice. On the other hand, a combinational treatment of TNFalpha and IFNgamma significantly facilitated the EC apoptosis (Extended Data Fig. 5b, c). Did authors examine the combined effects of TNFalpha and IFNgamma on EC apoptosis in STING KO mice, in order to validate the TNFalpha-TNFR1 signaling axis interplayed

between TAM and ECs?

Following the reviewer's insightful comment, we additionally examined the combined effects of TNF α and IFN γ on the EC apoptosis in STING KO mice in order to validate the TNF α -TNFR1 signalling axis interplayed between TAMCs and tumour ECs. We included this data and its description into the revised manuscript as below.

In Results (p 7-8): Similar to those in WT mice, i.t. TNF α injection induced tumour EC apoptosis by 15.9%, and co-treatment with TNF α and IFN γ further increased tumour EC apoptosis to 43.4% in *STING* KO mice (**Extended Data Fig. 4c-e**). While i.t. IFN γ alone did not induce significant apoptosis in LLC tumour, i.t. TNF α induced whole tumour cell apoptosis by 11.9% in *STING* KO mice. However, co-treatment with TNF α and IFN γ induced whole tumour cell apoptosis by 37.6% in *STING* KO mice (**Extended Data Fig. 4c-e**). To further examine the role of TNF α in cGAMP-induced anti-tumour effects, we utilized anti-TNF α neutralizing antibody (Ab) (**Extended Data Fig. 4f**). Anti-TNF α neutralizing Ab diminished cGAMP-induced anti-tumour growth by 47%, and reduced apoptosis of tumour ECs and whole tumour cells by 55% and 48%, respectively (**Extended Data Fig. 4f-k**). In the *TNFR1* KO mice, cGAMP-induced anti-tumour growth was reduced by 64%, and apoptosis of tumour ECs and whole tumour cells was reduced by 65% and 66%, respectively (**Extended Data Fig. 5**). These findings imply that TNF α -TNFR1 activation in tumour bystander cells including tumour-associated macrophages, rather than in tumour cells and ECs, substantially contributed to the cGAMP-induced extensive tumour EC apoptosis and anti-tumour growth.

Extended Data Fig. 4. Activation of TNF α -TNFR1 following cGAMP-treatment plays a critical role in tumour EC Apoptosis and anti-tumour effect. c–e, LLC tumours in WT and *STING* KO mice were analysed 24 h after i.t. injection of TNF α (210 ng/70 μ l), IFN γ (350 ng/70 μ l) or combination of TNF α and IFN γ in 70 μ l. Representative images and comparisons of apoptosis in tumour ECs and whole tumour cells (whole cells). n = 5 mice/group from four independent experiments. Vertical bars indicate mean \pm SD.

3) A discussion should be amended: does AKTi potentiate the cGAMP-induced anti-tumor effects in spontaneous breast cancer by regulating the survival pathway of ECs and/or via enhancing the innate immune responses of tumor and bystander cells to cGAMP?

We appreciate this critical comment. We amended the Result and the Discussion in response to this comment and other reviewer's comments as below.

In Results (p 12): Thus, our results demonstrated that AKTi potentiated the cGAMP-induced anti-tumour effect by inducing extensive apoptosis of tumour ECs, enhancing immunogenic tumour cell apoptosis, and increasing recruitment of T1CD8TC in spontaneous breast cancer.

In Results (p 13): Thus, we found that AKTi also potentiated the cGAMP-induced anti-tumour effect in implanted tumours, by further inducing extensive apoptosis of tumour ECs and increasing recruitment of T1CD8TC into the tumour.

In Discussion (p 15): Nevertheless, our findings indicate that AKTi alone did not induce apoptosis in tumour ECs and whole tumour cells in spontaneous tumours. We hypothesized that AKT inhibition might enforce STING activation-induced apoptosis of tumour ECs in primary tumours based on our RNA-seq data and a previous report (Bieler et al., *Oncogene*, 26(39), 5722–5732, 2007). Indeed, AKTi treatment abrogated unresponsiveness to STING therapy, mediated through effective apoptosis of tumour ECs. Moreover, AKTi potentiated the cGAMP-induced anti-tumour effect, which partly could be mediated by enhancing the innate immune responses of tumour cells and bystander cells to cGAMP (Extended Data Fig. 12). Supporting this scenario, previous studies (Wu et al., *NATURE CELL BIOLOGY* |, 21, 2019) demonstrated that intracellular STING signalling and AKT pathways are antagonistically interacted for tumour progression mediated through innate immune responses, while AKTi increases T cell persistency and infiltration as well as early memory phenotype in tumour sites to enhance anti-tumour effect (Crompton et al., *Cancer Research*, 75(2), 296–305, 2015; Klebanoff et al., *JCI Insight*, 2(23), 2017).

4) Introduction and Discussion are over sketchy.

In response to reviewer's suggestion, we extensively modified Introduction and Discussion in the revised manuscript.

Introduction: Stimulator of interferon genes (STING), encoded by *TMEM173*, activates the innate immune system in response to cytosolic double-stranded DNA derived from viral or bacterial infection, chemical- or irradiation-induced cellular damage, and DNA leakage from the nucleus or mitochondria due to pathologic conditions, such as cancer (Ablasser et al., *Nature*, 503(7477), 530–534, 2013; Bakhroum et al., *Nature*, 553(7689), 467–472, 2018; Barber, *Nature Reviews Immunology*, 15(12), 760–770, 2015; Vanpouille-Box et al., *Cancer Cell*, 34(3), 361–

378, 2018). DNA sensor, cyclic GMP-AMP synthase (cGAS), produces STING-activating ligand, cyclic GMP-AMP (cGAMP), from cytosolic ATP and GMP (Sun et al., *Science*, 339(6121), 786–791, 2013). Upon activation via binding to cGAMP, STING is transferred from endoplasmic reticulum to Golgi apparatus, where it triggers Tank-binding kinase 1 followed by downstream signal activations including the nuclear factor- κ B pathway and type I interferon production (Deng et al., *Immunity*, 41(5), 843–852, 2014; Gaidt et al., *Cell*, 171(5), 1110–1124.e18, 2017). Enhanced interferon production induces cytotoxic effects directly against cancer cells, as well as activating dendritic cell (DC) maturation and promoting CD8⁺ T cell priming in tumour-draining lymph nodes (TDLN) (Qi Chen et al., In *Nature Immunology* (Vol. 17, Issue 10, pp. 1142–1149), 2016; Corrales et al., *Cell Reports*, 11(7), 1018–1030, 2015; Woo et al., *Immunity*, 41(5), 830–842, 2014). Throughout these actions, STING agonists shift anti-tumour immunity from immunologically silenced “cold” tumours to active “hot” tumours (Schadt et al., *Cell Reports*, 29(5), 1236–1248.e7, 2019). By doing so, they have been considered as synergistic agents for immune checkpoint inhibitors targeting PD-1 and PD-L1, which rescue exhausted T cells to kill tumour cells (Fu et al., *Science Translational Medicine*, 7(283), 2015; Marcus et al., *Immunity*, 49(4), 754–763.e4, 2018; Yang et al., *Journal of Clinical Investigation*, 129(10), 4350–4364, 2019). Although STING agonists has been tried for cancer therapy under tremendous attention, underlying mechanisms behind the anti-tumour effects are yet poorly understood (Reisländer et al., *Molecular Cell*, 80(1), 21–28, 2020; Won & Bakhoum, *Cancer Discovery*, 10(1), 26–39, 2020).

Murine-specific STING agonist, 5,6-dimethylxanthenone-4-acetic acid (DMXAA), which also has been known as a tumour vascular disrupting agent (VDA), exerts a potent anti-tumour effect that is caused by tumour endothelial cell (EC)-specific apoptosis and extensive haemorrhage within tumour (Roberts et al., *Journal of Experimental Medicine*, 204(7), 1559–1569, 2007). Tumour antigen release from dead tumour cells is the first step in adaptive immunity generation; therefore, tumour vascular destruction by STING agonists might be indispensable for achieving sufficient anti-tumour immunity (D. S. Chen & Mellman, *Immunity*, 39(1), 1–10, 2013; Zhang & Zhang, *Cellular and Molecular Immunology*, 17(8), 807–821, 2020)(Demaria et al., *Proceedings of the National Academy of Sciences of the United States of America*, 112(50), 15408–15413, 2015). Notably, TNF α has been proposed as a mediator of STING-induced tumour EC apoptosis (Daei Farshchi Adli et al., *Chemical Biology and Drug Design*, 91(5), 996–1006, 2018; Francica et al., *Cancer Immunology Research*, 6(4), 422–433, 2018). However, the source of TNF α upon STING activation remains unknown, and it is unclear why STING activation triggers apoptosis specifically to tumour ECs. Moreover, most previous studies have been conducted in subcutaneous tumour implantation models, which have limited ability to reflect human primary tumours (Chin et al., *Science*, 369(6506), 993–999, 2020; Fu et al., *Science Translational Medicine*, 7(283), 283ra52–283ra52, 2015; Sivick et al., *Cell Reports*, 25(11), 3074–3085.e5, 2018).

Activation of intracellular AKT signalling plays a critical role in survival in several cell types, including ECs (Luo et al., *Cancer Cell*, 4(4), 257–262, 2003; Manning & Toker, *Cell*, 169(3), 381–405, 2017; Méndez-Pertuz et al., *Nature Communications*, 8(1), 1–16, 2017). Of note, AKT is a major downstream molecule for conveying intracellular signalling of vascular growth factors and their receptors including angiotensin-1/Tie2 and VEGF-A/VEGFR2 (Augustin et al., *Nature Reviews Molecular Cell Biology*,

10(3), 165–177, 2009; Simons et al., *Nature Reviews Molecular Cell Biology*, 17(10), 611–625, 2016), which are key molecules in tumour angiogenesis. Accordingly, AKT pathway has been considered an attractive therapeutic target since AKT hyper-activation is associated with tumour aggressiveness and poor response to treatment. However, despite promising results in preclinical models, clinical trials of AKT inhibitors have failed to prove effectiveness, and none of the tested agents are currently used for cancer treatment (Hideshima et al., *Blood*, 107(10), 4053–4062, 2006; Martini et al., *Annals of Medicine*, 46(6), 372–383, 2014).

In the present study, we aimed to explore how STING activation initially affects tumour vasculatures implanted tumours, as well as in spontaneously growing breast cancer and melanoma. We additionally present a rationale for how co-treatment with an AKT inhibitor may help treat spontaneous primary tumours that are refractory to STING agonist therapy.

Discussion: In this study, we examined the poorly understood role of the STING agonist cGAMP as a tumour-specific VDA, and demonstrated that this early activity of cGAMP is a key for establishing anti-tumour effect. We also showed that refractoriness of i.t. STING therapy could be relieved by addition of AKTi through effective tumour EC apoptosis in spontaneous tumours (**Extended Data Fig. 12**).

One representative VDA, flavone acetic acid, selectively block vascular flow in the tumour core by promoting tumour EC apoptosis, leading to powerful anti-tumour effect in murine tumour models (Rivera & Bergers, *Science*, 349(6249), 694–695, 2015). Its derivatives have been developed from the trait of its chemical structure possessing amenable site with active region of xanthone-4-acetic acid, which gave rise to the development of DMXAA as potent as 16-fold (Tozer et al., *Nature Reviews Cancer*, 5(6), 423–435, 2005). Interestingly, the scheme of DMXAA on anti-tumour effect has changed from VDA to immunotherapeutic agent as a STING agonist (Conlon et al., *The Journal of Immunology*, 190(10), 5216–5225, 2013; Downey et al., *PLoS ONE*, 9(6), 10–12, 2014). Currently, i.t. STING agonist therapy is generally recognized to increase the production of type I interferon in tumours, which promotes CD8⁺ T cell priming and infiltration into the tumour, eventually converting immunologically “cold” tumours to “hot” tumours (Corrales et al., *Cell Reports*, 11(7), 1018–1030, 2015; Demaria et al., *Proceedings of the National Academy of Sciences of the United States of America*, 112(50), 15408–15413, 2015; Yang et al., *Journal of Clinical Investigation*, 129(10), 4350–4364, 2019). Therefore, STING therapy has been considered a promising treatment for use in combination with immune checkpoint inhibitors, since they are more effective in hot tumours than cold tumours (Huang et al., *Nature Communications*, 10(1), 1–15, 2019). However, throughout this study, we addressed the strong and rapid apoptotic role of STING agonist, cGAMP, in tumour ECs as a VDA. Through scRNA-seq and bulk RNA-seq analyses of the tumour ECs, we revealed that tip-like and proliferative EC subpopulations were vulnerable to STING activation-induced apoptosis. More importantly, we found that TAMCs-derived TNF α was a key mediator of the extensive apoptosis of tumour ECs (**Extended Data Fig. 12**).

STING plays multifaceted roles in tumour progression depending on target cells such as tumour cell or cells composing tumour microenvironment such as ECs, cancer-

associated fibroblasts, and immune cells (Bakhroum et al., *Nature*, 553(7689), 467–472, 2018; Qing Chen et al., *Nature*, 533(7604), 493–498, 2016; Gulen et al., *Nature Communications*, 8(1), 2017; Song et al., *Scientific Reports*, 7, 39858, 2017). For instance, Yang et al. (Yang et al., *Journal of Clinical Investigation*, 129(10), 4350–4364, 2019) have recently reported that STING agonists normalize tumour vessels mediated by up-regulating the genes related to type I/II interferon and vascular stabilization and enhancing pericyte coverage. However, it occurs at several days after i.t. injection with a prior and marked reduction of tumour ECs. In comparison, this study shows that STING agonist strongly and rapidly induces tumour EC apoptosis within a day after the injection. Moreover, our findings indicate that the massive tumour cell death, following extensive cGAMP-induced tumour EC apoptosis, led to sufficient TICD8TC expansion and suppressed tumour growth in implanted tumours. On the other hand, when tumour ECs were resistant to cGAMP-induced apoptosis, the sparse tumour cell death led to failures of TICD8T expansion and suppression of tumour growth in spontaneous tumours (**Extended Data Fig. 12**).

AKT activation plays a central role in survival in several cell types, including ECs (Luo et al., *Cancer Cell*, 4(4), 257–262, 2003; Manning & Toker, *Cell*, 169(3), 381–405, 2017; Méndez-Pertuz et al., *Nature Communications*, 8(1), 1–16, 2017). Our analysis actually revealed that the tumour ECs of human breast ductal adenocarcinomas retain highly phosphorylated AKT compared with the ECs in adjacent normal tissue region. Nevertheless, our findings indicate that AKTi alone did not induce apoptosis in tumour ECs and whole tumour cells in spontaneous tumours. We hypothesized that AKT inhibition might enforce STING activation-induced apoptosis of tumour ECs in primary tumours based on our RNA-seq data and a previous report (Bieler et al., *Oncogene*, 26(39), 5722–5732, 2007). Indeed, AKTi treatment abrogated unresponsiveness to STING therapy, mediated through effective apoptosis of tumour ECs. Moreover, AKTi potentiated the cGAMP-induced anti-tumour effect, which partly could be mediated by enhancing the innate immune responses of tumour cells and bystander cells to cGAMP (**Extended Data Fig. 12**). Supporting this scenario, previous studies (Wu et al., *NATURE CELL BIOLOGY* |, 21, 2019) demonstrated that intracellular STING signalling and AKT pathways are antagonistically interacted for tumour progression mediated through innate immune responses, while AKTi increases T cell persistency and infiltration as well as early memory phenotype in tumour sites to enhance anti-tumour effect (Crompton et al., *Cancer Research*, 75(2), 296–305, 2015; Klebanoff et al., *JCI Insight*, 2(23), 2017).

In conclusion, our present findings unravel long-standing questions regarding the mechanism underlying STING activation-induced tumour EC apoptosis. Our results highlight the critical role of STING agonist as a VDA, which results in extensive tumour cell death releasing massive tumour antigen, supporting the activation of anti-tumour immunity. Based on these findings, we propose that combination treatment of STING agonist and AKT inhibitor should be considered for clinical human tumours, which could be refractory to single therapy with either agent alone.

Minor points:

- 1) **Extended Data Fig. 1j, lacks annotations of PBS and cGAMP.**

We apologize for the missing annotations in Extended Data Fig. 1j. We included them in the revised Extended Data.

2) Extended Data Fig. 3d, lacks the statistical analysis.

We included the outcome of statistical analysis into Extended Data Fig. 3d (Fig. 3d in revised version).

3) The sequence of Extended Data Fig. 5a-c and 5d-f better to be exchanged.

We rearranged the Figures in the Extended Data as Extended Data Fig. 4 and 5 accordingly in the revised version.

4) An overlap of text in Fig. 5d.

The overlap was corrected.

5) Spelling errors: Line 23 of P9; Line 26 of P9.

The spelling errors were corrected.

Reviewer #3

In this study, Jeong et al. investigate the mechanism how STING agonists (cGAMP, DDXAA) disrupt tumor-associated blood vessels and reduce primary tumor growth in several mouse tumor models. Using STING and TNFR1 knockout mice (global ko and myeloid-specific ko in case of STING), they show that upon STING activation, tumor-associated myeloid cells release TNF- α that induces endothelial apoptosis within the tumor, reducing tumor growth in transplanted (LLC, B16, MMTV-PyMT-derived cell line) but not in a transgenic (MMTV-PyMT) tumor model. The resistance of MMTV-PyMT tumors towards STING agonists could be overcome using an AKT inhibitor.

This study is experimentally well performed, and the data is presented very clearly. My main criticism is that many findings presented here are not novel. It has been shown a long time before (even before the discovery of STING) that flavone acetic acid analogues such as DDXAA can act as vascular disrupting agents that trigger endothelial apoptosis and reduce primary tumor growth in multiple mouse tumor models, accompanied by a marked IFN response. It has also been shown that this involves macrophage-derived cytokines, including TNF- α (see Ref 27; Baguley, *Lancet Oncol* 2003, PMID 12623359; and references therein). The most significant new finding in this study is that a transgenic breast cancer model (MMTV-PyMT) is resistant to STING agonists, requiring additional AKT inhibition to induce endothelial apoptosis and tumor growth delay. However, this has been investigated in only one transgenic tumor model, and the clinical relevance of this observation is not clear.

We appreciate these insightful comments. We addressed more of our significant findings by including additional data and discussion as following.

Specific comments:

Major:

1-1). Are blood vessels equally resistant to STING agonists in other transgenic mouse cancer models?

We appreciate these critical comments, which are in line with the comment 2 by reviewer 4. To examine whether tumour vessels in other transgenic cancer model is resistant to STING agonist-induced tumour EC apoptosis, we additionally adopted a spontaneous melanoma model mouse, *Tyr-CreER^{T2};Braf^{CA};Pten^{loxP}* mouse (Dankort et al., *Nature Genetics*, 41(5), 544–552, 2009). *Tyr-CreER^{T2};Braf^{CA};Pten^{loxP}* mouse which turns on proto-oncogene *Braf* expression but turns off tumour suppressor gene *PTEN* expression in the tyrosinase containing melanocytes under the control of

estrogen receptor (can be activated by tamoxifen administration), which leads to development of melanoma. Consistent with the effects of cGAMP on the spontaneous breast cancer model, i.t. cGAMP treatment did not induce apoptosis of tumour ECs or whole tumour cells, although it slightly delayed tumour growth in the spontaneous melanoma model. However, combined treatment of i.t. cGAMP with i.p. AKTi induced extensive apoptosis in tumour ECs and whole tumour cells in the spontaneous melanoma model. We included additional data and description into the revised manuscript as below.

In Results (p 12-13): We next wondered whether co-treatment with cGAMP and AKTi would also be highly effective for suppressing tumour growth in another spontaneous tumour, we adopted *Tyr-CreER^{T2};Braf^{CA};Pten^{loxP}* spontaneous melanoma mice⁵¹ (Extended Data Fig. 10a). Combined treatment with i.t. cGAMP and i.p. AKTi induced extensive apoptosis in tumour ECs and whole tumour cells by 47% and 53%, respectively, while either cGAMP or AKTi alone did not induce notable apoptosis in spontaneous melanoma (Extended Data Fig. 10a–d). The combined treatment led to almost complete suppression of tumour growth by 91% in *Tyr-CreER^{T2};Braf^{CA};Pten^{loxP}* spontaneous melanoma mice. In contrast, treatment of these mice with i.t. cGAMP or i.p. AKTi alone suppressed tumour growth by only 44% or 10%, respectively (Extended Data Fig. 10e-g).

Extended Data Fig. 10. Combined treatment of cGAMP and AKTi induces tumour EC apoptosis and potent anti-tumour effect in spontaneous melanoma. a–d,

Diagram depicting tamoxifen swabbing, treatment and tumour sampling in *TyrCreER-T2;Braf-CA;Pten-loxP* spontaneous melanoma mice. Representative images of tumour vessels and apoptosis. White arrowheads indicate tumour EC apoptosis. Scale bars, 50 μ m. Comparisons of apoptosis in tumour ECs and whole tumour cells (whole cells). $n = 6$ mice/group from four independent experiments. Vertical bars indicate mean \pm SD. **e–g**, Diagram depicting tamoxifen swabbing and treatment schedule. Comparisons of tumour growths. $n = 7$ mice/group from four independent experiments. Plots and bars indicate mean \pm SD. Plot indicates each individual tumour growth. *P* values by Welch's one-way ANOVA test followed by Dunnett's T3 test (**c**, **d**, **f**). ns, not significant.

1-2) What is the clinical relevance of the finding that endothelial cells can only be forced into apoptosis using both STING agonists and AKT inhibitors in the MMTV-PyMT model?

Genetically engineered cancer models are obviously more clinically relevant than implantation models because they not only allow *de novo* tumour formation in a native immune-proficient microenvironment and faithfully recapitulate molecular and histopathological features of human cancers, but also have a strong predictive power for drug or immunotherapy responses and resistances (Gengenbacher et al., *Nature Reviews Cancer*, 2017; Kersten et al., *EMBO Molecular Medicine*, 2017). The resistances of tumour EC apoptosis to the cGAMP treatment in two primary cancer models used in this study could be in line with the failure of STING agonist (MK-1454) monotherapy for treating palpable solid tumours in phase I clinical trial (NCT03010176) (Harrington et al., *Annals of Oncology*, 29 (October), viii712, 2018).

1-3) Are there any indications that the AKT pathway is over-activated in the tumor-associated endothelium of (breast) cancer patients compared to normal tissue?

In order to address the comment regarding the AKT over-activation, we conducted additional experiment with a paraffin embedded, sectioned tissue array containing human breast ductal adenocarcinomas (31 cases of patients) to compare AKT activation by IHC for AKT phosphorylation at S473 at the ECs between breast adenocarcinoma regions and their adjacent normal tissue regions. Compared with ECs in the normal tissue region, the tumour ECs had highly increased AKT phosphorylation (Extended Data, Fig. 9). We included this data and its description into the revised manuscript.

In Results (p 11) **In fact, immunohistochemical analysis on the tissue array of human breast ductal adenocarcinomas revealed that tumour ECs in tumour core region had highly activated AKT compared with the ECs in adjacent normal tissue region (Extended Data, Fig. 9).**

In Methods (p 19): **To examine AKT activation in the tumour ECs, we purchased a paraffin embedded, sectioned tissue array containing human breast ductal**

adenocarcinomas and their adjacent normal tissues (31 cases of patients) (USBiomax, #OD-CT-RpBre03-004). The sections were deparaffinised, antigen retrieved, incubated with a primary antibody (phospho-AKT Ser473 antibody from Cell Signaling Technology), amplified the signal with the chromogen, counterstained, mounted for visualization according to manufacturer's instruction (Ventana BenchMark XT Staining Systems). The images were acquired using Axio Zoom.C16 (Carl Zeiss).

Extended Data Fig. 9. Tumour ECs of human breast carcinomas have stronger AKT phosphorylation at S473 (pAKT) compared with those of their adjacent normal tissues **a**, Representative images of pAKT in the ECs of sectioned breast adenocarcinoma tissues and their corresponding tumours from 4 patients (Pt.). Red arrows indicate blood vessels. Scale bars, 60 μ m. **b**, Comparison of pAKT ECs/total ECs. $n = 7$ /group from two independent experiments. P values by two-tailed t-test. Vertical bars indicate mean \pm SD.

2. Other primary breast cancer models (e.g. E0771 or 4T1) should be investigated to elucidate whether the difference in STING agonist responsiveness is due to the cancer type or due to the type of model (e.g. transplanted cell line vs. transgenic, spontaneous model).

Following the reviewer's indication, we evaluated the anti-tumour effect of cGAMP treatment on an orthotopic syngeneic breast cancer model using 4T1 cells. i.t. cGAMP-treatment induced extensive apoptosis of tumour ECs and whole tumour cells (48.2% and 38.4%, respectively), inhibited tumour growth by 81.4% compared

with i.t. PBS treatment (Revised Extended Data Fig. 6). Based on these findings, we concluded that the anti-tumour effects of STING agonist differ between implantation tumour models and spontaneous tumour models.

In Results (p 9): To ensure these findings, we generated another implanted breast tumour model through the inoculation of 4T1 breast carcinoma cells into the mammary fat pad (Extended Data Fig. 6a). Similar to those in implanted *MMTV-PyMT* cells breast tumour, i.t. cGAMP injection induced extensive apoptosis of tumour ECs (48%) and whole tumour cells (38%) and marked suppression of tumour growth by 81% in implanted 4T1 cells breast tumour (Extended Data Fig. 6b-f).

In Methods (p 17): To generate an orthotopic implanted 4T1 breast tumour model, 5×10^5 cells of 4T1 cells in 25 μ l of PBS were mixed with 25 μ l of Matrigel, and implanted to the mammary fat pad of 8 to 9-weeks-old female syngeneic Balb/c mice.

Extended Data Fig. 6. cGAMP induces tumour EC apoptosis and strong anti-tumour effect in 4T1 breast tumours. a–c, Diagram depicting generation of orthotopic implanted 4T1 breast tumours in Balb/c mice, i.t. PBS or cGAMP treatment, and sampling 24 h later. Representative images and comparisons of apoptosis in tumour ECs (white arrowheads) and whole tumour cells (whole cells). Scale bars, 1.0 mm (yellow bars) and 100 μ m (white bars). n = 6 mice/group from two independent experiments. Vertical bars indicate mean \pm SD. d–f, Diagram depicting treatment schedule in orthotopic implanted 4T1 breast tumour mice. Comparison of tumour growth. n = 7 mice/group from two independent experiments. Plots and bars indicate mean \pm SD. Plot indicates each individual tumour growth. P values by two-tailed t-test (c, e)

3. Is STING agonist/AKTi therapy responsiveness in the MMTV-PyMT model dependent on CD8+ T cells? Otherwise, the authors should reduce their statements in the results and discussion sections regarding anti-tumor immunity-promoting effects of this therapeutic approach.

This is valid point that should be addressed. To answer this question, it requires an additional, definitive experiment using CD8⁺ T cell depleted/MMTV-PyMT mice, which could be performed as a separate study in the future.

However, given that the combined treatment with i.t. cGAMP and i.p. AKTi increased CD8⁺ T cell infiltration by 75-fold, while treatment with i.t. cGAMP alone resulted in a 14-fold increase and i.p. AKTi alone yielded no change at 2 weeks after the treatment in the LLC tumour model (Revised **Fig. 10k, I**), we concluded that AKTi potentiates the cGAMP-induced anti-tumour effect by not only inducing extensive apoptosis of tumour ECs but also enhancing the innate immune responses of tumour and bystander cells to STING agonist. We modified the Results and Discussion in the revised manuscript as below.

In Results (p 12): Thus, our results demonstrated that AKTi potentiated the cGAMP-induced anti-tumour effect by inducing extensive apoptosis of tumour ECs, enhancing immunogenic tumour cell apoptosis, and increasing recruitment of TICD8TC in spontaneous breast cancer.

In Results (p 13): Thus, we found that AKTi also potentiated the cGAMP-induced anti-tumour effect in implanted tumours, by further inducing extensive apoptosis of tumour ECs and increasing recruitment of TICD8TC into the tumour.

In Discussion (p 15): Nevertheless, our findings indicate that AKTi alone did not induce apoptosis in tumour ECs and whole tumour cells in spontaneous tumours. We hypothesized that AKT inhibition might enforce STING activation-induced apoptosis of tumour ECs in primary tumours based on our RNA-seq data and a previous report⁴³. Indeed, AKTi treatment abrogated unresponsiveness to STING therapy, mediated through effective apoptosis of tumour ECs. Moreover, AKTi potentiated the cGAMP-induced anti-tumour effect, which partly could be mediated by enhancing the innate immune responses of tumour cells and bystander cells to cGAMP (**Extended Data Fig. 12**). Supporting this scenario, previous studies⁵⁷ demonstrated that intracellular STING signalling and AKT pathways are antagonistically interacted for tumour progression mediated through innate immune responses, while AKTi increases T cell persistency and infiltration as well as early memory phenotype in tumour sites to enhance anti-tumour effect^{58,59}.

4. “Proliferative” ECs are present in the MMTV-PyMT, yet this model does not respond to STING agonists alone. Thus, the conclusion that STING agonists predominantly target “tip-like” and “proliferative” ECs is not well founded but the data, and should be toned down. Furthermore, it would be interesting which type of vessels preferentially resist the treatment in the sensitive models. Are those predominantly large, mature (pericyte-covered) vessels as it was described for some anti-angiogenic therapies?

We appreciate this critical point. We re-evaluated the original data (original Fig. 5), performed additional scRNA-seq on the ECs of spontaneous and implanted breast tumours treated with PBS or cGAMP, analysed them accordingly, and incorporated them and their descriptions into the revised manuscript as below.

Arterial and stalk ECs are covered with vascular smooth muscle cells and pericytes, while tip and proliferative ECs are lack of pericyte coverage in growing organs (Carmeliet & Jain, *Nature*, 2011; De Palma *et al.*, 2017). In solid tumour, the ECs lack or have loose pericyte coverage in the intratumoural region, while they are covered with pericytes in a diverse manner in the peritumoral region. Moreover, the proportion of pericyte coverage around EC is higher in the spontaneous than implanted breast tumour. Furthermore, our scRNA-seq analysis revealed that the arterial and stalk-like ECs were relatively well preserved in the cGAMP-treated spontaneous and implanted breast tumours. Therefore, we could conclude that the arterial and stalk-like ECs were protected from cGAMP-induced apoptosis in both tumours. We also included a part of these descriptions into the revised text accordingly as below.

In Results (p 10-11): **Tip-like and proliferative ECs are vulnerable to STING activation-induced apoptosis, but less sensitive to spontaneous tumour versus to implanted tumour** To gain a further insight on the response of tumour ECs to STING agonist, we performed scRNA-seq on the ECs of *MMTV-PyMT* spontaneous tumour and its orthotopic implanted breast tumour, which were sampled at 3 h after i.t. PBS or cGAMP (**Fig. 8a**). Compared with those treated with PBS, cGAMP highly increased expressions of the genes related to immune activation and stimulation of type I interferon pathway such as *CCL5*, *CXCL9*, *CXCL10*, *IRF7*, *IRGM1*, *ISG15*, *OASL2* and *PHF11D* in both tumour ECs (**Fig. 8b, c**). Five distinct EC clusters- stalk-like, tip-like, proliferative, arterial and AQP7⁺ ECs were present in the pooled tumour ECs treated with PBS (**Fig. 8d, e** and **Extended Data Fig. 8a, b**). AQP7⁺ ECs are a breast-specific, fully-differentiated subpopulation that highly expresses AQP7, CD35 and FABP5 mRNAs and actively participates in the glycerol and fatty acid metabolism and transport⁴⁴. Distinctiveness of 5 clustering was largely blunted in the pooled ECs treated with cGAMP (**Fig. 8d, e**), implying that tumour ECs were largely affected by i.t. STING agonist. We noted that cGAMP treatment markedly reduced the population of proliferative ECs in spontaneous tumour, while it markedly reduced both proliferative and tip-like ECs in implanted tumour (**Fig 8f, g**). The latter findings are similar to those in implanted LLC tumours (**Fig. 4f-j**). Gene ontology enrichment analysis on the ECs of implanted tumour *versus* spontaneous tumour revealed differential transcriptional responses to both PBS and cGAMP treatment (**Extended Data Fig. 8c**). Of note, the genes related to apoptosis were enriched in the tip-like ECs of the implanted tumours treated with cGAMP (red underline in **Extended Data Fig. 8c**), implying that the tip-like ECs are vulnerable to STING agonist. Because there was a lack of proliferative ECs mainly due to apoptotic death in the cGAMP-treated tumours (**Extended Data Fig. 8c**), we could not analyse the character of the proliferative ECs. These findings denote that tip-like and proliferative ECs are vulnerable to STING activation-induced apoptosis, and these can partly explain why only the ECs of implanted, but not spontaneous, breast cancer underwent apoptosis

following cGAMP treatment. Moreover, the ECs retaining maturation gene profiles within tumours such as arterial, stalk-like and AQP7⁺ ECs (Fig. 8d, e and Extended Data Fig. 8a, b) are likely to be resistant to STING activation-induced apoptosis.

Fig. 8. Implanted, but not spontaneous, breast tumour ECs are vulnerable to STING activation-induced apoptosis. **a**, Schematic diagram for depicting treatment, sampling, and scRNA-seq of tumour ECs in MMTV-PyMT spontaneous and their implanted breast tumours. n = 2 mice/each group. **b**, **c**, Violin plots depicting the normalized expression levels of top-ranked differentially expressed genes in spontaneous and implanted tumour ECs. **d**, UMAP plots comparing five clusters of tumour ECs by unsupervised clustering, integrating datasets in spontaneous and implanted tumour ECs between PBS- and cGAMP- treatment. **e**, The origin of datasets in each cluster. Each dot represents single cell. **f**, **g**, Compositional differences of

tumour EC subpopulations by treatments in spontaneous and implanted tumour ECs.

Minor:

1. Activated STING is known to induce a type-I interferon response, yet the authors chose to study the expression levels and function of type-II interferon (gamma) in their models. Why was that? Furthermore, the authors' finding that IFN-gamma could not induce vascular regression (Fig. S5a-c) is somewhat in contrast to a previous publication by Kammertoens et al (Nature 2017, PMID 28445461). This should be discussed at least.

In response to this comment and the comment 2 of reviewer 1, we additionally measured IFN β in the tumour and included the data as revised Figure 5d. The reason why we measured IFN γ was that IFN γ has been reported to destruct tumour vessels synergistically with TNF α (Huyghe *et al.*, *EMBO Molecular Medicine*, 2020; Johansson *et al.*, *Proceedings of the National Academy of Sciences*, 2012; Verhoef *et al.*, *Current Treatment Options in Oncology*, 2007). In addition, combined infusion of IFN γ , TNF α and melphalan into the limb artery has been able to attain effective regression of sarcoma in the patients (Verhoef *et al.*, *Curr. Treat. Options Oncol.* 2007)

The paper by Kammertoens *et al.* (Nature 2017) demonstrated the regressive role of IFN γ in tumour vessels through a non-apoptotic action using the genetically modified mice. In fact, the IFN γ -induced vascular regression was observed at 4-5 days after activation of IFN γ signalling. In contrast, they found that only minimal vascular regression occurred at 24 hr after the IFN γ induction. Although they also found vascular remodelling (enlarged and leaky tumour vessels) at 24 h after TNF α induction, they did not find significant apoptosis in the tumour ECs at this time point.

In comparison, we sought to determine whether i.t. IFN γ induces tumour EC apoptosis at early time point, that was, 24 hr after the treatment. Although no significant effect of i.t. IFN γ on tumour EC apoptosis was observed, we found synergistic effect of i.t. IFN γ and TNF α on tumour EC apoptosis. Therefore, we believe these two findings are complementary rather than contradictory.

2. The sequencing data (single cell and bulk) generated in this study should be uploaded to a repository and need to be available to the reviewers and the public.

We uploaded the sequencing data on Gene Expression Omnibus as accession numbers for bulk RNA-seq data (GSE159203) and scRNA-seq data (GSE159013). We also included this information into the revised manuscript.

Data availability Single-cell RNA sequencing data and bulk RNA-sequencing data are available in National Center for Biotechnology Information's Gene Expression

Omnibus under accession number GSE159013 (scRNA-seq on tumour ECs of LLC tumour), GSE159203 (bulk RNA-seq on ECs from normal mammary pad and breast tumours), and GSE171451 (scRNA-seq on tumour ECs from *MMTV-PyMT* spontaneous breast tumours and implanted breast tumours). Further information and requests for resources and reagents should be directed to and will be fulfilled by Gou Young Koh (gykoh@kaist.ac.kr).

3. Why was the transplanted MMTV-PyMT cell line tumor model treated so often (Figs. 4k-m)? Would treatment twice (as was done with the other transplantable models) sufficient to achieve the same effects?

The reason was because we designed them to compare the effect of cGAMP on the tumour EC apoptosis and anti-tumour effect between spontaneous and implanted breast tumour models under the same treatment regimen when the tumour volumes reached similar sizes (9 weeks after birth in spontaneous tumour *versus* 10 days after implantation). During the initial setting, because two i.t injections of cGAMP did not induce an apparent anti-tumour effect in the spontaneous tumour, we optimized the treatment by increasing the frequency of the injection.

4. The volume used for intratumoral injections (70 µl) appears to be huge. Such an injection would likely lead to a massive pressure increase in the tumor tissue, which could lead to confounding effects. How did the authors avoid backflow from the injection site? Please comment.

Thank you for this constructive comment. We included the description into the revised Methods as below

In Methods (p. 17): To avoid confounding effect caused by i.t. injection *per se*, the same volume (70 µl) of PBS was injected into the tumour of control group. To prevent the backflow leakage, i.t. injection was performed slowly over 1 min while holding the injection site with smooth forceps, followed by gentle manual compression for 15 sec.

5. Fig. 1E: The dotted outline of the blood vessels is very unclear and seems somewhat arbitrary. Please provide better / additional images to demonstrate how the vessel outlines were identified.

We changed the region in order to get better images to demonstrate how the vessel outlines were identified in the revised Figure 1e.

6. Fig. 2E: There is one gene name missing in this graph (there is one more row in the heat map than gene names). Furthermore, the color scheme (red – violet) is somewhat unfortunate, as it makes a poor contrast between highly and lowly

expressed genes. Please consider changing to a red-blue color scheme as in Fig. 5.

Thank you for pointing out the error and for the constructive comment. We corrected the error and changed the colour for better contrast in the revised Figure 4d.

7. Some reference seem to be unsuitable to support the sentences that cite them, e.g. reference 10 and 28. Please find better or additional references or rephrase the corresponding statements.

We re-checked all references and edited them accordingly in the revised manuscript.

8. The LysM-Cre mouse does not specifically target (tumor-associated) macrophages, but many more myeloid cell types, including monocytes and granulocytes. Please rephrase the corresponding statements in the results.

We rephrased “tumour-associated macrophages (TAM)” to “tumour-associated myeloid cells (TAMCs)” in the revised manuscript.

9. CD8+ T cells should not be called “cytotoxic” unless expression of cytotoxic markers (e.g. granzyme) has been shown.

Following the reviewer’s comment, we rephrased “tumour-infiltrating cytotoxic T cells (TICT)” to “tumour-infiltrating CD8⁺ T cells (TICD8TC)” in the revised manuscript.

10. I would suggest to call the MMTV-PyMT tumors “transgenic” or “spontaneous” instead of “primary” to avoid confusion (also a transplanted tumor could be called “primary”).

Thank you for this constructive comment. To avoid confusion, we changed “primary” to “spontaneous” in the revised manuscript.

11. The AKT – mTOR pathway might not only protect endothelial cells from apoptosis, but could also directly interfere with STING activation in macrophages, e.g. via ULK1/2. Please discuss.

We appreciate this constructive comment. AKT inhibitor can prevent mTOR phosphorylation, which leads to hyper-activation of STING pathway to promote anti-tumour immunity (Moretti et al., *Cell*, 171(4), 809-823.e13, 2017). As responded to comment 3, following reviewer’s this comment, we revised the Discussion as above

(please see the response to major comment 3).

12. The discussion chapter is very short and too focused on the role of tumor immunity, which has not been demonstrated to be relevant for the treatment effects observed by the authors (see major comment 3). I would suggest a major revision of this chapter, including previous studies regarding the anti-tumoral effects of DDXAA and other flavone acetic acid derivatives.

We extensively revised the Discussion by including relevant discussions as below.

Discussion: In this study, we examined the poorly understood role of the STING agonist cGAMP as a tumour-specific VDA, and demonstrated that this early activity of cGAMP is a key for establishing anti-tumour effect. We also showed that refractoriness of i.t. STING therapy could be relieved by addition of AKTi through effective tumour EC apoptosis in spontaneous tumours (**Extended Data Fig. 12**).

One representative VDA, flavone acetic acid, selectively block vascular flow in the tumour core by promoting tumour EC apoptosis, leading to powerful anti-tumour effect in murine tumour models (Rivera & Bergers, *Science*, 349(6249), 694–695, 2015). Its derivatives have been developed from the trait of its chemical structure possessing amenable site with active region of xanthone-4-acetic acid, which gave rise to the development of DMXAA as potent as 16-fold (Tozer et al., *Nature Reviews Cancer*, 5(6), 423–435, 2005). Interestingly, the scheme of DMXAA on anti-tumour effect has changed from VDA to immunotherapeutic agent as a STING agonist (Conlon et al., *The Journal of Immunology*, 190(10), 5216–5225, 2013; Downey et al., *PLoS ONE*, 9(6), 10–12, 2014). Currently, i.t. STING agonist therapy is generally recognized to increase the production of type I interferon in tumours, which promotes CD8⁺ T cell priming and infiltration into the tumour, eventually converting immunologically “cold” tumours to “hot” tumours (Corrales et al., *Cell Reports*, 11(7), 1018–1030, 2015; Demaria et al., *Proceedings of the National Academy of Sciences of the United States of America*, 112(50), 15408–15413, 2015; Yang et al., *Journal of Clinical Investigation*, 129(10), 4350–4364, 2019). Therefore, STING therapy has been considered a promising treatment for use in combination with immune checkpoint inhibitors, since they are more effective in hot tumours than cold tumours (Huang et al., *Nature Communications*, 10(1), 1–15, 2019). However, throughout this study, we addressed the strong and rapid apoptotic role of STING agonist, cGAMP, in tumour ECs as a VDA. Through scRNA-seq and bulk RNA-seq analyses of the tumour ECs, we revealed that tip-like and proliferative EC subpopulations were vulnerable to STING activation-induced apoptosis. More importantly, we found that TAMCs-derived TNF α was a key mediator of the extensive apoptosis of tumour ECs (**Extended Data Fig. 12**).

STING plays multifaceted roles in tumour progression depending on target cells such as tumour cell or cells composing tumour microenvironment such as ECs, cancer-associated fibroblasts, and immune cells (Bakhom et al., *Nature*, 553(7689), 467–472, 2018; Qing Chen et al., *Nature*, 533(7604), 493–498, 2016; Gulen et al., *Nature Communications*, 8(1), 2017; Song et al., *Scientific Reports*, 7, 39858, 2017). For instance, Yang et al. (Yang et al., *Journal of Clinical Investigation*, 129(10), 4350–

4364, 2019) have recently reported that STING agonists normalize tumour vessels mediated by up-regulating the genes related to type I/II interferon and vascular stabilization and enhancing pericyte coverage. However, it occurs at several days after i.t. injection with a prior and marked reduction of tumour ECs. In comparison, this study shows that STING agonist strongly and rapidly induces tumour EC apoptosis within a day after the injection. Moreover, our findings indicate that the massive tumour cell death, following extensive cGAMP-induced tumour EC apoptosis, led to sufficient TICD8TC expansion and suppressed tumour growth in implanted tumours. On the other hand, when tumour ECs were resistant to cGAMP-induced apoptosis, the sparse tumour cell death led to failures of TICD8T expansion and suppression of tumour growth in spontaneous tumours (**Extended Data Fig. 12**).

AKT activation plays a central role in survival in several cell types, including ECs (Luo et al., *Cancer Cell*, 4(4), 257–262, 2003; Manning & Toker, *Cell*, 169(3), 381–405, 2017; Méndez-Pertuz et al., *Nature Communications*, 8(1), 1–16, 2017). Our analysis actually revealed that the tumour ECs of human breast ductal adenocarcinomas retain highly phosphorylated AKT compared with the ECs in adjacent normal tissue region. Nevertheless, our findings indicate that AKTi alone did not induce apoptosis in tumour ECs and whole tumour cells in spontaneous tumours. We hypothesized that AKT inhibition might enforce STING activation-induced apoptosis of tumour ECs in primary tumours based on our RNA-seq data and a previous report (Bieler et al., *Oncogene*, 26(39), 5722–5732, 2007). Indeed, AKTi treatment abrogated unresponsiveness to STING therapy, mediated through effective apoptosis of tumour ECs. Moreover, AKTi potentiated the cGAMP-induced anti-tumour effect, which partly could be mediated by enhancing the innate immune responses of tumour cells and bystander cells to cGAMP (**Extended Data Fig. 12**). Supporting this scenario, previous studies (Wu et al., *NATURE CELL BIOLOGY* |, 21, 2019) demonstrated that intracellular STING signalling and AKT pathways are antagonistically interacted for tumour progression mediated through innate immune responses, while AKTi increases T cell persistency and infiltration as well as early memory phenotype in tumour sites to enhance anti-tumour effect (Crompton et al., *Cancer Research*, 75(2), 296–305, 2015; Klebanoff et al., *JCI Insight*, 2(23), 2017).

In conclusion, our present findings unravel long-standing questions regarding the mechanism underlying STING activation-induced tumour EC apoptosis. Our results highlight the critical role of STING agonist as a VDA, which results in extensive tumour cell death releasing massive tumour antigen, supporting the activation of anti-tumour immunity. Based on these findings, we propose that combination treatment of STING agonist and AKT inhibitor should be considered for clinical human tumours, which could be refractory to single therapy with either agent alone.

References for responses to the comments of reviewer 3

Carmeliet, P., & Jain, R. K. (2011). Molecular mechanisms and clinical applications of angiogenesis. *Nature*, 473(7347), 298–307. <https://doi.org/10.1038/nature10144>

Dankort, D., Curley, D. P., Cartlidge, R. A., Nelson, B., Karnezis, A. N., Damsky, W. E., You, M. J., DePinho, R. A., McMahon, M., & Bosenberg, M. (2009). BrafV600E cooperates with Pten loss to induce metastatic melanoma. *Nature Genetics*, 41(5), 544–552.

<https://doi.org/10.1038/ng.356>

De Palma, M., Biziato, D., & Petrova, T. V. (2017). Microenvironmental regulation of tumour angiogenesis. <https://doi.org/10.1038/nrc.2017.51>

Gengenbacher, N., Singhal, M., & Augustin, H. G. (2017). Preclinical mouse solid tumour models: Status quo, challenges and perspectives. *Nature Reviews Cancer*, 17(12), 751–765. <https://doi.org/10.1038/nrc.2017.92>

Harrington, K. J., Brody, J., Ingham, M., Strauss, J., Cemerski, S., Wang, M., Tse, A., Khilnani, A., Marabelle, A., & Golan, T. (2018). Preliminary results of the first-in-human (FIH) study of MK-1454, an agonist of stimulator of interferon genes (STING), as monotherapy or in combination with pembrolizumab (pembro) in patients with advanced solid tumors or lymphomas. *Annals of Oncology*, 29(October), viii712. <https://doi.org/10.1093/annonc/mdy424.015>

Huyghe, L., Van Parys, A., Cauwels, A., Van Lint, S., De Munter, S., Bultinck, J., Zabeau, L., Hostens, J., Goethals, A., Vanderroost, N., Verhee, A., Uzé, G., Kley, N., Peelman, F., Vandekerckhove, B., Brouckaert, P., & Tavernier, J. (2020). Safe eradication of large established tumors using neovasculature-targeted tumor necrosis factor-based therapies. *EMBO Molecular Medicine*, 12(2), e11223. <https://doi.org/10.15252/emmm.201911223>

Johansson, A., Hamzah, J., Payne, C. J., & Ganss, R. (2012). Tumor-targeted TNF α stabilizes tumor vessels and enhances active immunotherapy. *Proceedings of the National Academy of Sciences*, 109(20), 7841–7846. <https://doi.org/10.1073/PNAS.1118296109>

Kersten, K., Visser, K. E., Miltenburg, M. H., & Jonkers, J. (2017). Genetically engineered mouse models in oncology research and cancer medicine. *EMBO Molecular Medicine*, 9(2), 137–153. <https://doi.org/10.15252/emmm.201606857>

Moretti, J., Roy, S., Bozec, D., Martinez, J., Chapman, J. R., Ueberheide, B., Lamming, D. W., Chen, Z. J., Horng, T., Yeretssian, G., Green, D. R., & Blander, J. M. (2017). STING Senses Microbial Viability to Orchestrate Stress-Mediated Autophagy of the Endoplasmic Reticulum. *Cell*, 171(4), 809-823.e13. <https://doi.org/10.1016/j.cell.2017.09.034>

Verhoef, C., Wilt, J. H. W., Grünhagen, D. J., Geel, A. N., Hagen, T. L. M., & Eggermont, A. M. M. (2007). Isolated limb perfusion with melphalan and TNF- α in the treatment of extremity sarcoma. *Current Treatment Options in Oncology*, 8(6), 417–427. <https://doi.org/10.1007/s11864-007-0044-y>

Reviewer #4

General: In this manuscript, the authors use a range of tumor mouse models to determine the antitumor effects of STING agonists. They show that intratumoral injection of cGAMP, a STING agonist, induces apoptosis of tumor endothelial cells in subcutaneous xenograft models of LLC- and B16-induced tumors, but not in primary breast tumors. The authors then show that apoptosis in implanted tumors is stimulated by TNFalpha-TNFR1 signaling, while in primary tumors, additional inhibition of AKT is required for efficient induction of apoptosis via this pathway. These findings have implications for treatment with STING agonists, suggesting that combination with AKT inhibitors might represent a more efficient strategy. As such, the manuscript presents potentially important data, but in its current form, it suffers from a number of shortcomings that will need to be addressed, as highlighted below.

We appreciate this favourable and encouraging comment.

Major comments:

1. The authors claim in the introduction: “Moreover, most previous studies have been conducted in subcutaneous tumour implantation models, which have limited ability to reflect human primary tumours.” However, this reviewer feels that also this study was performed to a large extent in subcutaneous tumor implantation model. Crucial experiments, such as the single cell analysis, were done in subcutaneous LLC. Why was it not (also) performed in a primary tumor model? The paper would benefit from switching the focus from the LLC model to primary tumor models.

We appreciate this critical and constructive comment. Following the reviewer’s recommendation, we newly performed scRNA-seq on tumor ECs of spontaneous primary tumour (*MMTV-PyMT* tumour) in order to clarify and focus on the key roles of STING agonist in the ECs of primary tumour. Indeed, this additional information helps understand the significance of STING agonist on the ECs of spontaneous tumour. We included these additional data and their description accordingly into the revised manuscript.

In Results (p 10-11): **Tip-like and proliferative ECs are vulnerable to STING activation-induced apoptosis, but less sensitive to spontaneous tumour versus to implanted tumour** To gain a further insight on the response of tumour ECs to STING agonist, we performed scRNA-seq on the ECs of *MMTV-PyMT* spontaneous tumour and its orthotopic implanted breast tumour, which were sampled at 3 h after i.t PBS or cGAMP (**Fig. 8a**). Compared with those treated with PBS, cGAMP highly increased expressions of the genes related to immune activation and stimulation of type I interferon pathway such as *CCL5*, *CXCL9*, *CXCL10*, *IRF7*, *IRGM1*, *ISG15*, *OASL2* and *PHF11D* in both tumour ECs (**Fig. 8b, c**). Five distinct EC clusters- stalk-like, tip-like, proliferative, arterial and AQP7⁺ ECs were present in the pooled tumour ECs treated with PBS (**Fig.**

8d, e and Extended Data Fig. 8a, b). AQP7⁺ ECs are a breast-specific, fully-differentiated subpopulation that highly expresses AQP7, CD35 and FABP5 mRNAs and actively participates in the glycerol and fatty acid metabolism and transport (Kalucka et al., *Cell*, 180(4), 764-779.e20, 2020). Distinctiveness of 5 clustering was largely blunted in the pooled ECs treated with cGAMP (**Fig. 8d, e**), implying that tumour ECs were largely affected by i.t. STING agonist. We noted that cGAMP treatment markedly reduced the population of proliferative ECs in spontaneous tumour, while it markedly reduced both proliferative and tip-like ECs in implanted tumour (**Fig 8f, g**). The latter findings are similar to those in implanted LLC tumours (**Fig. 4f-j**). Gene ontology enrichment analysis on the ECs of implanted tumour *versus* spontaneous tumour revealed differential transcriptional responses to both PBS and cGAMP treatment (**Extended Data Fig. 8c**). Of note, the genes related to apoptosis were enriched in the tip-like ECs of the implanted tumours treated with cGAMP (red underline in **Extended Data Fig. 8c**), implying that the tip-like ECs are vulnerable to STING agonist. Because there was a lack of proliferative ECs mainly due to apoptotic death in the cGAMP-treated tumours (**Extended Data Fig. 8c**), we could not analyse the character of the proliferative ECs.

Fig. 8. Implanted, but not spontaneous, breast tumour ECs are vulnerable to STING activation-induced apoptosis. **a**, Schematic diagram for depicting treatment, sampling, and scRNA-seq of tumour ECs in *MMTV-PyMT* spontaneous and their implanted breast tumours. $n = 2$ mice/each group. **b, c**, Violin plots depicting the normalized expression levels of top-ranked differentially expressed genes in spontaneous and implanted tumour ECs. **d**, UMAP plots comparing five clusters of tumour ECs by unsupervised clustering, integrating datasets in spontaneous and implanted tumour ECs between PBS- and cGAMP- treatment. **e**, The origin of datasets in each cluster. Each dot represents single cell. **f, g**, Compositional differences of tumour EC subpopulations by treatments in spontaneous and implanted tumour ECs.

In Results (p 11): These findings denote that tip-like and proliferative ECs are vulnerable to STING activation-induced apoptosis, and these can partly explain why only the ECs of implanted, but not spontaneous, breast cancer underwent apoptosis following cGAMP treatment. Moreover, the ECs retaining maturation gene profiles within tumours such as arterial, stalk-like and AQP7⁺ ECs (Fig. 8d, e and Extended Data Fig. 8a, b) are likely to be resistant to STING activation-induced apoptosis.

2. The authors document a difference in implanted versus spontaneous tumors in MMTV-PyMT model. This is a very interesting point, which could be important for interpretation of results from animal studies. Indeed, the authors present this as the crucial point of their study, and it is thus warranted to confirm these results in another spontaneous model, for example in one of the spontaneous lung tumor models. This would be particularly appropriate given the primary focus on the LLC model, and the fact that spontaneous MMTV-PyMT tumors show large heterogeneity in the data and a reduced growth rate compared to corresponding implanted tumors (compare Fig. 4e and 4i).

We appreciate this constructive comment, which is in line with the comment 1 of the reviewer 3. In response to this comment, we adopted another spontaneous tumour model - *Tyr-CreER^{T2};Braf^{CA};Pten^{loxP}* spontaneous melanoma mouse model (Dankort et al., *Nature Genetics*, 41(5), 544–552, 2009) instead of spontaneous lung tumour model because orthotopic lung model is not available and it is inaccessible for intratumoural injection of cGAMP. Tamoxifen inducible *Tyr-CreER^{T2}* turns on proto-oncogene *Braf* expression while turn off tumour suppressor gene *PTEN* expression in the tyrosinase containing melanocytes under control of estrogen receptor (Moynihan et al., *Nature Medicine*, 2016; Riedel et al., *Nature Immunology*, 2016), which leads to develop melanoma.

Consistent with the effects of cGAMP on the spontaneous breast cancer model, i.t. cGAMP treatment did not induce apoptosis in tumour ECs or whole tumour cells, although it slightly delayed tumour growth in the spontaneous melanoma model. However, combined treatment of i.t. cGAMP with i.p. AKTi induced extensive apoptosis in tumour ECs and whole tumour cells in the spontaneous melanoma

model. We included additional data and description into the revised manuscript as below.

In Results (page 12-13): We next wondered whether co-treatment with cGAMP and AKTi would also be highly effective for suppressing tumour growth in another spontaneous tumour, we adopted *Tyr-CreER^{T2};Braf^{CA};Pten^{loxP}* spontaneous melanoma mice ⁵¹ (Extended Data Fig. 10a). Combined treatment with i.t. cGAMP and i.p. AKTi induced extensive apoptosis in tumour ECs and whole tumour cells by 47% and 53%, respectively, while either cGAMP or AKTi alone did not induce notable apoptosis in spontaneous melanoma (Extended Data Fig. 10a–d). The combined treatment led to almost complete suppression of tumour growth by 91% in *Tyr-CreER^{T2};Braf^{CA};Pten^{loxP}* spontaneous melanoma mice. In contrast, treatment of these mice with i.t. cGAMP or i.p. AKTi alone suppressed tumour growth by only 44% or 10%, respectively (Extended Data Fig. 10e-g).

Extended Data Fig. 10. Combined treatment of cGAMP and AKTi induces tumour EC apoptosis and potent anti-tumour effect in spontaneous melanoma. a–d, Diagram depicting tamoxifen swabbing, treatment and tumour sampling in *TyrCreER-T2;Braf-CA;Pten-loxP* spontaneous melanoma mice. Representative images of tumour vessels and apoptosis. White arrowheads indicate tumour EC apoptosis. Scale bars, 50 μ m. Comparisons of apoptosis in tumour ECs and whole tumour cells (whole cells). n = 6 mice/group from four independent experiments. Vertical bars indicate mean \pm SD. **e–g,** Diagram depicting tamoxifen swabbing and treatment schedule. Comparisons of tumour growths. n = 7 mice/group from four independent experiments.

Plots and bars indicate mean \pm SD. Plot indicates each individual tumour growth. *P* values by Welch's one-way ANOVA test followed by Dunnett's T3 test (**c**, **d**, **f**). ns, not significant.

3. Generally speaking, the abstract, introduction and discussion sections require work. Currently, these three sections largely repeat each other. Discussion is limited to a summary of results, and the authors should instead discuss their findings in a context of the already published literature and how their results relate to a therapy of human patients. Tumors in patients are always spontaneous and can hardly be treated by intratumor injections. Furthermore, authors should discuss if similar differences between ECs in spontaneous and implanted tumors have ever been observed for other pathways that do not involve STING?

Thank you for this constructive comment, which is in line with the comments of other reviewers. Following the reviewer's suggestions and indications, we extensively revised the Abstract, Introduction and Discussion as below.

Introduction: Stimulator of interferon genes (STING), encoded by *TMEM173*, activates the innate immune system in response to cytosolic double-stranded DNA derived from viral or bacterial infection, chemical- or irradiation-induced cellular damage, and DNA leakage from the nucleus or mitochondria due to pathologic conditions, such as cancer¹⁻⁴. DNA sensor, cyclic GMP-AMP synthase (cGAS), produces STING-activating ligand, cyclic GMP-AMP (cGAMP), from cytosolic ATP and GMP⁵. Upon activation via binding to cGAMP, STING is transferred from endoplasmic reticulum to Golgi apparatus, where it triggers Tank-binding kinase 1 followed by downstream signal activations including the nuclear factor- κ B pathway and type I interferon production^{6,7}. Enhanced interferon production induces cytotoxic effects directly against cancer cells, as well as activating dendritic cell (DC) maturation and promoting CD8⁺ T cell priming in tumour-draining lymph nodes (TDLN)⁸⁻¹⁰. Throughout these actions, STING agonists shift anti-tumour immunity from immunologically silenced "cold" tumours to active "hot" tumours¹¹. By doing so, they have been considered as synergistic agents for immune checkpoint inhibitors targeting PD-1 and PD-L1, which rescue exhausted T cells to kill tumour cells¹²⁻¹⁴. Although STING agonists has been tried for cancer therapy under tremendous attention, underlying mechanisms behind the anti-tumour effects are yet poorly understood^{15,16}.

Murine-specific STING agonist, 5,6-dimethylxanthenone-4-acetic acid (DMXAA), which also has been known as a tumour vascular disrupting agent (VDA), exerts a potent anti-tumour effect that is caused by tumour endothelial cell (EC)-specific apoptosis and extensive haemorrhage within tumour¹⁷. Tumour antigen release from dead tumour cells is the first step in adaptive immunity generation; therefore, tumour vascular destruction by STING agonists might be indispensable for achieving sufficient anti-tumour immunity^{18,19,20}. Notably, TNF α has been proposed as a mediator of STING-induced tumour EC apoptosis^{21,22}. However, the source of TNF α upon STING activation remains unknown, and it is unclear why STING activation triggers apoptosis specifically to tumour ECs. Moreover, most previous studies have been conducted in

subcutaneous tumour implantation models, which have limited ability to reflect human primary tumours^{23–25}.

Activation of intracellular AKT signalling plays a critical role in survival in several cell types, including ECs^{26–28}. Of note, AKT is a major downstream molecule for conveying intracellular signalling of vascular growth factors and their receptors including angiopoietin-1/Tie2 and VEGF-A/VEGFR2^{29,30}, which are key molecules in tumour angiogenesis. Accordingly, AKT pathway has been considered an attractive therapeutic target since AKT hyper-activation is associated with tumour aggressiveness and poor response to treatment. However, despite promising results in preclinical models, clinical trials of AKT inhibitors have failed to prove effectiveness, and none of the tested agents are currently used for cancer treatment^{31,32}.

In the present study, we aimed to explore how STING activation initially affects tumour vasculatures implanted tumours, as well as in spontaneously growing breast cancer and melanoma. We additionally present a rationale for how co-treatment with an AKT inhibitor may help treat spontaneous primary tumours that are refractory to STING agonist therapy.

Discussion: In this study, we examined the poorly understood role of the STING agonist cGAMP as a tumour-specific VDA, and demonstrated that this early activity of cGAMP is a key for establishing anti-tumour effect. We also showed that refractoriness of i.t. STING therapy could be relieved by addition of AKTi through effective tumour EC apoptosis in spontaneous tumours (**Extended Data Fig. 12**).

One representative VDA, flavone acetic acid, selectively block vascular flow in the tumour core by promoting tumour EC apoptosis, leading to powerful anti-tumour effect in murine tumour models⁵². Its derivatives have been developed from the trait of its chemical structure possessing amenable site with active region of xanthone-4-acetic acid, which gave rise to the development of DMXAA as potent as 16-fold⁵³. Interestingly, the scheme of DMXAA on anti-tumour effect has changed from VDA to immunotherapeutic agent as a STING agonist^{54,55}. Currently, i.t. STING agonist therapy is generally recognized to increase the production of type I interferon in tumours, which promotes CD8⁺ T cell priming and infiltration into the tumour, eventually converting immunologically “cold” tumours to “hot” tumours^{9,14,20}. Therefore, STING therapy has been considered a promising treatment for use in combination with immune checkpoint inhibitors, since they are more effective in hot tumours than cold tumours⁵⁶. However, throughout this study, we addressed the strong and rapid apoptotic role of STING agonist, cGAMP, in tumour ECs as a VDA. Through scRNA-seq and bulk RNA-seq analyses of the tumour ECs, we revealed that tip-like and proliferative EC subpopulations were vulnerable to STING activation-induced apoptosis. More importantly, we found that TAMCs-derived TNF α was a key mediator of the extensive apoptosis of tumour ECs (**Extended Data Fig. 12**).

STING plays multifaceted roles in tumour progression depending on target cells such as tumour cell or cells composing tumour microenvironment such as ECs, cancer-associated fibroblasts, and immune cells^{3,33–35}. For instance, Yang *et al.*¹⁴ have recently reported that STING agonists normalize tumour vessels mediated by up-regulating the genes related to type I/II interferon and vascular stabilization and enhancing pericyte coverage. However, it occurs at several days after i.t. injection with a prior and marked reduction of tumour ECs. In comparison, this study shows that

STING agonist strongly and rapidly induces tumour EC apoptosis within a day after the injection. Moreover, our findings indicate that the massive tumour cell death, following extensive cGAMP-induced tumour EC apoptosis, led to sufficient TICD8TC expansion and suppressed tumour growth in implanted tumours. On the other hand, when tumour ECs were resistant to cGAMP-induced apoptosis, the sparse tumour cell death led to failures of TICD8T expansion and suppression of tumour growth in spontaneous tumours (**Extended Data Fig. 12**).

AKT activation plays a central role in survival in several cell types, including ECs^{26–28}. Our analysis actually revealed that the tumour ECs of human breast ductal adenocarcinomas retain highly phosphorylated AKT compared with the ECs in adjacent normal tissue region. Nevertheless, our findings indicate that AKTi alone did not induce apoptosis in tumour ECs and whole tumour cells in spontaneous tumours. We hypothesized that AKT inhibition might enforce STING activation-induced apoptosis of tumour ECs in primary tumours based on our RNA-seq data and a previous report⁴³. Indeed, AKTi treatment abrogated unresponsiveness to STING therapy, mediated through effective apoptosis of tumour ECs. Moreover, AKTi potentiated the cGAMP-induced anti-tumour effect, which partly could be mediated by enhancing the innate immune responses of tumour cells and bystander cells to cGAMP (**Extended Data Fig. 12**). Supporting this scenario, previous studies⁵⁷ demonstrated that intracellular STING signalling and AKT pathways are antagonistically interacted for tumour progression mediated through innate immune responses, while AKTi increases T cell persistency and infiltration as well as early memory phenotype in tumour sites to enhance anti-tumour effect^{58,59}.

In conclusion, our present findings unravel long-standing questions regarding the mechanism underlying STING activation-induced tumour EC apoptosis. Our results highlight the critical role of STING agonist as a VDA, which results in extensive tumour cell death releasing massive tumour antigen, supporting the activation of anti-tumour immunity. Based on these findings, we propose that combination treatment of STING agonist and AKT inhibitor should be considered for clinical human tumours, which could be refractory to single therapy with either agent alone.

4. Among others, the authors should discuss a recent study in JCI, where LLC as well as breast MMTV-PyMT tumor models were extensively used for treatment with STING agonists (PMID: 31343989) e.g., do they also see a recovery of ECs and tumor vessel normalization at later timepoints after treatment as was reported in that study? The authors should also remove from their manuscript data already published elsewhere (such as extended data Fig.2 and more) and focus on the novel aspects of their own study.

Yes, we have seen tumour vessel normalization (for example, increase in NG2+ pericyte coverage) at 7 days after i.t cGAMP in the LLC tumour. Because the observation time points and the interpretations on the data between our study and their study (Yang et al., *J Clin. Invest.*, 2019) are different, we would like to keep our data as they are, while discussing the previous study in the revised Discussion.

Discussion (p 14-15): For instance, Yang *et. al.* ¹⁴ have recently reported that STING agonists normalize tumour vessels mediated by up-regulating the genes related to type I/II interferon and vascular stabilization and enhancing pericyte coverage. However, it occurs at several days after i.t. injection with a prior and marked reduction of tumour ECs. In comparison, this study shows that STING agonist strongly and rapidly induces tumour EC apoptosis within a day after the injection.

5. Tumor ECs are stained by CD31, but how are tumor cells identified? If “tumor cells” are identified just as a whole tumor area, the authors should instead call it by its name - these are not tumor cells, but a whole bulk of tumor tissue including all the stromal cell types that are present. Statements like: “Extensive apoptosis was observed in 24% of tumour ECs, and in 27% of tumour cells (Extended Data Fig. 1i–k)” are not correct.

This is a valid point that needs to be addressed. As the reviewer pointed out, the term “tumour cells” is not correct. We replaced it with “whole tumour cells” following its definition as “all cells within the tumour” (hereafter described as “whole tumour cells”).

In Results (p 5): Moreover, cGAMP-treated tumours exhibited extensive apoptosis [43% of all cells within tumour (hereafter described as “whole tumour cells”)], while PBS-treated tumours showed only mild apoptosis (6% of whole tumour cells) (Fig. 1e, f).

6. Following-up on the previous point, the percentages (and also fold changes in other cases) specified for each of these experiments make the text very difficult to read.

We believe the descriptions regarding the percentages and fold changes make understandable to be read and readily deliver more accurate information for the readers.

7. In their scRNAseq experiment, the authors show in Fig. 2d that single cells isolated from PBS-treated and cGAMP-treated animals overlap. This is also reflected in the clustering analysis, where no cluster is specific for either PBS- or cGAMP-treated group, indicating that there indeed is no difference. In light of this result, comparisons of gene expression between these two samples make little sense. Quantitative comparison of expression levels can however be done on a protein level e.g. immunostaining of tissue slides, as the authors did in case of SELP staining.

We appreciated this critical comment. To compare these two datasets, we aimed to identify cell populations that are shared while controlling for technical variabilities

such as batch effects. Therefore, we integrated two datasets following functions implemented in R package Seurat. As a result, cell populations with shared expression of marker genes are clustered together. Although the cell types were similar between datasets, we observed substantial changes in the transcriptome and the ratio of each cell type as we have shown in revised Extended Data Fig. 3.

Following the reviewer's comment, we performed additional immunostaining of placental growth factor (PGF), a specific marker for tip-like ECs, in the tumours to quantify the populations of tip-like ECs. (Revised Fig. 4i, j). The proportion of tip-like ECs was 41% less in cGAMP-treated tumours compared with PBS-injected tumours, which is consistent with the data of scRNA-seq.

In (Result 7): Accordingly, compared with PBS-treated ECs, the cGAMP-treated ECs had a 5.2-fold higher population of selectin P⁺ (SELP⁺, a representative marker for stalk-like ECs) ECs, while they had a 41% less population of placental growth factor⁺ (PGF⁺, a representative marker for tip-like ECs) ECs (Fig. 4g-j).

Fig. 4. Tumour ECs of implanted LLC tumours are vulnerable to STING activation-induced apoptosis. g-j, Representative images and comparisons of SELP⁺/CD31⁺ stalk-like ECs and PGF⁺/CD31⁺ tip-like ECs (white arrowheads) between PBS- and cGAMP-treated tumour ECs. Scale bars, 50 μ m. Each dot indicates a value from one mouse and n = 7-8 mice/group from four independent experiments. Horizontal bars indicate mean \pm SD. *P* values by two-tailed t-test.

8. In Fig. 2f the authors show fractional distribution of individual clusters. The preparation of single cell suspensions used for scRNAseq is not specified in the methods section, but from the context it seems that the analysis was done on n=1 PBS-treated versus n=1 cGAMP-treated sample. If that is the case, any conclusions about the fraction size in the two samples are irrelevant – differences between the two samples might be just caused by chance. In case the authors want to compare the fractional contribution between conditions, they would need to generate at least three samples of each condition. The authors therefore need to refrain from quantitative claims about the distribution of EC phenotypes in their manuscript, based on their scRNAseq data (e.g. "compared to PBS-treated ECs, the cGAMP-treated ECs exhibited a 13% increase of stalk-like ECs, and 9% and 4% reductions of tip-like and proliferative ECs, respectively (Fig. 2f)"). Another reason, why the comparison of fractions is not correct is that the treatment induces massive apoptosis of ECs. These dying

cells are not captured in sequencing droplets. Hence, we are looking at an unknown fraction of surviving cells. How can we know that the stalk-like cell fraction increased, and the difference is not caused by a different fraction of apoptotic cells in this sample?

This is valid point to be addressed. Because the sample size of each group was 1, we tried to refrain from making quantitative claims in the revised manuscript. However, when we additionally performed scRNA-seq on the ECs of other implanted breast tumour (n=2) in response to comment 1, we found similar changes of subpopulations in the ECs of the tumours treated with cGAMP compared with those treated with PBS (please see our response to comment 1). Therefore, it was unlikely to have occurred by a chance. Moreover, the comparative analysis results from GSEA on the ECs of implanted LLC tumours treated with cGAMP versus PBS and gene ontology analysis on implanted versus spontaneous breast tumour ECs revealed differential transcriptional responses (Extended Data Fig. 3 and 8). Of note, the genes related to apoptosis were enriched in the ECs of the tumours treated with cGAMP, implying that the tip-like ECs are vulnerable to STING agonist. Furthermore, because there was a lack of proliferative ECs mainly due to apoptotic death in the cGAMP-treated tumours (Extended Data Fig. 8c), we could not analyse the character of the proliferative ECs. Finally, as responded to comment 8, the immunostaining for SELP and PGF indicating stalk-like and tip-like ECs showed that stalk-like EC population was increased while tip-like EC population was reduced (Revised Extended Data Fig. 3g-j). Together, we validated and confirmed our claims by additional experiments and descriptions into the revised manuscript. Because we responded the most of comments in above, we respond to the rest of comments at here by showing additional results in below.

Extended Data Fig. 8. Implanted breast tumour ECs are more susceptible to cGAMP-induced apoptosis and more inflammatory compared with spontaneous breast tumour ECs **a, b**, Heatmap visualizing distinctive expression profiles in PBS- or cGAMP-treated tumour ECs derived from both spontaneous and implanted tumours. n = 2 mice/each group. Scaled expression levels of top ten differentially expressed genes for indicated clusters are shown. **c**, Gene ontology (GO) enrichment analyses on highly expressed genes in implanted tumour ECs compared with spontaneous tumour ECs. Left and right panels represent GO enrichment analysis results on PBS- and cGAMP-treated tumour ECs. n = 2 mice/each group.

9. In Fig. 2e authors show that PGF is a marker of “stalk-like cells” while in Fig. 5e this gene is a part of a “tip-like” EC signature. As Placental growth factor is a recognized tip cell marker (del Toro et al., 2010; Strasser et al., 2010; Zhao et al., 2018, Goveia et al., 2020) the heatmap shown in Fig. 2 suggests that the clustering is sub-optimal, i.e. cluster of stalk-like ECs probably contains also part of tip ECs.

As this reviewer and the reviewer 3 indicated, we made an error by missing one

gene *FBLN2* in the list of top 10 highly expressed genes of the stalk-like EC cluster. As corrected, *PGF* (red underline) is the number 1 gene in the list of top 10 highly expressed genes of the tip-like EC cluster, and *PGF* is a definitive tip EC marker.

10. To facilitate an easy exploration of their data by the community, the authors should provide the reader with a list of top-ranking genes per cluster in their analysis in a supplementary table. Also, they should make available the quality metrics for their experiment to allow assessment of data quality – details of sample preparation (including info from how many animals each sample was prepared), number of sequenced cells per sample, number of genes/UMIs per cell, etc.

We appreciate this constructive comment. The top ranking genes per cluster are different between different tumours and treatments, we rather magnified the gene names in the right side of heatmap panels for the community. We also additionally provided the quality metrics (table. 1) and details of preparation in revised manuscript, Figure legends, and Methods.

11. Finally, the authors include scRNAseq data on LLC implanted tumors and bulk RNAseq data on MMTV-PyMT tumors, as these data were not integrated and the models are not comparable, the conclusions like “These findings recapitulated the scRNA-seq data indicating that tip-like and proliferative EC

subpopulations were vulnerable to STING activation-induced apoptosis,..” are not substantiated. Similarly, the authors cannot make conclusions about tip and stalk like cell susceptibility to STING agonists in breast tumor as they don’t know what the expression profile of EC phenotypes in breast tumors is. Are there stalk like cells and tip cells even present as separate clusters in breast tumors?

We appreciate this critical and constructive comment. As responded to the comment 1, we newly performed scRNA-seq on tumor ECs of spontaneous tumour (*MMTV-PyMT* breast tumour) and their implantation tumours to see whether the findings derived from implanted LLC tumours could be recapitulated. As expected, the findings (susceptibility to STING agonist, expression profiles, cluster, and subpopulations) of implanted breast tumours were basically similar to those from LLC tumours, so that the findings were recapitulated. Moreover, five distinct EC clusters- stalk-like, tip-like, proliferative, arterial and AQP7⁺ ECs are present in the spontaneous breast tumours. We believe this comment is fully responded in the response to comment 1.

12. In Figure 3g, the authors included results only from cGAMP treated animals. Results in PBS-treated animals should be included as a control.

Following the reviewer’s comment, the data of PBS-treated tumours were included into revised Fig. 5e, f accordingly.

13. The order of panels as well of accompanying text describing data in Extended Figure 5 is mixed-up. At this point this section is difficult to interpret.

We rearranged the panels in Extended Data Fig. 5 according to the order of description in the text.

14. How well is cGAMP/Akt1i combination tolerated by the animals? Clearly, Akt1i is toxic to normal HUVEC (Extended data fig. 7a). Were any adverse effects observed?

The mice treated with combined cGAMP and AKTi were relatively well tolerated, as definitive signs of drug-induced adverse effects such as hair loss, eyelid swelling or body weight loss were not detected.

Minor comments:

1. The p values indicated in figures sometimes do not fit in the image (e.g. Fig. 3e) or it is unclear to which comparison they refer (Fig.6g). This makes the graphs overcrowded and difficult to interpret.

Thank you for this constructive comment. We redrew the comparison brackets or removed the p values not only in Fig. 3e and Fig. 6g but also in other Figures for clearer comparisons in the revised manuscript.

2. Axis titles are missing on several occasions – Fig. 2d, Fig. 6e-g.

We apologize for the missing labelling on several occasions in Figures. We corrected the Figures by additional labelling in the revised manuscript.

3. Figure 1l, please use consistent color coding for CD31 staining.

We changed the colour in red in Figure 1l for consistency.

4. In Figure 6, the AKT comes out of the blue sky. Could they explain better the rationale for the use of AKT inhibitors in the manuscript (not only in the discussion, but also in the introduction). Can the authors find some indications in their RNAseq data leading to the AKT pathway? It would make this part better integrated in the story.

We already described our RNAseq data leading to the AKT pathway in the original manuscript (page 8, line 6-9). Nevertheless, in response to reviewer's suggestion, we explained more about the rationale for the use of AKTi in the revised Introduction and Discussion.

Introduction: Activation of intracellular AKT signalling plays a critical role in survival in several cell types, including ECs²⁶⁻²⁸. Of note, AKT is a major downstream molecule for conveying intracellular signalling of vascular growth factors and their receptors including angiopoietin-1/Tie2 and VEGF-A/VEGFR2^{29,30}, which are key molecules in tumour angiogenesis. Accordingly, AKT pathway has been considered an attractive therapeutic target since AKT hyper-activation is associated with tumour aggressiveness and poor response to treatment. However, despite promising results in

preclinical models, clinical trials of AKT inhibitors have failed to prove effectiveness, and none of the tested agents are currently used for cancer treatment^{31,32}.

In Results (p 9-10): Importantly, genes related to the PI3K-AKT-mTOR pathway (the strongest EC survival pathway against EC apoptosis) were enriched following cGAMP treatment in *MMTV-PyMT* and implanted breast tumours but not in normal mammary fat pads (**Extended Data Fig. 7d**), which is consistent with a previous report⁴³.

In Results (p11): Based on the bulk RNA sequencing data (**Extended Data Fig. 7d**) and a previous report⁴³, we hypothesized that inhibition of AKT signalling could potentiate spontaneous tumour to combined TNF α and IFN γ -induced tumour EC apoptosis. In fact, immunohistochemical analysis on the tissue array of human breast ductal adenocarcinomas revealed that tumour ECs in tumour core region had highly activated AKT compared with the ECs in adjacent normal tissue region (**Extended Data, Fig. 9**).

5. Figure 6 panels i-j-k show exactly the same data. While showing the tumor growth in individual animals might have at least some added value, the bar graph shown in k does not bring anything new to the story and should be removed.

Following the reviewer's comment, Fig.6k was removed in the revised manuscript.

6. In Extended Data Fig.6 it is not clear which sample represents which plot.

We additionally included related information and labelling accordingly in the revised manuscript.

References for responses to the comments of reviewer 4

Bieler, G., Hasmim, M., Monnier, Y., ...Lejeune, F., & Rüegg, C. (2007). Distinctive role of integrin-mediated adhesion in TNF-induced PKB/Akt and NF- κ B activation and endothelial cell survival. *Oncogene*, 26(39), 5722–5732. <https://doi.org/10.1038/sj.onc.1210354>

Dankort, D., Curley, D. P., Cartlidge, R. A., Nelson, B., Karnezis, A. N., Damsky, W. E., You, M. J., DePinho, R. A., McMahon, M., & Bosenberg, M. (2009). BrafV600E cooperates with Pten loss to induce metastatic melanoma. *Nature Genetics*, 41(5), 544–552. <https://doi.org/10.1038/ng.356>

Riedel, A., Shorthouse, D., Haas, L., Hall, B. A., & Shields, J. (2016). Tumor-induced stromal reprogramming drives lymph node transformation. *Nature Immunology*, 17(9), 1118–1127. <https://doi.org/10.1038/ni.3492>

Yang, H. *et al.* STING activation reprograms tumor vasculatures and synergizes with VEGFR2 blockade. *J. Clin. Invest.* **129**, 4350–4364 (2019).

REVIEWERS' COMMENTS

Reviewer #1 (Remarks to the Author):

The authors have satisfactorily addressed my concerns

Reviewer #2 (Remarks to the Author):

Most of my concerns raised in reviewing the previous manuscript have been adequately addressed in the revised manuscript.

Reviewer #3 (Remarks to the Author):

The authors have been very responsive to the issues raised by the reviewers and sufficiently addressed our previous concerns.

Reviewer #4 (Remarks to the Author):

General: The authors put an impressive effort into revising their manuscript and resolved most of the issues/comments from the first revision. There are few remaining issues described below.

Comments:

1. As far as this reviewer is concerned, the authors may keep the percentages and fold changes throughout the main text. Yet, I remain convinced that this does not help the comprehensibility of their work. There is however one exception, where the use of quantitative language is scientifically and statistically incorrect, and must be removed, see below in point 2.
2. Figure 4f, the fraction of subpopulations cannot be quantitatively estimated from single cell data generated in $n=1$ sample in each group. Please note that solving this issue, already pointed out in the first round of review, was not meant as a "nice to have" but rather a MUST. If the authors really want to state the percentages for fractional distribution of each cluster, they must accompany them with a proper statistical evaluation. While that would necessarily require repeating each single cell experiment at least two more times (generating 3 independent scRNAseq samples for each of the groups), the reviewer suggests removing these numbers and rephrasing the text that "scRNAseq SUGGESTED an increase/ decrease of a particular population". It is indeed nice that the authors were able to support these suggestions with protein level data, and that they found similar changes in the newly performed scRNAseq. This however does not change anything about the fact that they cannot statistically evaluate their experiments in this way. Importantly, the same applies to the newly added scRNAseq experiment in Figure 8.
3. Figure 8 presents newly generated scRNAseq data in spontaneous and implanted breast tumors. This data strongly improved the manuscript, but the presentation and analysis of data is sub-optimal. (i) In 8d,e the authors discuss differences between clusters present in PBS vs. cGAMP treated groups but they did not show a UMAP, where all cells would be analyzed together. This is needed to evaluate the differences between treated and control samples. Please add these UMAPs to figures and adapt the text and data interpretation accordingly. (ii) What differences did the authors find in EC transcriptional profiles between spontaneous and implanted tumors?

Minor comments:

1. The term "Whole tumor cells", which authors use instead of the previous incorrect "tumor cells" unfortunately does not explain better that the authors mean "all cells within the tumor". I suggest that the authors consult a language professional on this matter. More generally, the manuscript (as most other manuscripts at this stage) would benefit from a proper language correction, which I assume will be taken care of by the editorial office.
2. The quality of presentation in some data tables in the supplement is not good, e.g., titles like "eeks after b" on page 19. Please correct throughout.

We have modified the manuscript to address the issues raised by the reviewer 4 as following.

Reviewer #1-3

Reviewer 1: The authors have satisfactorily addressed my concerns

Reviewer 2: Most of my concerns raised in reviewing the previous manuscript have been adequately addressed in the revised manuscript.

Reviewer 3: The authors have been very responsive to the issues raised by the reviewers and sufficiently addressed our previous concerns.

We appreciate this favourable and encouraging comments.

Reviewer #4 (Remarks to the Author):

General: The authors put an impressive effort into revising their manuscript and resolved most of the issues/comments from the first revision. There are few remaining issues described below.

We appreciate this favourable and encouraging comments.

Comment 1: As far as this reviewer is concerned, the authors may keep the percentages and fold changes throughout the main text. Yet, I remain convinced that this does not help the comprehensibility of their work. There is however one exception, where the use of quantitative language is scientifically and statistically incorrect, and must be removed, see below in point 2.

Thank you for this thoughtful and constructive comment. In response to this comment, we removed all quantitative languages throughout the revised Results except the text related to the point 2. For instances,

(p5) Likewise, the systemic murine STING agonist DMXAA induced **extensive** apoptosis in 47% of ECs of tumour vessels, and in 47% of whole tumour cells at 24 h (Supplementary Fig. 1a–c).

(p5) Further analyses revealed that cGAMP **strongly** increased vascular leakage by 2.8-fold, induced red blood cell leakage by 12.2-fold, reduced blood perfusion by 54%, and increased hypoxia by 9.2-fold at 24 h (Fig. 1l, m). Similar findings were observed in the B16F10 melanoma implantation model. Intratumoural cGAMP injections **markedly** suppressed tumour growth by 95% (Supplementary Fig. 1f–h). **Extensive** Apoptosis was observed in 24% of tumour ECs, and in 27% of whole tumour cells (Supplementary Fig. 1i–k).

(p9) Similar to those in implanted *MMTV-PyMT* cells breast tumour, i.t. cGAMP injection induced **extensive** apoptosis of tumour ECs (48%) and whole tumour cells (38%) and **marked** suppression of tumour growth by 81% in implanted 4T1 cells breast tumour (Supplementary Fig. 6b-f).

Comment 2: Figure 4f, the fraction of subpopulations cannot be quantitatively estimated from single cell data generated in n=1 sample in each group. Please note that solving this issue, already pointed out in the first round of review, was not meant as a "nice to have" but rather a **MUST**. If the authors really want to state the percentages for fractional distribution of each cluster, they must accompany them with a proper statistical evaluation. While that would necessarily require repeating each single cell experiment at least two more times (generating 3 independent scRNAseq samples for each of the groups), the reviewer suggests removing these numbers and rephrasing the text that "scRNAseq **SUGGESTED** an increase/ decrease of a particular population". It is indeed nice that the authors were able to support these suggestions with protein level data, and that they found similar changes in the newly performed scRNAseq. This however does not change anything about the fact that they cannot statistically evaluate their experiments in this way. Importantly, the same applies to the newly added scRNAseq experiment in Figure 8.

We fully agree with this comment. Following the reviewer's indication, we removed percentage changes (numbers) and rephrased the text in the revised Results as below.

(p7) Notably, the scRNA-seq analysis suggested that, compared to PBS-treated ECs, the cGAMP-treated ECs exhibited increase of stalk-like EC population but reductions of tip-like and proliferative EC populations (**Fig. 4f**).

(p10) The scRNA-seq analysis suggested that cGAMP treatment markedly reduced the population of proliferative ECs in spontaneous tumour, while it markedly reduced both proliferative and tip-like ECs in implanted tumour (**Fig 8f, g**).

Comment 3. Figure 8 presents newly generated scRNAseq data in spontaneous and implanted breast tumors. This data strongly improved the manuscript, but the presentation and analysis of data is sub-optimal. (i) In 8d,e the authors discuss differences between clusters present in PBS vs. cGAMP treated groups but they did not show a UMAP, where all cells would be analyzed together. This is needed to evaluate the differences between treated and control samples. Please add these UMAPs to figures and adapt the text and data interpretation accordingly. (ii) What differences did the authors find in EC transcriptional profiles between spontaneous and implanted tumors?

We appreciate this constructive comment. In the revised manuscript, we integrated all the scRNA-seq data and presented it as a single UMAP plot (Supplementary Fig. 8a) showing the non-hierarchical clustering with an adequate description/interpretation accordingly. Of note, the ECs were clustered into five

distinctive groups including tip-like, stalk-like, proliferative, arterial and AQP7⁺ ECs. Moreover, we additionally included the differences and their implications in the EC transcriptional profiles between spontaneous and implanted tumours as following.

Extended Figure 8. a, UMAP plots integrating all tumour ECs of spontaneous and implanted breast tumours treated with PBS or cGAMP.

(p10-11) Five distinct EC clusters- stalk-like, tip-like, proliferative, arterial and AQP7⁺ ECs were present in the pooled ECs of both spontaneous and implanted tumour treated with PBS or cGAMP (Fig. 8d, e and Supplementary Fig. 8a-c). AQP7⁺ ECs are a breast-specific, fully-differentiated subpopulation that highly expresses AQP7, CD35 and FABP5 mRNAs and actively participates in the glycerol and fatty acid metabolism and transport⁴⁴. Distinctiveness of 5 clustering was largely blunted in the pooled ECs treated with cGAMP compared with those treated with PBS (Fig. 8d, e), implying that tumour ECs were largely affected by i.t. STING agonist. The scRNA-seq analysis suggested that cGAMP treatment markedly reduced the population of proliferative ECs in spontaneous tumour, while it markedly reduced both proliferative and tip-like ECs in implanted tumour (Fig 8f, g). The latter findings are similar to those in implanted LLC tumours (Fig. 4f-j). Gene ontology enrichment analysis on the ECs of implanted tumour *versus* spontaneous tumour revealed differential transcriptional responses to both PBS and cGAMP treatment (Supplementary Fig. 8d). The genes related to cellular responses to cytokine stimulus and interferon- γ were enriched in the all EC clusters of the implanted tumours treated with PBS (Supplementary Fig. 8d), implying that implanted tumour ECs could be more responding to anti-tumour immunotherapy compared with spontaneous tumour ECs. Of note, the genes related to apoptosis were enriched in the tip-like ECs of the implanted tumours treated with cGAMP (red underline in Supplementary Fig. 8d), implying that the tip-like ECs are vulnerable to STING agonist. Because there was a lack of proliferative ECs mainly due to apoptotic death in the cGAMP-treated tumours (Supplementary Fig. 8d), we could not analyse the character of the proliferative ECs. These findings denote that tip-like and proliferative ECs are vulnerable to STING activation-induced apoptosis, and these can partly explain why only the ECs of implanted, but not spontaneous, breast cancer underwent apoptosis following cGAMP treatment. Moreover, the ECs retaining maturation gene profiles within tumours such as arterial, stalk-like and AQP7⁺ ECs

(Fig. 8d, e and Supplementary Fig. 8a, b) are likely to be resistant to STING activation-induced apoptosis.

Minor comments:

1. The term "Whole tumor cells", which authors use instead of the previous incorrect "tumor cells" unfortunately does not explain better that the authors mean "all cells within the tumor". I suggest that the authors consult a language professional on this matter. More generally, the manuscript (as most other manuscripts at this stage) would benefit from a proper language correction, which I assume will be taken care of by the editorial office.

Although we consulted the term "whole tumor cells" to the oncologists and the language professionals, we could not obtain any better term to replace it. Therefore, we would like to keep this term in this manuscript.

2. The quality of presentation in some data tables in the supplement is not good, e.g., titles like "eeks after b" on page 19. Please correct throughout.

We thoroughly corrected the errors in the revised supplement.